# Bridging VMP and CEP:
# Theoretical Insights for Connecting Different Approximate Bayesian Inference Methods

## Abstract

Approximate Bayesian inference (ABI) methods have become indispensable tools in modern machine learning and statistics for approximating intractable posterior distributions. Despite the related extensive studies and applications across diverse domains, the theoretical connections among these methods have remained relatively unexplored. This paper takes the first step to uncover the underlying relationships between two widely employed ABI techniques: the variational message passing (VMP) and the conditional expectation propagation (CEP) methods. Through rigorous mathematical analysis, we demonstrate a strong connection between these two approaches under mild conditions, from optimization as well as graphical model perspectives. This newly unveiled connection not only enhances our understanding of the performance and convergence properties of VMP and CEP, but it also facilitates the cross-fertilization of their respective strengths. For instance, we prove the convergence of CEP under mild conditions and enable an online variant of VMP through this connection. Furthermore, our findings provide insights into the underlying relationships and distinctive characteristics of other ABI methods, shedding new light on the understanding and development of more advanced ABI techniques. To validate our theoretical findings, we derive and analyze various ABI methods within the context of Bayesian tensor decomposition, a fundamental tool in machine learning research. Specifically, we show that these two approaches yield the same updates within this context and illustrate how the established connection can be leveraged to construct a streaming version of the VMP-based Bayesian tensor decomposition algorithm.

## 1 Introduction

Approximating difficult-to-compute posterior distributions is one of the most fundamental challenges in modern machine learning and statistics. To address this challenge, approximate Bayesian inference (ABI) has made significant progress over the years (Blei et al., 2017; Zhang et al., 2019; Theodoridis, 2025; Cheng et al., 2022b; Murphy, 2022), showcasing remarkable performance. These methods have found extensive applications in diverse domains, such as bioinformatics (Daunizeau et al., 2014; Grønbech et al., 2020), computer vision (Chan & Vasconcelos, 2009; Soh & Cho, 2022; Fan et al., 2022), and speech recognition (Cohen & Smith, 2010; Xue et al., 2021). Variational inference (VI) (Jordan et al., 1999; Wainwright & Jordan, 2008) and expectation propagation (EP) (Minka & Picard, 2001), along with their modern variants (Zhang et al., 2019; Broderick et al., 2013; Li et al., 2015; Wang & Zhe, 2020; Vehtari et al., 2020), are two prominent classes of ABI methods widely used in practice.

The fundamental principle of VI involves formulating a family of distributions and subsequently finding the member within that family that best approximates the target distribution (Blei et al., 2017; Bishop, 2006; Theodoridis, 2025). The closeness between distributions is typically measured using the Kullback-Leibler (KL) divergence. In the context of the mean-field VI, the variables are assumed to be mutually independent and governed by their respective distributions. By decomposing the model evidence, VI transforms its objective into optimizing the evidence lower bound (ELBO). When analytical expectations can be derived,

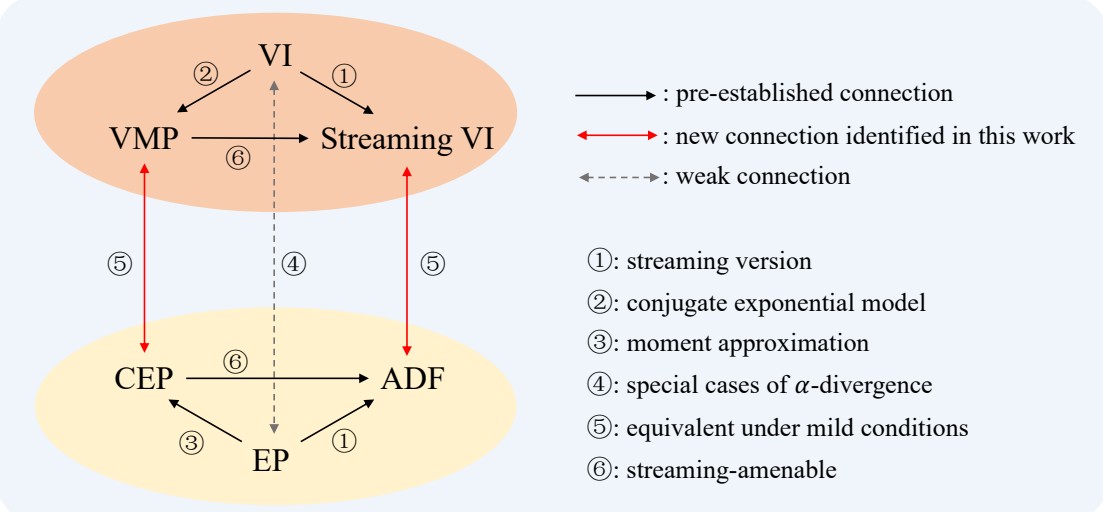

Figure 1: Connections among different approximate Bayesian inference (ABI) methods. Red arrows indicate the new connections established in this paper. VI denotes variational inference, VMP stands for variational message passing, EP refers to expectation propagation, CEP is conditional expectation propagation, and ADF represents assumed density filtering. The sixth arrow indicates that the corresponding algorithm can be extended to a streaming setting.

VI demonstrates favorable accuracy and speed. It also guarantees convergence to a local optimum (Beal, 2003). The streaming version of VI has also been developed to handle the streaming data case, as show by the first arrow in Fig. 1, but the algorithm design is not straightforward and demands additional effort (Broderick et al., 2013).

When the distributions of variables are restricted to the exponential family (Brown, 1986) and possess conjugate properties, mean-field VI can be implemented based on the convenient and efficient message-passing mechanism. The resulting algorithm is known as the variational message passing (VMP) (Winn et al., 2005), as shown by the second arrow in Fig. 1 . VMP operates by sending messages between nodes in the network and updating posterior beliefs through local operations performed at each node. By introducing additional variational parameters or utilizing approximation methods, VMP can be extended to models containing non-conjugate distributions (Winn et al., 2005; Wang & Blei, 2013). It also guarantees convergence and enables efficient evaluation of the model evidence (Winn et al., 2005).

EP is a generalized message-passing algorithm employed on factor graphs (Minka, 2013), which unifies and extends the concepts of assumed density filtering (ADF) (Maybeck, 1982) and loopy belief propagation (Frey & MacKay, 1997). The ADF can be viewed as a streaming or online version of EP, as shown by the first arrow in Fig. 1. In EP, we construct an approximation of the posterior by iteratively performing simple local computations which refine the factor that represents how each data point contributes to the posterior distribution. Notably, EP differs from VI in terms of the direction of the KL divergence. In various tasks, such as the clutter problem[1] and mixture weight estimation, EP has shown superior performance compared to VI (Zhou et al., 2023). Additionally, the local computations make EP amenable to parallelized and distributed computation, rendering it well-suited for addressing large-scale problems (Li et al., 2015; Hasenclever et al., 2017; Vehtari et al., 2020). However, applying EP encounters a critical challenge when dealing with models that have complex likelihoods, as the moment matching that is involved in the factor update procedure can become intractable. Additionally, convergence is not guaranteed as message passing updates are local (Vehtari et al., 2020).

---

[1]The clutter problem is about estimating the mean of a signal distribution when the data is mixed with noise from a known clutter distribution (Minka, 2013).

To address the computation barrier in EP, recent advances have introduced alternative approaches for moment computation in EP, such as the Monte Carlo simulations (Li et al., 2018) and the Laplace approximation (Smola et al., 2004). Unfortunately, these approximations often suffer from inefficiency and high computational costs, thereby diminishing the appeal of EP as a fast approximation method. Conditional expectation propagation (CEP) (Wang & Zhe, 2020) has recently emerged as a promising alternative, offering an efficient variant of EP. Instead of directly calculating the moments of the complete distribution, CEP first seeks the tractable and analytical conditional moments and then computes their expectations with respect to the approximate posterior of the remaining variables, as shown by the third arrow in Fig. 1. Like EP, CEP's local update nature makes it well-suited for large-scale datasets, but convergence guarantee remains an open question.

Since VMP and CEP are developed from different perspectives (VI and EP, respectively) and have distinct theoretical roots (different directions of the KL divergence), theoretical connections between them remain unexplored. To the best of our knowledge, the most related work is the power EP (Minka, 2004), which unifies the idea of VI and EP by utilizing the $\alpha$-divergence, as shown by the fourth arrow in Fig. 1. By adjusting the value of $\alpha$, it becomes possible to obtain an intermediate result between VI and EP. While power EP provides a general perspective that connects VI and EP, it does not fully uncover the intrinsic connections and differences between these two methods, nor does it consider the specific cases of CEP and VMP. Another related work is the Bayesian learning rule (Khan & Rue, 2023), which unifies different ABI methods through the natural gradient descent. However, it does not consider the EP algorithm and its variants.

This paper aims to unveil the underlying relationships between VMP and CEP, as shown the fifth arrow in Fig. 1. In particular, we demonstrate a strong connection between these two approaches from optimization as well as graphical model perspectives. This newly identified connection not only deepens our understanding of the performance and convergence properties of these two approaches but, also, it allows one to combine their repsective strengths. Additionally, it provides insights into the underlying relationships and distinct characteristics of other ABI methods, see the fifth arrow in Fig. 1.

Notably, the established connection provides a guarantee of the convergence of CEP, leveraging the corresponding property enjoyed by VMP. It turns out that the assimilation of the message factors in CEP leads to an increment of ELBO and ensures convergence. Additionally, the connection can also provide some insights into the convergence of the standard EP. Furthermore, the parallelized and distributed nature of CEP facilitates the seamless construction of an online or distributed variant of VMP and VI. This adaptivity enables the effective handling of large-scale datasets, particularly in scenarios involving continuous data streams or sequential data arrivals.

To corroborate our theoretical analysis, we present an example that showcases the application of VMP and CEP in the context of Bayesian tensor decomposition, which is a powerful tool in machine learning research and finds applications in various real-world scenarios (Cheng et al., 2022b;a; Fang et al., 2021b;a). In this particular context, besides demonstrating the connection between VMP and CEP, we also illustrate how this connection can be leveraged to develop a streaming version of the VMP-based tensor decomposition algorithm.

The remainder of this paper is organized as follows. Section 2 gives a brief review of different ABI methods and provides some useful lemmas. Section 3 contains the main theoretical results and some related extensions. In Section 4, using Bayesian tensor decomposition as an example, different ABI methods are derived and analyzed to validate our theoretical findings. Finally, Section 5 concludes with an overall discussion and suggestions for future research directions.

## 2    Preliminaries

This section provides a brief review of various ABI methods and presents some useful lemmas. Before delving into the details of each method, we introduce the general problem. Given a set of observations $\mathcal{D} = \{\mathbf{x}_1, \cdots, \mathbf{x}_N\}$ and a probabilistic model described via a set of latent variables $\boldsymbol{\theta}$, the joint distribution

can be expressed as

$$p(\boldsymbol{\theta}, \mathcal{D}) = p(\boldsymbol{\theta})p(\mathcal{D}|\boldsymbol{\theta}),$$

where $p(\boldsymbol{\theta})$ represents the respective prior distribution and $p(\mathcal{D}|\boldsymbol{\theta})$ denotes the data likelihood. The goal is to compute the posterior distribution, $p(\boldsymbol{\theta}|\mathcal{D})$, which can be expressed as

$$p(\boldsymbol{\theta}|\mathcal{D}) = \frac{p(\boldsymbol{\theta}, \mathcal{D})}{p(\mathcal{D})} = \frac{p(\boldsymbol{\theta}, \mathcal{D})}{\int p(\boldsymbol{\theta}, \mathcal{D})d\boldsymbol{\theta}},$$

where $p(\mathcal{D})$ denotes the model evidence. For many models of practical interest, it is infeasible to compute the posterior distribution directly due to the analytically intractable integration in the denominator. Therefore, approximation methods are essential in such cases. In this paper, we primarily focus on VI, EP, and their variants.

### 2.1 Variational Inference and Variational Message Passing

#### 2.1.1 Variational Inference (VI)

VI is a technique that approximates the posterior distribution by utilizing a probability distribution with density $q(\boldsymbol{\theta})$ from a tractable family of distributions $\mathcal{Q}$. The aim is to find the best variational approximation, $q^* \in \mathcal{Q}$, by minimizing the KL divergence between $q(\boldsymbol{\theta})$ and the true posterior $p(\boldsymbol{\theta}|\mathcal{D})$ (Cover, 1999), i.e.,

$$q^* = \min_{q \in \mathcal{Q}} \mathrm{KL}(q(\boldsymbol{\theta})\|p(\boldsymbol{\theta}|\mathcal{D})) = \min_{q \in \mathcal{Q}} \int q(\boldsymbol{\theta}) \ln \frac{q(\boldsymbol{\theta})}{p(\boldsymbol{\theta}|\mathcal{D})} d\boldsymbol{\theta}.$$

This transforms the inference task into an optimization problem, where the flexibility of the family $\mathcal{Q}$ controls the complexity of the optimization process. However, the objective function is not directly computable as it requires the model evidence. To overcome this challenge, VI employs a clever decomposition (e.g., Bishop, 2006; Theodoridis, 2025)

$$\ln p(\mathcal{D}) = \mathcal{L}(q) + \mathrm{KL}(q(\boldsymbol{\theta})\|p(\boldsymbol{\theta}|\mathcal{D})),$$

where

$$\mathcal{L}(q) = \int q(\boldsymbol{\theta}) \ln \frac{p(\boldsymbol{\theta}, \mathcal{D})}{q(\boldsymbol{\theta})} d\boldsymbol{\theta} \tag{1}$$

is the evidence lower bound (ELBO). Since the model evidence is a constant with respect to $\boldsymbol{\theta}$ and the KL divergence is non-negative, minimizing the latter is equivalent to maximizing $\mathcal{L}(q)$.

If there are no restrictions on $\mathcal{L}(q)$, the maximum of the ELBO occurs when $q(\boldsymbol{\theta})$ equals $p(\boldsymbol{\theta}|\mathcal{D})$, which, however, is intractable. Consequently, some restrictions on the functional form of $q(\boldsymbol{\theta})$ are required. In the context of the mean-field VI, the variables are assumed to be mutually independent, and each variable is governed by its own distribution. A typical member of the mean-field variational family can be expressed as (e.g., Blei et al., 2017; Theodoridis, 2025)

$$q(\boldsymbol{\theta}) = \prod_{m=1}^{M} q(\boldsymbol{\theta}_m). \tag{2}$$

Here the elements of $\boldsymbol{\theta}$ are partitioned into $M$ disjoint groups as the probability distribution factorized into $M$ groups, i.e., $\boldsymbol{\theta} = \{\boldsymbol{\theta}_1, \cdots, \boldsymbol{\theta}_M\}$. Then, the ELBO $\mathcal{L}(q)$ is optimized by iteratively updating each group in turn. Specifically, the optimal solution for each factor can be obtained by substituting (2) into (1), which gives

$$\begin{aligned}
\mathcal{L}(q) &= \int \prod_m q(\boldsymbol{\theta}_m) \left\{ \ln p(\boldsymbol{\theta}, \mathcal{D}) - \sum_m \ln q(\boldsymbol{\theta}_m) \right\} d\boldsymbol{\theta} \\
&= \int q(\boldsymbol{\theta}_m) \mathbb{E}_{q(\boldsymbol{\theta}_{\backslash m})}[\ln p(\boldsymbol{\theta}, \mathcal{D})] d\boldsymbol{\theta}_m - \int q(\boldsymbol{\theta}_m) \ln q(\boldsymbol{\theta}_m) d\boldsymbol{\theta}_m + \mathrm{const1} \\
&= -\mathrm{KL}\left(q(\boldsymbol{\theta}_m)\|\tilde{q}(\boldsymbol{\theta}_m)\right) + \mathrm{const1},
\end{aligned}$$

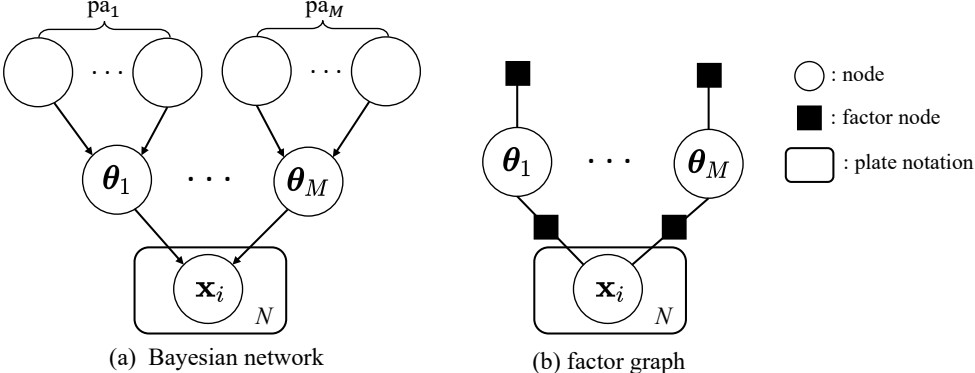

(a) Bayesian network  (b) factor graph

Figure 2: Two graphical model representations: (a) a Bayesian network used in variational message passing. Each latent variable $\boldsymbol{\theta}_m$ has a parent set $\text{pa}_m$, and the observed variables $\{\mathbf{x}_i\}_{i=1}^N$ are conditionally independent given the latent variables; (b) a factor graph used in expectation propagation. Circles denote latent variables $\{\boldsymbol{\theta}_1, \ldots, \boldsymbol{\theta}_M\}$ and observed variable $\mathbf{x}_i$, while squares represent factors. The plate indicates $N$ repeated observations.

where $\boldsymbol{\theta}_{\backslash m}$ represents the set of variables excluding the $m$th group and

$$\ln \tilde{q}(\boldsymbol{\theta}_m) = \mathbb{E}_{q(\boldsymbol{\theta}_{\backslash m})}[\ln p(\boldsymbol{\theta}, \mathcal{D})] + \text{const2}.$$

It can be seen that $\mathcal{L}(q)$ is optimized when the KL divergence equals to zero, which results in

$$\ln q^*(\boldsymbol{\theta}_m) = \mathbb{E}_{q(\boldsymbol{\theta}_{\backslash m})}[\ln p(\boldsymbol{\theta}, \mathcal{D})] + \text{const2} \tag{3}$$
$$= \mathbb{E}_{q(\boldsymbol{\theta}_{\backslash m})}[\ln p(\boldsymbol{\theta}_m | \boldsymbol{\theta}_{\backslash m}, \mathcal{D})] + \text{const3}.$$

After taking the exponential of both sides and normalizing, we obtain

$$q^*(\boldsymbol{\theta}_m) = \frac{\exp(\mathbb{E}_{q(\boldsymbol{\theta}_{\backslash m})}[\ln p(\boldsymbol{\theta}_m | \boldsymbol{\theta}_{\backslash m}, \mathcal{D})])}{\int \exp(\mathbb{E}_{q(\boldsymbol{\theta}_{\backslash m})}[\ln p(\boldsymbol{\theta}_m | \boldsymbol{\theta}_{\backslash m}, \mathcal{D})]) d\boldsymbol{\theta}_m}, \forall m. \tag{4}$$

Although the set of equations in (4) provides consistency conditions for maximizing the lower bound, they do not represent an explicit solution. This is because the optimum for each variable group $\boldsymbol{\theta}_m$ depends on the distributions of other groups, $\boldsymbol{\theta}_{\backslash m}$. Therefore, when applying VI, we typically seek a solution by first initializing all of the factors $q(\boldsymbol{\theta}_m)$ appropriately and then iteratively updating each factor, replacing the other variable groups with their current estimates. Convergence is guaranteed because the ELBO is convex with respect to each of the groups (e.g., Bishop, 2006). It is also worth noting that if the goal is to minimize the reverse KL divergence, i.e., $\text{KL}(p \| q)$, a closed-form solution for each variable group can also be derived; see Appendix B for details.

### 2.1.2 Variational Message Passing (VMP)

VMP is an implementation of the mean-field variational inference method tailored to conjugate-exponential models (Winn et al., 2005). It performs inference by passing local messages in a Bayesian network, allowing structured and modular updates of the variational distributions.

Before introducing the details of VMP, we briefly review the basic concepts of probabilistic graphical models. In a Bayesian network, the joint distribution over a set of variables is represented using a directed acyclic graph (DAG), where each node corresponds to a random variable, and directed edges represent conditional dependencies. The joint distribution factorizes as a product of conditional distributions, one for each node given its parents in the graph.

Fig. 2(a) illustrates a typical Bayesian network structure. The latent variables $\boldsymbol{\theta} = \{\boldsymbol{\theta}_1, \ldots, \boldsymbol{\theta}_M\}$ represent different components of the model parameters, each associated with its own set of parents, denoted by

pa$_m$. These parents may include hyperparameters or other latent variables. The observed data $\{\mathbf{x}_i\}_{i=1}^N$ are conditionally dependent on all or a subset of the latent variables.

In VMP, the probabilistic model is a conjugate-exponential model. Specifically, the distribution of variables/nodes, conditioned on their parents, are drawn from the exponential family and are conjugate with respect to the distributions over these parent variables. As a result, each complete conditional is also in the exponential family (e.g., Blei et al., 2017), i.e.,

$$p(\boldsymbol{\theta}_m | \boldsymbol{\theta}_{\backslash m}, \mathcal{D}) = h(\boldsymbol{\theta}_m) \exp \left\{ \boldsymbol{\eta}_m(\boldsymbol{\theta}_{\backslash m}, \mathcal{D})^T \boldsymbol{\phi}(\boldsymbol{\theta}_m) - Z_m(\boldsymbol{\eta}_m(\boldsymbol{\theta}_{\backslash m}, \mathcal{D})) \right\}, \tag{5}$$

where $h(\boldsymbol{\theta}_m)$ is a scaling constant; $\boldsymbol{\phi}(\boldsymbol{\theta}_m)$ is the vector of sufficient statistics; $\boldsymbol{\eta}_m$ are the natural parameters; and $Z_m(\cdot)$ is the log partition function. The subscript $m$ indicates that these quantities may vary across different nodes. For simplicity, here, we consider each group $\boldsymbol{\theta}_m$ to contain a single variable.

In the conjugate-exponential model, the update for node $m$ in the mean-field VI becomes significantly simplified. By substituting (5) into (4), the update can be expressed as (e.g., Blei et al., 2017)

$$\begin{aligned} q^*(\boldsymbol{\theta}_m) &\propto \exp(\mathbb{E}_{q(\boldsymbol{\theta}_{\backslash m})}[\ln p(\boldsymbol{\theta}_m | \boldsymbol{\theta}_{\backslash m}, \mathcal{D})]) \\ &= \exp \left\{ \ln h(\boldsymbol{\theta}_m) + \mathbb{E}_{q(\boldsymbol{\theta}_{\backslash m})}[\boldsymbol{\eta}_m(\boldsymbol{\theta}_{\backslash m}, \mathcal{D})]^T \boldsymbol{\phi}(\boldsymbol{\theta}_m) - \mathbb{E}_{q(\boldsymbol{\theta}_{\backslash m})}[Z_m(\boldsymbol{\eta}_m(\boldsymbol{\theta}_{\backslash m}, \mathcal{D}))] \right\} \\ &\propto h(\boldsymbol{\theta}_m) \exp \left\{ \mathbb{E}_{q(\boldsymbol{\theta}_{\backslash m})}[\boldsymbol{\eta}_m(\boldsymbol{\theta}_{\backslash m}, \mathcal{D})]^T \boldsymbol{\phi}(\boldsymbol{\theta}_m) \right\}. \end{aligned} \tag{6}$$

This reveals that the optimal variational distribution for a node has the same functional form as the corresponding prior distribution, indicating that we only need to update the parameters of the corresponding distribution. Furthermore, the updates for each one of the nodes can be implemented locally using the expected values (messages) from the rest of the other nodes. VMP involves the exchange of messages between nodes in the network and iteratively updating the posterior distribution until the convergence is reached (Li et al., 2024). The detailed algorithm for VMP is summarized in Appendix A.

## 2.2 Expectation Propagation and Conditional Expectation Propagation

### 2.2.1 Expectation Propagation (EP)

EP is a message-passing algorithm that builds on ADF and loopy belief propagation. Similar to VI, EP also approximates the posterior distribution by minimizing the KL divergence, but in the reverse direction. EP assumes that the joint distribution of the probabilistic model can be written in a factorized form:

$$p(\boldsymbol{\theta}, \mathcal{D}) = \prod_i f_i(\boldsymbol{\theta}),$$

where each $f_i(\boldsymbol{\theta})$ is called a factor and depends only on a subset of the variables. The range of index $i$ is determined by how the variables are grouped or partitioned in the model.

This factorization can be represented using a *factor graph*, which is a bipartite graph consisting of variable nodes and factor nodes. Variable nodes correspond to individual components of $\boldsymbol{\theta}$, while factor nodes correspond to the functions $f_i$. An edge is placed between a variable node and a factor node if the variable appears in that factor. This structure clearly shows how different variables are involved in different parts of the model.

In the case of independently and identically distributed (i.i.d.) observations, each factor $f_i(\boldsymbol{\theta})$ corresponds to the likelihood term $p(\mathbf{x}_i | \boldsymbol{\theta})$ for the $i$th data point, while $f_0(\boldsymbol{\theta})$ represents the prior distribution $p(\boldsymbol{\theta})$. Then, the joint distribution is written as

$$p(\boldsymbol{\theta}, \mathcal{D}) = p(\boldsymbol{\theta}) \prod_{i=1}^N p(\mathbf{x}_i | \boldsymbol{\theta}) = \prod_{i=0}^N f_i(\boldsymbol{\theta}). \tag{7}$$

Figure 2(b) illustrates a typical factor graph used in EP, where messages are passed between variable nodes and factor nodes based on the graph structure.

We are interested in evaluating the posterior distribution

$$p(\boldsymbol{\theta}|\mathcal{D}) = \frac{p(\boldsymbol{\theta}, \mathcal{D})}{p(\mathcal{D})} = \frac{1}{Z}\prod_i f_i(\boldsymbol{\theta}),$$

where $Z = p(\mathcal{D})$ is the normalization constant, which can be calculated as

$$Z = \int \prod_i f_i(\boldsymbol{\theta})d\boldsymbol{\theta}.$$

EP approximates the posterior by a product of factors , given by (Minka, 2013)

$$q(\boldsymbol{\theta}) = \frac{1}{\tilde{Z}}\prod_i \tilde{f}_i(\boldsymbol{\theta}),$$

where $\tilde{f}_i$ is an approximation of $f_i$ that belongs to the exponential family, and $\tilde{Z}$ is the associated normalization constant. Ideally, the determination of the involved factors $\{\tilde{f}_i(\boldsymbol{\theta})\}_{i=1}^N$ involves the minimization of the KL divergence from $p(\boldsymbol{\theta}|\mathcal{D})$ to $q(\boldsymbol{\theta})$, given by

$$\mathrm{KL}(p\|q) = \mathrm{KL}\left(\frac{1}{Z}\prod_i f_i(\boldsymbol{\theta})\|\frac{1}{\tilde{Z}}\prod_i \tilde{f}_i(\boldsymbol{\theta})\right).$$

However, this minimization is typically intractable due to the need to compute expectations with respect to the true distribution.

EP approximates the posterior by iteratively refining each factor while accounting for the influence of the others. At each step, it selects one factor, temporarily removes it, and updates it based on the remaining information. This process is repeated across all factors until convergence. To elaborate, EP follows four simple steps in each iteration (Minka, 2013). First, select a factor $\tilde{f}_i$ for updating and remove it from the approximation $q(\boldsymbol{\theta})$ to produce the *calibrating* distribution $q^{\backslash i}(\boldsymbol{\theta})$, defined as $q^{\backslash i}(\boldsymbol{\theta}) = q(\boldsymbol{\theta})/\tilde{f}_i(\boldsymbol{\theta})$. Note that $q^{\backslash i}$ can also be derived from the product of factors $i \neq j$, but in practice, the division is more convenient. Second, the calibrating distribution is combined with the factor $f_i(\boldsymbol{\theta})$ to obtain the *tilted* distribution

$$\hat{p}_i(\boldsymbol{\theta}) = \frac{1}{Z_i}f_i(\boldsymbol{\theta})q^{\backslash i}(\boldsymbol{\theta}), \tag{8}$$

where $Z_i$ is the associated normalization constant. Third, we obtain an approximation $q^{\natural}(\boldsymbol{\theta})$ of $\hat{p}_i(\boldsymbol{\theta})$ by minimizing the KL divergence between $\hat{p}_i(\boldsymbol{\theta})$ and $q^{\natural}(\boldsymbol{\theta})$. If $q^{\natural}(\boldsymbol{\theta})$ belongs to the exponential family, as it is often the case, the minimum can be obtained by moment matching (Maybeck, 1982), i.e.,

$$\mathbb{E}_{q^{\natural}(\boldsymbol{\theta})}[\boldsymbol{\phi}(\boldsymbol{\theta})] = \mathbb{E}_{\hat{p}_i(\boldsymbol{\theta})}[\boldsymbol{\phi}(\boldsymbol{\theta})], \tag{9}$$

where $\boldsymbol{\phi}(\boldsymbol{\theta})$ is the sufficient statistics of $q^{\natural}(\boldsymbol{\theta})$. A more detailed discussion of moment matching is provided in Appendix B. Note that the natural parameter of $q^{\natural}(\boldsymbol{\theta})$ is implicitly specified in the moment matching process. For example, if $q^{\natural}(\boldsymbol{\theta})$ is a Gaussian distribution $\mathcal{N}(\boldsymbol{\theta}|\boldsymbol{\mu}, \boldsymbol{\Sigma})$ then we can minimize the KL divergence by setting the mean $\boldsymbol{\mu}$ equal to the mean of $\hat{p}_i(\boldsymbol{\theta})$ and the covariance $\boldsymbol{\Sigma}$ equal to the covariance of $\hat{p}_i(\boldsymbol{\theta})$. Finally, the factor $\tilde{f}_i$ is update via $\tilde{f}_i(\boldsymbol{\theta}) \propto q^{\natural}(\boldsymbol{\theta})/q^{\backslash i}(\boldsymbol{\theta})$.

The rationale behind this update is to ensure that the approximate factor contributes to the posterior in a manner similar to the corresponding data likelihood. Due to the local refinement[2], the factors can be efficiently calculated in a distributed manner. However, convergence is not guaranteed in general.

---

[2]By local refinement, we mean that each factor is updated individually based on its local context in the model, without requiring access to the entire joint distribution.

### 2.2.2 Conditional Expectation Propagation (CEP)

While EP is known for its favorable accuracy and speed on diverse tasks, a significant challenge in its application arises from the computational intractability of the expectations $\mathbb{E}_{\hat{p}_i(\boldsymbol{\theta})}[\boldsymbol{\phi}(\boldsymbol{\theta})]$ in (9) for models with complex data likelihood. To overcome this limitation, several methods have been proposed. One such method is the CEP, which offers efficient and analytical updates.

In CEP, the approximate factor is assumed to be further factorized with respect to the variable groups $\{\boldsymbol{\theta}_1, \cdots, \boldsymbol{\theta}_M\}$, which can be expressed as

$$\tilde{f}_i(\boldsymbol{\theta}) = \prod_m \tilde{f}_i(\boldsymbol{\theta}_m), \tag{10}$$

where $\{\tilde{f}_i(\boldsymbol{\theta}_m)\}_{m=1}^M$ are constrained to be in the exponential family. As a result, the approximate posterior $q(\boldsymbol{\theta}_m)$ and the calibrating distribution $q^{\backslash i}(\boldsymbol{\theta})$ are both factorized over the variable groups. It is worth noting that the factorized message factors are also widely used in EP algorithms for large-scale applications.

Given the factorized form in (10), the objective is to update each subfactor $\tilde{f}_i(\boldsymbol{\theta}_m)$. By utilizing the law of iterated expectations, the moment $\mathbb{E}_{\hat{p}_i(\boldsymbol{\theta})}[\boldsymbol{\phi}(\boldsymbol{\theta}_m)]$ required for updating $\tilde{f}_i(\boldsymbol{\theta}_m)$ can be expressed as (Wang & Zhe, 2020)

$$\mathbb{E}_{\hat{p}_i(\boldsymbol{\theta})}[\boldsymbol{\phi}(\boldsymbol{\theta}_m)] = \mathbb{E}_{\hat{p}_i(\boldsymbol{\theta}_{\backslash m})}\left[\mathbb{E}_{\hat{p}_i(\boldsymbol{\theta}_m|\boldsymbol{\theta}_{\backslash m})}[\boldsymbol{\phi}(\boldsymbol{\theta}_m)]\right], \tag{11}$$

where $\hat{p}_i(\boldsymbol{\theta}_m|\boldsymbol{\theta}_{\backslash m})$ is the conditional distribution and $\hat{p}_i(\boldsymbol{\theta}_{\backslash m})$ is the marginal distribution. The conditional moment $\mathbb{E}_{\hat{p}_i(\boldsymbol{\theta}_m|\boldsymbol{\theta}_{\backslash m})}[\boldsymbol{\phi}(\boldsymbol{\theta}_m)]$ often has an analytical form since the rest of the variables except $\boldsymbol{\theta}_m$ are fixed. More generally, the conditional moment can be represented with a quadrature formula. (Wang & Zhe, 2020)

To compute the moment in (11), EP requires the computation of the expectation of the conditional moment with respect to the marginal posterior $\hat{p}_i(\boldsymbol{\theta}_{\backslash m})$. However, this computation is also intractable for models with complex likelihoods. To address this challenge, CEP assumes that $q(\boldsymbol{\theta}_{\backslash m})$ and $\hat{p}_i(\boldsymbol{\theta}_{\backslash m})$ are close in high-density regions as their moments are matched (Wang & Zhe, 2020). In the sequel, CEP employs $q(\boldsymbol{\theta}_{\backslash m})$ as a surrogate for $\hat{p}_i(\boldsymbol{\theta}_{\backslash m})$ in the respective computation. The goal now becomes to calculate the expectation $\mathbb{E}_{q(\boldsymbol{\theta}_{\backslash m})}\left[\mathbb{E}_{\hat{p}_i(\boldsymbol{\theta}_m|\boldsymbol{\theta}_{\backslash m})}[\boldsymbol{\phi}(\boldsymbol{\theta}_m)]\right]$.

Note that the conditional moment $\mathbb{E}_{\hat{p}_i(\boldsymbol{\theta}_m|\boldsymbol{\theta}_{\backslash m})}[\boldsymbol{\phi}(\boldsymbol{\theta}_m)]$ is a function of the sufficient statistics of $\boldsymbol{\theta}_{\backslash m}$, denoted as $\mathbb{E}_{\hat{p}_i(\boldsymbol{\theta}_m|\boldsymbol{\theta}_{\backslash m})}[\boldsymbol{\phi}(\boldsymbol{\theta}_m)] = h(\boldsymbol{\phi}(\boldsymbol{\theta}_{\backslash m}))$, where $\boldsymbol{\phi}(\boldsymbol{\theta}_{\backslash m}) = \{\boldsymbol{\phi}(\boldsymbol{\theta}_1), \cdots, \boldsymbol{\phi}(\boldsymbol{\theta}_{m-1}), \boldsymbol{\phi}(\boldsymbol{\theta}_{m+1}), \cdots, \boldsymbol{\phi}(\boldsymbol{\theta}_M)\}$ is the set of sufficient statistics. If the expectation $\mathbb{E}_{q(\boldsymbol{\theta}_{\backslash m})}[h(\boldsymbol{\phi}(\boldsymbol{\theta}_m))]$ remains intractable, it can be approximated using the *delta approximation method* as $h(\mathbb{E}_{q(\boldsymbol{\theta}_{\backslash m})}[\boldsymbol{\phi}(\boldsymbol{\theta}_m)])$. The delta approximation method can be interpreted as a special case of the Laplace approximation or, equivalently, as a first-order Taylor expansion around the mean, as discussed in Appendix C. Similar to expectation propagation (EP), the messages can be computed in a distributed fashion. However, the convergence of the overall procedure remains an open problem. The full algorithm for CEP is provided in Appendix A.

### 2.3 Assumed Density Filtering

ADF is an online Bayesian inference method that can be seen as a special case of EP. It provides an efficient approach for approximating posterior distributions in a sequential manner. ADF is obtained by initializing all the approximating factors, except the first one, to unity and then updating each factor once in a single pass. The ADF algorithm shares similarities with EP but it simplifies certain aspects of the process. Particularly, in ADF, the removal step, which involves creating a calibrating distribution by removing a factor from the approximation, is ignored. Instead, the calibrating distribution is replaced by the full approximation, given by (Li et al., 2015)

$$q^{\backslash i}(\boldsymbol{\theta}) = q(\boldsymbol{\theta}).$$

Consequently, the tilted distribution in ADF can be expressed as:

$$\hat{p}_i(\boldsymbol{\theta}) \propto f_i(\boldsymbol{\theta})q(\boldsymbol{\theta}).$$

The subsequent steps in ADF are the same as in EP.

## 2.4 Connections and Differences

As discussed above, ABI methods are commonly categorized into two groups: VI-based methods (e.g., VI, VMP) and EP-based methods (e.g., EP, ADF, CEP). Both aim to approximate the posterior by minimizing a divergence between the true and approximate distributions, but they differ in the choice of divergence and resulting algorithmic behavior.

VI-based methods minimize $\text{KL}(q\|p)$, where $q$ is the variational approximation and $p$ is the true posterior. This leads to algorithms that monotonically improve the ELBO and converge to a local optimum. However, such minimization typically favors compact approximations, often capturing only one mode of a multimodal posterior (Bishop, 2006).

In contrast, EP-based methods minimize $\text{KL}(p\|q)$ in a local, factor-wise manner. This tends to yield approximations that better reflect the full support of the posterior (Bishop, 2006). EP relies on local message passing over factor graphs and can be extended to distributed or online versions. However, its convergence is not theoretically guaranteed, and moment computations involved in updates may be intractable in practice.

So far, most existing studies have focused on the connections within each family—such as between VI and VMP, or between EP and its variants like ADF and CEP—while the relationships across the two families have remained largely unexplored. Fig. 1 provides an overview of the connections among these methods: the top and bottom parts represent VI- and EP-based methods, respectively. Black arrows indicate known connections, and red arrows highlight new links identified in this work, as discussed in Section 1.

# 3 Main Results

The previous section reviewed several representative approximate Bayesian inference methods, emphasizing the roles of the mean-field assumption and moment matching. To support the theoretical development, Appendix B presents two key lemmas that characterize the behavior of KL divergence minimization under different model assumptions. These lemmas provide the foundation for the analysis in this section.

We now present the main theoretical results of the paper. The first subsection introduces our central theorem, which establishes a connection between VMP and CEP under mild conditions. The second subsection extends this result and explores its implications for other inference algorithms. The third subsection offers a probabilistic interpretation of the theorem, providing complementary insights to aid understanding. The final subsection summarizes the results and discusses practical considerations for implementation.

## 3.1 Connection between VMP and CEP

To establish the connection between CEP and VMP, we initially present the following lemma.

**Lemma 3:** *Assume $p(\boldsymbol{\theta})$ is a fixed distribution and $q(\boldsymbol{\theta})$ factorizes with respect to variable groups, i.e.,*

$$q(\boldsymbol{\theta}) = \prod_m q(\boldsymbol{\theta}_m),$$

*where each factor $q(\boldsymbol{\theta}_m)$ belongs to the exponential family. Then minimizing the divergence $KL(p\|q)$ with respect to $q$ gives*

$$\mathbb{E}_{q(\boldsymbol{\theta})}[\phi(\boldsymbol{\theta}_m)] = \mathbb{E}_{p(\boldsymbol{\theta})}[\phi(\boldsymbol{\theta}_m)], \forall m, \tag{12}$$

*where $\phi(\boldsymbol{\theta}_m)$ is the sufficient statistics of $q(\boldsymbol{\theta}_m)$.*

*Proof:* See Appendix D.

Lemma 3 can be seen as a combination of Lemma 1 and Lemma 2, establishing a connection between the conditional moment matching and the minimization of KL divergence. In CEP, the optimal factor is given

by

$$\tilde{f}_i(\boldsymbol{\theta}_m) \propto q^{\natural}(\boldsymbol{\theta}_m)/q^{\backslash i}(\boldsymbol{\theta}_m),$$

where the tilted distribution $q^{\backslash i}(\boldsymbol{\theta}_m)$ is defined in equation 8 and the variational distribution $q^{\natural}(\boldsymbol{\theta})$ is obtained through moment matching, satisfying

$$\mathbb{E}_{q^{\natural}(\boldsymbol{\theta})}[\phi(\boldsymbol{\theta}_m)] = \mathbb{E}_{\hat{p}_i(\boldsymbol{\theta})}[\phi(\boldsymbol{\theta}_m)]. \tag{13}$$

To establish the connection between CEP and VMP, it is necessary to derive an analytical form for the factor $\tilde{f}_i(\boldsymbol{\theta}_m)$. Generally, $\tilde{f}_i(\boldsymbol{\theta}_m)$ does not possess an analytical form due to the involvement of moment matching in the computation of $q^{\natural}(\boldsymbol{\theta})$. However, by comparing (12) and (13), we can show that for conjugate-exponential models under mild conditions, $\tilde{f}_i(\boldsymbol{\theta}_m)$ does indeed have an analytical form.

**Lemma 4:** *Consider a conjugate-exponential probabilistic model represented as a Bayesian network. If the expectations are calculated using the delta approximation method, the optimal factor in CEP is expressed as*

$$\tilde{f}_i(\boldsymbol{\theta}_m) \propto \frac{\hat{p}_i(\boldsymbol{\theta}_m | \mathbb{E}_q[\phi(\boldsymbol{\theta}_{\backslash m})])}{q^{\backslash i}(\boldsymbol{\theta}_m)}. \tag{14}$$

*Proof:* See Appendix D.

Based on the analytical form of $\tilde{f}_i(\boldsymbol{\theta}_m)$, we can show the connection between the CEP and VMP, and state the following theorem.

**Theorem 1:** *Consider a conjugate-exponential probabilistic model represented as a Bayesian network. Suppose the variational distribution follows the mean-field assumption and the observations are i.i.d. Then the CEP and VMP yield the same update equations under the following conditions:*

- *The update in CEP is performed on the variable groups.*

- *The expectations in CEP are calculated using the delta approximation method.*

*Proof:* To prove Theorem 1, we first give the following lemma.

**Lemma 5:** *Consider a conjugate-exponential probabilistic model represented as a Bayesian network. Suppose the variational distribution follows the mean-field assumption and the observations are i.i.d. If the update in CEP is performed on the variable groups, then a sufficient condition for the equivalence between the update equations of CEP and VMP is*

$$\ln \tilde{f}_i(\boldsymbol{\theta}_m) = \mathbb{E}_{q(\boldsymbol{\theta}_{\backslash m})}[\ln f_i(\boldsymbol{\theta})]. \tag{15}$$

*Proof:* To prove this lemma, we start by considering the logarithm of the optimal distribution in VMP, given by:

$$\begin{aligned}
\ln q^*(\boldsymbol{\theta}_m) &= \mathbb{E}_{q(\boldsymbol{\theta}_{\backslash m})}[\ln p(\boldsymbol{\theta}, \mathcal{D})] + \text{const4} \\
&= \mathbb{E}_{q(\boldsymbol{\theta}_{\backslash m})}[\sum_i \ln f_i(\boldsymbol{\theta})] + \text{const4} \\
&= \sum_i \mathbb{E}_{q(\boldsymbol{\theta}_{\backslash m})}[\ln f_i(\boldsymbol{\theta})] + \text{const4}.
\end{aligned} \tag{16}$$

where $p(\boldsymbol{\theta}, \mathcal{D}) = \prod_i f_i(\boldsymbol{\theta})$ follows from (7) under the i.i.d. assumption. Note that we also use $f_i(\boldsymbol{\theta})$ to represent the likelihood and prior, as in CEP. On the other hand, if the update in CEP is performed on the variable groups, then the optimal distribution for each variable group can be expressed as the product of the approximate factors:[3]

$$q^*(\boldsymbol{\theta}_m) \propto \prod_i \tilde{f}_i(\boldsymbol{\theta}_m).$$

---

[3]Here, we also use the notation $q^*(\boldsymbol{\theta}_m)$ to denote the optimal variational distribution in CEP.

Taking the logarithm of both sides gives:

$$\ln q^*(\boldsymbol{\theta}_m) = \sum_i \ln \tilde{f}_i(\boldsymbol{\theta}_m) + \text{constant5}. \tag{17}$$

By comparing (16) and (17), it can be seen that the updates of CEP and VMP are the same if (15) holds. ∎

Then we show that (15) in Lemma 5 holds if the expectations are approximated using the delta approximation method. From Lemma 4, the logarithm of message factor $\tilde{f}_i(\boldsymbol{\theta}_m)$ in CEP can be represented as

$$\ln \tilde{f}_i(\boldsymbol{\theta}_m) = \ln \hat{p}_i(\boldsymbol{\theta}_m | \mathbb{E}_q[\boldsymbol{\phi}(\boldsymbol{\theta}_{\backslash m})]) - \ln q^{\backslash i}(\boldsymbol{\theta}_m). \tag{18}$$

From (16), it can be seen that the optimal variational distribution in VMP consists of some independent terms, which can be expressed as

$$\begin{aligned}
\mathbb{E}_{q(\boldsymbol{\theta}_{\backslash m})}[\ln f_i(\boldsymbol{\theta})] &= \mathbb{E}_{q(\boldsymbol{\theta}_{\backslash m})}[\ln f_i(\boldsymbol{\theta})] + \ln q^{\backslash i}(\boldsymbol{\theta}_m) - \ln q^{\backslash i}(\boldsymbol{\theta}_m) \\
&= \mathbb{E}_{q(\boldsymbol{\theta}_{\backslash m})}[\ln f_i(\boldsymbol{\theta}) q^{\backslash i}(\boldsymbol{\theta}_m)] - \ln q^{\backslash i}(\boldsymbol{\theta}_m) \\
&= \mathbb{E}_{q(\boldsymbol{\theta}_{\backslash m})}[\ln \hat{p}_i(\boldsymbol{\theta}_m | \boldsymbol{\theta}_{\backslash m})] - \ln q^{\backslash i}(\boldsymbol{\theta}_m).
\end{aligned} \tag{19}$$

Comparing (18) and (19), we see that the equation (1) holds if and only if

$$\mathbb{E}_{q(\boldsymbol{\theta}_{\backslash m})}[\ln \hat{p}_i(\boldsymbol{\theta}_m | \boldsymbol{\theta}_{\backslash m})] = \ln \hat{p}_i(\boldsymbol{\theta}_m | \mathbb{E}_q[\boldsymbol{\phi}(\boldsymbol{\theta}_{\backslash m})]).$$

Now, we show that this equality holds under the conditions in Theorem 1. Since the model is conjugate-exponential, the conditional distribution $\hat{p}_i(\boldsymbol{\theta}_m | \boldsymbol{\theta}_{\backslash m})$ is in the exponential family and can be expressed as:

$$\hat{p}_i(\boldsymbol{\theta}_m | \boldsymbol{\theta}_{\backslash m}) = h(\boldsymbol{\theta}_m) \exp\left\{ \boldsymbol{\eta}_m(\boldsymbol{\theta}_{\backslash m})^T \boldsymbol{\phi}(\boldsymbol{\theta}_m) - Z_m(\boldsymbol{\eta}_m(\boldsymbol{\theta}_{\backslash m})) \right\}.$$

where $\boldsymbol{\phi}(\boldsymbol{\theta}_m)$ is the vector of sufficient statistics; $\boldsymbol{\eta}_m$ are the natural parameters; and $Z_m(\cdot)$ is the log partition function. Its logarithm can be expressed by

$$\ln \hat{p}_i(\boldsymbol{\theta}_m | \boldsymbol{\theta}_{\backslash m}) = \ln h(\boldsymbol{\theta}_m) + \boldsymbol{\eta}_m(\boldsymbol{\theta}_{\backslash m})^T \boldsymbol{\phi}(\boldsymbol{\theta}_m) - Z_m(\boldsymbol{\eta}_m(\boldsymbol{\theta}_{\backslash m})).$$

Taking expectation with respect to $q(\boldsymbol{\theta}_{\backslash m})$ yields

$$\mathbb{E}_{q(\boldsymbol{\theta}_{\backslash m})}[\ln \hat{p}_i(\boldsymbol{\theta}_m | \boldsymbol{\theta}_{\backslash m})] = \ln h(\boldsymbol{\theta}_m) + \mathbb{E}_{q(\boldsymbol{\theta}_{\backslash m})}[\boldsymbol{\eta}_m(\boldsymbol{\theta}_{\backslash m})]^T \boldsymbol{\phi}(\boldsymbol{\theta}_m) + \text{const6}.$$

For a conjugate-exponential model, $\ln \hat{p}_i(\boldsymbol{\theta}_m | \boldsymbol{\theta}_{\backslash m})$ is a *multi-linear* function of the sufficient statistics of $\boldsymbol{\theta}_m$ and the variables in $\boldsymbol{\theta}_{\backslash m}$ (i.e., $\phi(\boldsymbol{\theta}_j), \forall j = 1, \cdots, M$). As a result, $\boldsymbol{\eta}_m(\boldsymbol{\theta}_{\backslash m})$ is a multi-linear function of $\phi(\boldsymbol{\theta}_j), \forall j = 1, \cdots, M$, and we have

$$\begin{aligned}
\mathbb{E}_{q(\boldsymbol{\theta}_{\backslash m})}[\ln \hat{p}_i(\boldsymbol{\theta}_m | \boldsymbol{\theta}_{\backslash m})] &= \ln h(\boldsymbol{\theta}_m) + \mathbb{E}_{q(\boldsymbol{\theta}_{\backslash m})}[\boldsymbol{\eta}_m(\boldsymbol{\phi}(\boldsymbol{\theta}_{\backslash m}))]^T \boldsymbol{\phi}(\boldsymbol{\theta}_m) + \text{const6} \\
&= \ln h(\boldsymbol{\theta}_m) + \boldsymbol{\eta}_m(\mathbb{E}_q[\boldsymbol{\phi}(\boldsymbol{\theta}_{\backslash m})])^T \boldsymbol{\phi}(\boldsymbol{\theta}_m) + \text{const6} \\
&= \ln \hat{p}_i(\boldsymbol{\theta}_m | \mathbb{E}_q[\boldsymbol{\phi}(\boldsymbol{\theta}_{\backslash m})]).
\end{aligned} \tag{20}$$

Thus, equation (15) holds, and by Lemma 5, we conclude that the updates of CEP and VMP are equivalent under the specified conditions. ∎

In practical applications, these preconditions and conditions are often satisfied, enabling the derivation of analytical updates, as demonstrated in the example provided in Section 4. Below, we delve deeper into these conditions and discuss their respective implications.

The preconditions in Theorem 1 establish a foundation for the effective application of both VMP and CEP, as outlined in the preliminaries. Specifically, the i.i.d. assumption enables the update in VMP to be expressed

as a summation of $N$ terms, each corresponding to a factor in the CEP framework. This highlights that the VMP update can be interpreted as the merging of messages sent from the data nodes, aligning with the message-passing nature of VMP, which will be discussed further in the next subsection. Consequently, it becomes easy to derive a streaming version of VMP, which is described in detail in the next section. Additionally, the conjugate-exponential condition ensures that the updates of the factors in CEP can be formulated analytically, avoiding the need for moment matching. When this assumption does not hold, the direct connection between VMP and CEP may break down, as the factor updates in CEP may no longer have analytical solutions.

Regarding the specific conditions, the first ensures that the update of $q(\boldsymbol{\theta}_m)$ in CEP is expressed as the product of a number of factors, enabling efficient parallel or distributed computation and significantly reducing computational costs. The second condition simplifies the expectation calculations in CEP, thereby enhancing the tractability of the inference process. For models of practical interest, the outer expectation in CEP is typically intractable, making this approximation both necessary and effective (Pan et al., 2020; Fang et al., 2021a). Note that equation (20) can be seen as a reprarmeterization process, where the statistic parameters of $\boldsymbol{\theta}_{\backslash m}$ are replcaed with their expectations regarding the corresponding distribution. This procedure is discussed in the original VMP paper, along with a simple illustrative example (Winn et al., 2005).

It is also worth noting that the connection between CEP and VMP can be viewed from a more general perspective. To see this, note that a fundamental assumption in CEP is that the message factor $\tilde{f}_i$ factorizes with respect to variable groups, allowing the approximate posterior to be expressed as

$$q(\boldsymbol{\theta}) \propto \prod_i \tilde{f}_i(\boldsymbol{\theta}) = \prod_i \prod_m \tilde{f}_{im}(\boldsymbol{\theta}_m) = \prod_m q(\boldsymbol{\theta}_m),$$

which in fact corresponds to the mean-field assumption. From Lemma 2, we know that the optimal solution of $\min_{q(\boldsymbol{\theta}_m)} \mathrm{KL}(p(\boldsymbol{\theta}|\mathcal{D})\|q(\boldsymbol{\theta}))$ is given by $q^*(\boldsymbol{\theta}_m) = p(\boldsymbol{\theta}_m|\mathcal{D})$, which can be further written as

$$\begin{aligned}
q^*(\boldsymbol{\theta}_m) &= p(\boldsymbol{\theta}_m|\mathcal{D}) \\
&= \int p(\boldsymbol{\theta}_m, \boldsymbol{\theta}_{\backslash m}|\mathcal{D}) d\boldsymbol{\theta}_{\backslash m} \\
&= \int p(\boldsymbol{\theta}_m|\boldsymbol{\theta}_{\backslash m}, \mathcal{D}) p(\boldsymbol{\theta}_{\backslash m}|\mathcal{D}) d\boldsymbol{\theta}_{\backslash m} \\
&= \mathbb{E}_p[p(\boldsymbol{\theta}_m|\boldsymbol{\theta}_{\backslash m}, \mathcal{D})].
\end{aligned}$$

By applying the approximations in CEP, i.e., using $q(\boldsymbol{\theta}_m)$ as a surrogate of $p(\boldsymbol{\theta}_m)$ in moment computation and the delta approximation, the optimal variational distribution can be approximated as

$$q^*(\boldsymbol{\theta}_m) = \mathbb{E}_p[p(\boldsymbol{\theta}_m|\boldsymbol{\theta}_{\backslash m}, \mathcal{D})] \approx p(\boldsymbol{\theta}_m|\mathbb{E}_q[\boldsymbol{\phi}(\boldsymbol{\theta}_{\backslash m})], \mathcal{D}). \tag{21}$$

In VMP, each optimal variational distribution becomes

$$\begin{aligned}
\ln q^*(\boldsymbol{\theta}_m) &= \mathbb{E}_{q(\boldsymbol{\theta}_{\backslash m})}[\ln p(\boldsymbol{\theta}, \mathcal{D})] + \text{constant2} \\
&= \mathbb{E}_{q(\boldsymbol{\theta}_{\backslash m})}[\ln p(\boldsymbol{\theta}_m|\boldsymbol{\theta}_{\backslash m}, \mathcal{D})] + \text{constant3} \\
&= \ln p(\boldsymbol{\theta}_m|\mathbb{E}_q[\boldsymbol{\phi}(\boldsymbol{\theta}_{\backslash m})], \mathcal{D}),
\end{aligned}$$

which leads to

$$q^*(\boldsymbol{\theta}_m) = p(\boldsymbol{\theta}_m|\mathbb{E}_q[\boldsymbol{\phi}(\boldsymbol{\theta}_{\backslash m})], \mathcal{D}). \tag{22}$$

By comparing (21) and (22), it can be seen that the inherent objective of both VMP and CEP is to approximate the conditional marginal distribution $p(\boldsymbol{\theta}_m|\mathbb{E}_q[\boldsymbol{\phi}(\boldsymbol{\theta}_{\backslash m})], \mathcal{D})$. VMP and CEP start from different KL formulations. However, in both cases, the theoretical optimal is the same due to the properties of the KL. This justifies the derived connection.

### 3.2 Extensions and Implications

The previous section established a strong connection between CEP and VMP. Expanding on this connection, we present new theoretical results regarding the convergence and scalability of several ABI methods.

#### 3.2.1 Convergence of CEP

As previously mentioned, the convergence of EP is not generally guaranteed. To address this issue, some approaches apply energy optimization techniques directly to the associated objective function rather than relying on local updates. For instance, they implement EP based on the convergent double-loop optimization algorithm (Opper et al., 2005; Hasenclever et al., 2017). However, these approaches require additional designs and exhibit increased computational complexities.

Since CEP is developed from EP, its convergence properties also remain an open question. Nevertheless, by leveraging the established connection with VMP, we can demonstrate that CEP is guaranteed to converge under certain mild conditions. Specifically, we present the following corollary.

**Corollary 1:** *Consider a conjugate-exponential probabilistic model represented as a Bayesian network. Suppose the variational distribution follows the mean-field assumption and the observations are i.i.d. If the conditions in Theorem 1 hold, then CEP updates are guaranteed to converge to a local minimum of the KL divergence.*

*Proof:* See Appendix D.

From (16), the optimal variational distribution $q^*(\boldsymbol{\theta}_m)$ in VMP is

$$\ln q^*(\boldsymbol{\theta}_m) = \sum_i \mathbb{E}_{q(\boldsymbol{\theta}_{\backslash m})}[\ln f_i(\boldsymbol{\theta})] + \text{const4}.$$

As shown in Section 2, this update increases the ELBO at each iteration, ensuring the convergence property of VMP. According to Lemma 5, we have $\mathbb{E}_{q(\boldsymbol{\theta}_{\backslash m})}[\ln f_i(\boldsymbol{\theta})] = \ln \tilde{f}_i(\boldsymbol{\theta}_m)$. Since $\tilde{f}_i(\boldsymbol{\theta}_m)$ is the message factor, the term $\mathbb{E}_{q(\boldsymbol{\theta}_{\backslash m})}[\ln f_i(\boldsymbol{\theta})]$ can be interpreted as the message sent from the $i$th data node. Thus, the update in VMP can be viewed as merging all the messages sent by data nodes. In other words, in CEP, merging the message factors $\tilde{f}_i(\boldsymbol{\theta}_m)$ sent by data nodes increases the ELBO, thereby ensuring convergence.

It is important to note that in the standard implementations of CEP, updates are performed on the factors instead of the variable groups. In other words, $\tilde{f}_i$ is updated sequentially in each iteration. This factor-based update mechanism allows for a more fine-grained local optimization, which might be the reason for its superior performance in various tasks. However, this type of local optimization does not guarantee convergence in general. If the updates in CEP are performed on the factors rather than on the variable groups, the convergence guarantee is lost.

To see this, note that if the updates in CEP are performed on the factors, the increase of ELBO is not guaranteed. Specifically, the ELBO can be expressed as

$$\begin{aligned}
\mathcal{L} &= \int \prod_m q(\boldsymbol{\theta}_m) \left\{ \ln p(\boldsymbol{\theta}, \mathcal{D}) - \sum_m \ln q(\boldsymbol{\theta}_m) \right\} d\boldsymbol{\theta} \\
&= \int q(\boldsymbol{\theta}_m) \mathbb{E}_{q(\boldsymbol{\theta}_{\backslash m})}[\ln p(\boldsymbol{\theta}, \mathcal{D})] d\boldsymbol{\theta}_m - \int q(\boldsymbol{\theta}_m) \ln q(\boldsymbol{\theta}_m) d\boldsymbol{\theta}_m \\
&= \int \prod_i \tilde{f}_i(\boldsymbol{\theta}_m) \sum_i \mathbb{E}_{q(\boldsymbol{\theta}_{\backslash m})}[f_i(\boldsymbol{\theta})] d\boldsymbol{\theta}_m + \int \prod_i \tilde{f}_i(\boldsymbol{\theta}_m) \sum_i \ln \tilde{f}_i(\boldsymbol{\theta}_m) d\boldsymbol{\theta}_m.
\end{aligned}$$

The optimal factor in CEP can be written as

$$\begin{aligned}
\ln \tilde{f}_i(\boldsymbol{\theta}_m) &= \ln \hat{p}_i(\boldsymbol{\theta}_m | \mathbb{E}_q[\boldsymbol{\phi}(\boldsymbol{\theta}_{\backslash m})]) - \ln q^{\backslash i}(\boldsymbol{\theta}_m) \\
&= \ln q^{\backslash i}(\boldsymbol{\theta}_m) f_i(\boldsymbol{\theta}_m | \mathbb{E}_q[\boldsymbol{\phi}(\boldsymbol{\theta}_{\backslash m})]) - \ln q^{\backslash i}(\boldsymbol{\theta}_m) \\
&= \ln f_i(\boldsymbol{\theta}_m | \mathbb{E}_q[\boldsymbol{\phi}(\boldsymbol{\theta}_{\backslash m})]),
\end{aligned}$$

which leads to $\tilde{f}_i(\boldsymbol{\theta}_m) = f_i(\boldsymbol{\theta}_m|\mathbb{E}_q[\boldsymbol{\phi}(\boldsymbol{\theta}_{\backslash m})])$. Due to the multiplication and integration involved in ELBO, optimizing $\mathcal{L}$ with respect to $\tilde{f}_i(\boldsymbol{\theta}_m)$ does not yield the same results as in CEP. Therefore, each update does not necessarily increase the ELBO, and CEP may not converge in this scenario. Similarly, this local optimization of message factors is also the reason why standard EP may not converge.

### 3.2.2 Connections to Streaming Bayes

The concept of EP is developed from ADF, an online Bayesian algorithm designed for streaming data. As CEP is a variant of EP, it can be readily adapted into a streaming version. Furthermore, due to the strong connections between CEP and VMP updates, it is straightforward to construct a streaming version of VMP. The resulting method shares a close connection with streaming variational Bayes, although it is developed from a distinct perspective and offers different interpretations.

In Section II, we observe that ADF differs from EP in the factor removing step. In ADF, the removing step is ignored, and the calibrating distribution is replaced by the full approximation obtained from the previous iteration. The updated approximating posterior is computed by directly multiplying the previous approximation with the newly updated message factor associated with the added data.

Mathematically, assuming the current approximation is denoted as $q(\boldsymbol{\theta})$, the new posterior in ADF is

$$q^*(\boldsymbol{\theta}) = \min_{\hat{q}(\boldsymbol{\theta})} \mathrm{KL}(\hat{p}_i(\boldsymbol{\theta}) \| \hat{q}(\boldsymbol{\theta})),$$

where $\hat{p}_i(\boldsymbol{\theta}) \propto f_i(\boldsymbol{\theta})q(\boldsymbol{\theta})$. The resulting $q^*(\boldsymbol{\theta})$ is then used as the current approximation in the next iteration. In a conjugate-exponential model with mutually independent variable groups, the update of the posterior for each variable has a closed-form solution, given by

$$q^*(\boldsymbol{\theta}_m) = \hat{p}_i(\boldsymbol{\theta}_m) = \mathbb{E}_{\hat{p}_i(\boldsymbol{\theta}_{\backslash m})}[\hat{p}_i(\boldsymbol{\theta}_m|\boldsymbol{\theta}_{\backslash m})].$$

Upon the arrival of new data, we can optimize each variable group and multiply their distributions together to obtain the new approximation $q^*(\boldsymbol{\theta})$.

As mentioned in the previous subsection, the variable update in VMP merges all the messages sent from the other nodes simultaneously. If the data arrives in a streaming manner, we can sequentially merge the messages to update the variables. Building upon this insight, we can easily modify the VMP to a streaming version. Specifically, when a new sample $\mathbf{x}_i$ arrives, the updated estimate of the posterior in VMP is

$$\begin{aligned} \ln q^*(\boldsymbol{\theta}_m) &= \mathbb{E}_{q(\boldsymbol{\theta}_{\backslash m})}[\ln p(\boldsymbol{\theta}, \mathcal{D})] + \mathrm{const2} \qquad\qquad (23)\\ &= \mathbb{E}_{q(\boldsymbol{\theta}_{\backslash m})}[\ln f_i(\boldsymbol{\theta})q(\boldsymbol{\theta})] + \mathrm{const3}\\ &= \mathbb{E}_{q(\boldsymbol{\theta}_{\backslash m})}[\ln \hat{p}_i(\boldsymbol{\theta})]\\ &= \mathbb{E}_{q(\boldsymbol{\theta}_{\backslash m})}[\ln \hat{p}_i(\boldsymbol{\theta}_m|\boldsymbol{\theta}_{\backslash m})]\\ &= \ln \hat{p}_i(\boldsymbol{\theta}_m|\mathbb{E}_q[\boldsymbol{\phi}(\boldsymbol{\theta}_{\backslash m})]) \end{aligned}$$

Here, the joint distribution can be expressed as $p(\boldsymbol{\theta}, \mathcal{D}) = \hat{p}_i(\boldsymbol{\theta}) \propto f_i(\boldsymbol{\theta})q(\boldsymbol{\theta})$. Similar to ADF, we can optimize each variable group through (23) and then multiply the respective distributions together to obtain the new approximate estimate.

It is worth noting that the algorithm can be easily extended to scenarios where data arrive in a batch version. Additionally, standard VI with i.i.d. observations can also be easily modified to a streaming version through this framework. Moreover, it can be seen that the primary difference between streaming VMP and ADF is that the expectations are taken with respect to different distributions. Based on the connection between VMP and CEP, we can present the following corollary.

**Corollary 2:** *Consider a conjugate-exponential probabilistic model represented as a Bayesian network. Suppose the variational distribution follows the mean-field assumption and the observations are i.i.d. Then, streaming VMP and ADF yield the same update equations under the following conditions:*

- *The current approximation $q(\boldsymbol{\theta}_{\backslash m})$ is used as an surrogate of $\hat{p}_i(\boldsymbol{\theta}_{\backslash m})$ in the computation of the expectation in ADF;*

- *The expectations in ADF are calculated using the delta approximation method.*

*Proof:* See Appendix D.

Since VMP is a special case of VI, it follows that streaming VI also has the same update equations to ADF under these conditions, provided that the underlying probabilistic model is a conjugate-exponential model.

Note that the streaming version of VMP or CEP performs a one-pass update, discarding the data once they are updated, which significantly reduces the storage requirements. Additionally, the variable update in VMP can be implemented in a distributed manner since the messages can also be calculated in parallel. The resulting algorithm is similar to the distributed VMP (Masegosa et al., 2016).

### 3.3 Interpretation via Graphical Models

Since both VMP and CEP are closely related to graphical models, we can gain further insights into their connection from a graphical model perspective. Specifically, we assume that the model takes the form of a Bayesian network, and the joint distribution can be expressed as[4]

$$p(\mathbf{V}) = \prod_i p(\mathbf{v}_i | \mathrm{pa}_i), \tag{24}$$

where $\mathbf{V} = \{\boldsymbol{\theta}, \mathcal{D}\}$ contains all the visible and hidden variables; $\mathrm{pa}_i$ denotes the set of variables corresponding to the parents of node $i$; and $\mathbf{v}_i$ denotes the variable or group of variables associated with node $i$. An example of the considered Bayesian network is given in Fig. 2(a).

Assume that the variational distribution is fully factorized with respect to the hidden variables, which means each variable group has only one variable. In VMP, the optimized form for each variable is given by

$$
\begin{aligned}
\ln q^*(\boldsymbol{\theta}_j) &= \mathbb{E}_{q(\boldsymbol{\theta}_{\backslash j})}[\ln p(\boldsymbol{\theta}, \mathcal{D})] + \mathrm{const2} \\
&= \langle \ln p(\mathbf{V}) \rangle_{q(\boldsymbol{\theta}_{\backslash j})} + \mathrm{const2},
\end{aligned}
\tag{25}
$$

where $\langle \cdot \rangle_{q(\boldsymbol{\theta}_{\backslash j})}$ denotes the expectation with respect to $q(\boldsymbol{\theta}_{\backslash j})$. Substituting the joint probability distribution (24) into (25) leads to:

$$\ln q^*(\boldsymbol{\theta}_j) = \left\langle \sum_i \ln p(\mathbf{v}_i | \mathrm{pa}_i) \right\rangle_{q(\boldsymbol{\theta}_{\backslash j})} + \mathrm{const2}.$$

Here we only need to consider the variables in the Markov blanket of node $j$ since the terms that do not depend on $\boldsymbol{\theta}_j$ are constant under the expectation. Then we have

$$\ln q^*(\boldsymbol{\theta}_j) = \langle \ln p(\boldsymbol{\theta}_j | \mathrm{pa}_j) \rangle_{q(\boldsymbol{\theta}_{\backslash j})} + \sum_{k \in \mathrm{ch}_j} \langle \ln p(\mathbf{v}_k | \mathrm{pa}_k) \rangle_{q(\boldsymbol{\theta}_{\backslash j})} + \mathrm{const5}, \tag{26}$$

where $\mathrm{ch}_j$ denotes the index set that corresponds to the children of node $j$. The parent node of $\mathbf{v}_k$ includes the node $j$ and the co-parents $\mathrm{cp}_j$.

In a conjugate-exponential model, we have

$$\ln p(\boldsymbol{\theta}_j | \mathrm{pa}_j) = \boldsymbol{\eta}_j(\mathrm{pa}_j)^T \boldsymbol{\phi}_j(\theta_j) + Z_j(\mathrm{pa}_j) + \ln h_j(\boldsymbol{\theta}_j), \tag{27}$$

and

$$
\begin{aligned}
\ln p(\mathbf{v}_k | \mathrm{pa}_k) &= \boldsymbol{\eta}_k(\boldsymbol{\theta}_j, \mathrm{cp}_j)^T \boldsymbol{\phi}_k(\mathbf{v}_k) + Z_k(\boldsymbol{\theta}_j, \mathrm{cp}_j) + \ln h_k(\mathbf{v}_k) \\
&= \boldsymbol{\eta}_{kj}(\mathbf{v}_k, \mathrm{cp}_j)^T \boldsymbol{\phi}_j(\boldsymbol{\theta}_j) + \lambda(\mathbf{v}_k, \mathrm{cp}_j),
\end{aligned}
\tag{28}
$$

---

[4]We adopt a similar notation to that used in the original VMP paper. A more detailed discussion about the basics of graphical models can be found in Bishop (2006).

where $\lambda$ is a function that contains the terms irrelevant to $\boldsymbol{\eta}_{kj}$ and $\boldsymbol{\phi}_j(\boldsymbol{\theta}_j)$. The second equation holds due to the conjugacy property. Substituting (27) and (28) into (26) will give

$$\ln q^*(\boldsymbol{\theta}_j) = \left[ \left\langle \boldsymbol{\eta}_j(\mathrm{pa}_j) \right\rangle_{q(\boldsymbol{\theta}_{\backslash j})} + \sum_{k \in \mathrm{ch}_j} \left\langle \boldsymbol{\eta}_{kj}(\mathbf{v}_k, \mathrm{cp}_j) \right\rangle_{q(\boldsymbol{\theta}_{\backslash j})} \right]^T \boldsymbol{\phi}_j(\boldsymbol{\theta}_j) + \ln h_j(\boldsymbol{\theta}_j) + \mathrm{const6}.$$

It follows that the optimal variational distribution $q^*(\boldsymbol{\theta}_j)$ is also an exponential family distribution and has the same form as $p(\boldsymbol{\theta}_j|\mathrm{pa}_j)$, of which the natural parameter is given by

$$\boldsymbol{\eta}_j^* = \left\langle \boldsymbol{\eta}_j(\mathrm{pa}_j) \right\rangle + \sum_{k \in \mathrm{ch}_j} \left\langle \boldsymbol{\eta}_{kj}(\mathbf{v}_k, \mathrm{cp}_j) \right\rangle, \tag{29}$$

where the expectation are with respect to $q(\boldsymbol{\theta}_{\backslash j})$ and we omit it here for notational simplicity. Equation (29) can also be interpreted as merging the messages sent by the nearby nodes. As the probabilistic model is conjugate-exponential, we can reparameterise these functions in terms of these expectations, which leads to

$$\tilde{\boldsymbol{\eta}}_j^* = \tilde{\boldsymbol{\eta}}_j(\{\langle \boldsymbol{\phi}_s \rangle\}_{s \in \mathrm{pa}_j}) + \sum_{k \in \mathrm{ch}_j} \tilde{\boldsymbol{\eta}}_{kj}(\langle \boldsymbol{\phi}_k \rangle, \{\langle \boldsymbol{\phi}_t \rangle\}_{t \in \mathrm{cp}_k}).$$

To show the connection between VMP and CEP, consider a conjugate-exponential model with i.i.d. observations, with a graphical illustration shown in Fig. 2(a). For this model, the natural parameter of the optimal variational distribution in VMP is

$$\tilde{\boldsymbol{\eta}}_j^* = \tilde{\boldsymbol{\eta}}_j(\{\langle \boldsymbol{\phi}_s \rangle\}_{s \in \mathrm{pa}_j}) + \sum_{\mathbf{x}_k \in D} \tilde{\boldsymbol{\eta}}_{kj}(\mathbf{x}_k, \{\langle \boldsymbol{\phi}_t \rangle\}_{t \neq j}). \tag{30}$$

In CEP, the optimal variational distribution is also in the exponential family and can be expressed as

$$\ln q^*(\boldsymbol{\theta}_j) = \ln p(\boldsymbol{\theta}_j|\mathrm{pa}_j) + \sum_{k=1}^N \ln \tilde{f}_k(\boldsymbol{\theta}_j), \tag{31}$$

where $\ln p(\boldsymbol{\theta}_j|\mathrm{pa}_j)$ is given by (27). We need to determine the form of $\ln \tilde{f}_k(\boldsymbol{\theta}_j)$. From Lemma 5, we have

$$\ln \tilde{f}_k(\boldsymbol{\theta}_j) = \mathbb{E}_{q(\boldsymbol{\theta}_{\backslash j})}[\ln f_k(\boldsymbol{\theta}_j, \boldsymbol{\theta}_{\backslash j})]. \tag{32}$$

From (28), the likelihood is also an exponential family distribution and can be expressed as

$$\ln f_k(\boldsymbol{\theta}_j, \boldsymbol{\theta}_{\backslash j}) = \ln p(\mathbf{x}_k|\boldsymbol{\theta}_j, \boldsymbol{\theta}_{\backslash j}) \tag{33}$$
$$= \boldsymbol{\eta}_{kj}(\mathbf{v}_k, \mathrm{cp}_j)^T \boldsymbol{\phi}_j(\boldsymbol{\theta}_j) + \lambda(\mathbf{v}_k, \mathrm{cp}_j).$$

By substituting (33) into (32), the approximate factor can be expressed as

$$\ln \tilde{f}_k(\boldsymbol{\theta}_j) = \tilde{\boldsymbol{\eta}}_{kj}(\mathbf{x}_k, \{\langle \boldsymbol{\phi}_t \rangle\}_{t \neq j})^T \boldsymbol{\phi}_j(\boldsymbol{\theta}_j) + \lambda(\mathbf{x}_k, \{\langle \boldsymbol{\phi}_t \rangle\}_{t \neq j})). \tag{34}$$

Substituting it into (31), the resulting distribution shares the same natural parameters as in (30).

Generally, the factors $\{\tilde{f}_k\}$ can be interpreted as the messages sent from the data nodes, after replacing the message sent from the other co-parent nodes with the corresponding moments. Additionally, for a fully factorized model, the standard EP will reduce to loopy belief propagation. More discussions concerning the performance and convergence of LBP can be found in Frey & MacKay (1997); Li et al. (2019); Du et al. (2018a).

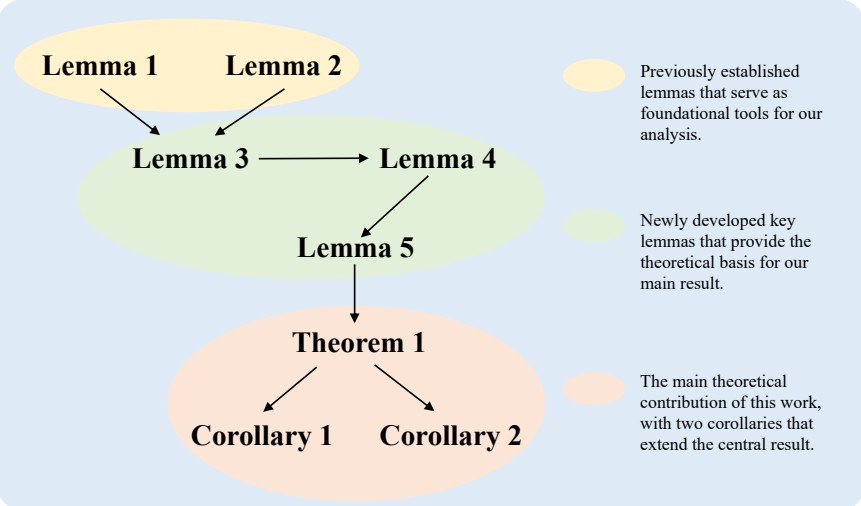

Figure 3: A summary of the theoretical results and their connections in this study.

### 3.4 Summary and Practical Suggestions

The previous subsections have unveiled some relationships among various ABI methods, shedding light on their theoretical properties. This subsection presents a brief summary and connections among our established theoretical results (see Fig. 3) and provides recommendations on applying these findings to address practical inference problems.

We begin by outlining the theoretical connections developed in this study, as illustrated in Fig. 3. **Lemma 1** introduces the concept of moment matching, showing that minimizing the KL divergence between an exponential-family distribution and a target distribution results in matching their moments. **Lemma 2** shows that minimizing the KL divergence between a target distribution and a factorized distribution leads to matching their marginal distributions. Building on these two results, **Lemma 3** establishes a connection between moment matching and the minimization of the KL divergence for factorized distributions. This connection further allows the derivation of a closed-form expression for the message factor in CEP under certain conditions, as formalized in **Lemma 4**.

Next, **Lemma 5** provides a sufficient condition under which CEP and VMP yield the same updates. Combining the results from **Lemmas 4** and **5**, we derive **Theorem 1**, the central result of this paper, which demonstrates the equivalence between the updates of CEP and VMP under specific conditions. Based on this theorem, two corollaries are obtained. **Corollary 1** establishes that CEP is guaranteed to converge under certain conditions, thus providing theoretical justification for its use in various applications. **Corollary 2** reveals that ADF and streaming VMP share the same update rules under similar conditions, offering insights for designing streaming or online variants of ABI methods.

Based on the theoretical results developed in this work, we offer practical recommendations for selecting and applying different ABI methods:

- *Convergent CEP*: As established in **Corollary 1**, the CEP algorithm is guaranteed to converge under the conditions specified in **Theorem 1**. This provides a clear guideline for safely applying CEP in practice without concern for convergence issues, which have traditionally been one of the main limitations of EP-based methods.

- *Streaming or parallel VMP*: The update steps of VMP can be interpreted as aggregating messages from other variable groups. This observation, supported by our theoretical analysis, shows that both algorithms are naturally compatible with streaming or parallel settings when the conditions

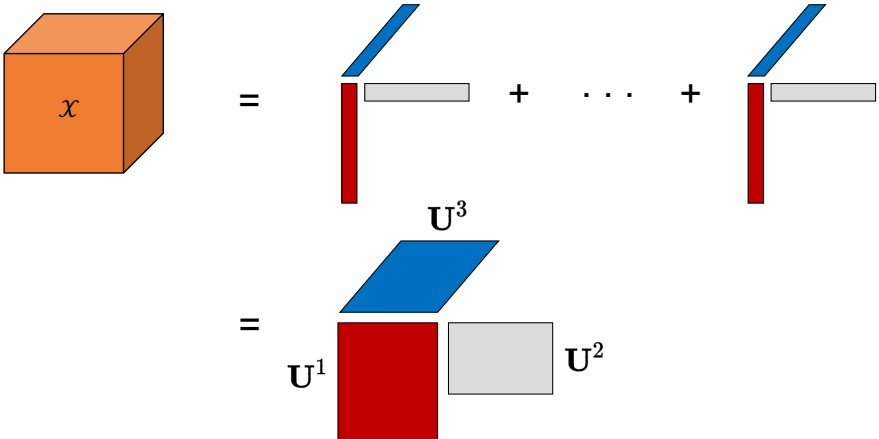

Figure 4: A graphical illustration of the three-dimensional CPD.

in **Theorem 1** are met. In particular, (23) provides a closed-form formulation for constructing streaming variants of VMP by leveraging their structural similarities with CEP and ADF.

- *Streaming ABI from scratch*: When designing a streaming ABI algorithm from the ground up, a natural first step is to assess whether the model satisfies the conjugate-exponential assumption with independent variable partitions. If this condition holds, streaming VMP offers efficient closed-form updates. Otherwise, one may consider using moment matching to project the posterior distribution into a tractable exponential family form. Our results, particularly **Corollary 2**, provide a theoretical connection between ADF and streaming VMP, offering a unified view that can guide algorithmic choices based on model structure and computational constraints.

While these suggestions represent direct applications of our theoretical findings, the broader insights revealed in this work open up new possibilities for designing more flexible and efficient Bayesian inference algorithms. We believe that exploring these directions, especially in the context of large-scale or streaming data, is a promising avenue for future research.

## 4 Example

In this section, we demonstrate the strong connections between the updates of VMP and CEP in the context of a Bayesian tensor decomposition model. Our emphasis is on the canonical polyadic decomposition (CPD), which is an essential technique in machine learning and has been used in various real-world applications. Our choice of Bayesian CPD as the illustrative example is motivated by its prominent role in the original CEP paper (Wang & Zhe, 2020), where the inference algorithm is extensively discussed. We start by introducing a probabilistic model for the CPD approach. To this end, we apply both VMP and CEP to infer the associated posterior distribution. We then extend the resulting algorithm to a streaming setting, enabling it to process sequentially arriving data. Finally, we evaluate the proposed method on an image completion task, demonstrating the equivalence of the two approaches and their effectiveness in practical scenarios.

### 4.1 Probabilistic modeling

We denote a $K$-mode tensor by $\mathcal{X} \in \mathbb{R}^{d_1 \times \cdots \times d_K}$, where $d_k$ is the dimension of the $k$-th mode. The entry value at location $\mathbf{i} = (i_1, \cdots, i_K)$ is denoted as $x_{\mathbf{i}}$. To perform tensor decomposition, we introduce an $R$-dimensional embedding vector $\mathbf{u}_j^k$ to represent each object in mode $k$. Then, a $d_k \times R$ matrix can be constructed by stacking all the embedding vectors in mode $k$, i.e., $\mathbf{U}^k = [\mathbf{u}_1^k, \cdots, \mathbf{u}_{d_k}^k]^T$. Tensor decomposition aims to find the embedding matrices of all modes $\mathcal{U} = \{\mathbf{U}^1, \cdots, \mathbf{U}^K\}$ from the observed entries.

Mathematically, the CPD of a given tensor $\mathcal{X}$ is written as

$$\mathcal{X} = [\![\mathbf{U}^1, \cdots, \mathbf{U}^K]\!],$$

where $[\![\cdot]\!]$ is the Kruskal operator. A graphical illustration of a three-dimensional CPD ($K = 3$) is shown in Fig. 4. For each entry $x_\mathbf{i}$, we have

$$x_\mathbf{i} = \sum_{r=1}^{R} \prod_{k=1}^{K} u_{i_k,r}^k = \mathbf{1}^T(\mathbf{u}_{i_1}^1 \circ \cdots \circ \mathbf{u}_{i_K}^K),$$

where $\circ$ is the Hadamard product.

Consider a $K$-mode tensor $\mathcal{Y}$ with $N$ observed entries denoted as $\{y_\mathbf{i}\}_{\mathbf{i} \in \mathcal{S}}$. Here, $\mathcal{S}$ represents the index set, and its cardinality is $|\mathcal{S}| = N$. We assume that the observations are contaminated with i.i.d. Gaussian noise. Then the likelihood can be expressed as

$$p(y_\mathbf{i}|\mathcal{U}, \tau) = \mathcal{N}(y_\mathbf{i}|\mathbf{1}^T(\mathbf{u}_{i_1}^1 \circ \cdots \circ \mathbf{u}_{i_K}^K), \tau^{-1}),$$

where $\tau$ is the noise precision. We further assign a conjugate Gamma prior over $\tau$, given by

$$p(\tau|a_0, b_0) = \mathrm{Gam}(\tau|a_0, b_0).$$

For each embedding vector $\mathbf{u}_s^k$, we assign a Gaussian prior with mean $\boldsymbol{\beta}_s^k$ and covariance $v\mathbf{I}$, given by

$$p(\mathcal{U}) = \prod_{k=1}^{K} \prod_{s=1}^{d_k} \mathcal{N}(\mathbf{u}_s^k|\boldsymbol{\beta}_s^k, v\mathbf{I}),$$

where $\{\boldsymbol{\beta}_s^k\}$ and $v$ are pre-defined hyperparameters.

Consequently, the joint probability distribution is

$$\begin{aligned} p(\{y_\mathbf{i}\}_{\mathbf{i} \in \mathcal{S}}, \mathcal{U}, \tau) = &\mathrm{Gam}(\tau|a_0, b_0) \prod_{k=1}^{K} \prod_{s=1}^{d_k} \mathcal{N}(\mathbf{u}_s^k|\boldsymbol{\beta}_s^k, v\mathbf{I}) \\ &\cdot \prod_{\mathbf{i} \in \mathcal{S}} \mathcal{N}(y_\mathbf{i}|\mathbf{1}^T(\mathbf{u}_{i_1}^1 \circ \cdots \circ \mathbf{u}_{i_K}^K), \tau^{-1}). \end{aligned} \quad (35)$$

Note that the prior and likelihood are conjugate and belong to the exponential family, thus the probabilistic model is a conjugate-exponential model. Additionally, the observations are assumed to be i.i.d., therefore satisfying the conditions in Theorem 1.

## 4.2 VMP

In VMP, the variables are assumed to be mutually independent, allowing us to factorize the variational distribution as

$$q(\mathcal{U}, \tau) = q(\tau) \prod_{k=1}^{K} \prod_{s=1}^{d_k} q(\mathbf{u}_s^k).$$

Since the probabilistic model is conjugate-exponential, the variational distribution for each variable is identical to its prior distribution. Consequently, the variational distribution is parameterized by

$$q(\mathcal{U}, \tau) = \mathrm{Gam}(\tau|a, b) \prod_{k=1}^{K} \prod_{s=1}^{d_k} \mathcal{N}(\mathbf{u}_s^k|\boldsymbol{\mu}_s^k, \boldsymbol{\Sigma}_s^k).$$

Due to the conjugacy property, we can derive closed-form updates for each variable. Here we present the key steps and leave the detailed derivation in Appendix E. Specifically, the optimal variational distribution of $\mathbf{u}_s^k$ is given by

$$q^*(\mathbf{u}_s^k) = \mathcal{N}(\mathbf{u}_s^k | \boldsymbol{\mu}_s^{k^*}, \boldsymbol{\Sigma}_s^{k^*}),$$

with the mean $\boldsymbol{\mu}_s^{k^*}$ and covariance $\boldsymbol{\Sigma}_s^{k^*}$ given by

$$\boldsymbol{\mu}_s^{k^*} = \boldsymbol{\Sigma}_s^{k^*} \left( \langle \tau \rangle \sum_{\mathbf{i} \in \mathcal{S}, i_k = s} y_{\mathbf{i}} \langle \mathbf{z}_{\mathbf{i}}^{\backslash k} \rangle + v \boldsymbol{\beta}_s^k \right), \tag{36}$$

$$\boldsymbol{\Sigma}_s^{k^*} = \left( \langle \tau \rangle \sum_{\mathbf{i} \in \mathcal{S}, i_k = s} \langle \mathbf{z}_{\mathbf{i}}^{\backslash k} \mathbf{z}_{\mathbf{i}}^{\backslash k^T} \rangle + v \mathbf{I} \right)^{-1},$$

where $\langle \cdot \rangle$ denotes the expectation $\mathbb{E}_q[\cdot]$ and

$$\mathbf{z}_{\mathbf{i}}^{\backslash k} = \mathbf{u}_{i_1}^1 \circ \cdots \circ \mathbf{u}_{i_{k-1}}^{k-1} \circ \mathbf{u}_{i_{k+1}}^{k+1} \circ \cdots \circ \mathbf{u}_{i_K}^K.$$

The optimal variational distribution of noise precision $\tau$ is given by

$$q^*(\tau) = \mathrm{Gam}(\tau | a^*, b^*),$$

with $a^*$ and $b^*$ computed as follows

$$a^* = a_0 + \frac{N}{2},$$

$$b^* = b_0 + \frac{1}{2} \sum_{\mathbf{i} \in \mathcal{S}} [y_{\mathbf{i}}^2 - 2 y_{\mathbf{i}} \langle \mathbf{1}^T \mathbf{z}_{\mathbf{i}} \rangle + \langle (\mathbf{1}^T \mathbf{z}_{\mathbf{i}})^2 \rangle],$$

where $\mathbf{z}_{\mathbf{i}} = \mathbf{u}_{i_1}^1 \circ \cdots \circ \mathbf{u}_{i_K}^K$.

## 4.3 CEP

In the context of CEP, the approximation factor $\tilde{f}_{\mathbf{i}}$ is assumed to be factorized with variables, given by

$$\tilde{f}_{\mathbf{i}}(\mathcal{U}, \tau) = \tilde{f}_{\mathbf{i}}(\tau) \prod_{k=1}^K \tilde{f}_{\mathbf{i}}^k(\mathbf{u}_{i_k}^k),$$

where the factors have the same form as the prior distribution but with different parameters. Specifically, $\tilde{f}_{\mathbf{i}}(\tau) = \mathrm{Gam}(\tau | a_{\mathbf{i}}, b_{\mathbf{i}})$ and $\tilde{f}_{\mathbf{i}}^k(\mathbf{u}_{i_k}^k) = \mathcal{N}(\mathbf{u}_{i_k}^k | \mathbf{m}_{\mathbf{i}}^k, \mathbf{S}_{\mathbf{i}}^k)$. Consequently, the approximate distribution is given by

$$q(\mathcal{U}, \tau) \propto \mathrm{Gam}(\tau | a_0, b_0) \prod_{k=1}^K \prod_{s=1}^{d_k} \mathcal{N}(\mathbf{u}_s^k | \boldsymbol{\beta}_s^k, v \mathbf{I})$$

$$\cdot \prod_{\mathbf{i} \in \mathcal{S}} \tilde{f}_{\mathbf{i}}(\tau) \prod_{k=1}^K \tilde{f}_{\mathbf{i}}^k(\mathbf{u}_{i_k}^k).$$

It can be seen that the approximate distribution is factorized over the variables, i.e.,

$$q(\mathcal{U}, \tau) = q(\tau) \prod_{k=1}^K \prod_{s=1}^{d_k} q(\mathbf{u}_s^k),$$

where

$$q(\tau) \propto \text{Gam}(\tau|a_0, b_0) \prod_{\mathbf{i} \in \mathcal{S}} \tilde{f}_{\mathbf{i}}(\tau), \tag{37}$$

$$q(\mathbf{u}_s^k) \propto \mathcal{N}(\mathbf{u}_s^k|\boldsymbol{\beta}_s^k, v\mathbf{I}) \prod_{\mathbf{i} \in \mathcal{S}, i_k = s} \tilde{f}_{\mathbf{i}}^k(\mathbf{u}_{i_k}^k).$$

To update the variational distribution $q(\mathbf{u}_s^k)$, we need to determine the optimal factor $\tilde{f}_{\mathbf{i}}^k(\mathbf{u}_{i_k}^k)$. The first step is to obtain the calibrating distribution

$$q^{\backslash \mathbf{i}}(\mathcal{U}, \tau) \propto \frac{q(\mathcal{U}, \tau)}{\tilde{f}_{\mathbf{i}}(\tau) \prod_{k=1}^K \tilde{f}_{\mathbf{i}}^k(\mathbf{u}_{i_k}^k)}.$$

Next, we construct the tilted distribution as

$$\hat{p}_{\mathbf{i}}(\mathcal{U}, \tau) \propto q^{\backslash \mathbf{i}}(\mathcal{U}, \tau) \mathcal{N}(y_{\mathbf{i}}|\mathbf{1}^T(\mathbf{u}_{i_1}^1 \circ \cdots \circ \mathbf{u}_{i_K}^K), \tau^{-1}).$$

Since only the moments for the precision $\tau$ and the embedding vectors that associate with entry $\mathbf{i}$, $\mathbf{u_i} = \{\mathbf{u}_{i_1}^1, \cdots, \mathbf{u}_{i_k}^K\}$, are needed and the other embeddings vectors will be marginalized out, we can focus on the marginal titled distribution for $\{\mathbf{u_i}, \tau\}$,

$$\hat{p}_{\mathbf{i}}(\mathbf{u_i}, \tau) \propto q^{\backslash \mathbf{i}}(\tau) \prod_{k=1}^K q^{\backslash \mathbf{i}}(\mathbf{u}_{i_k}^k) \mathcal{N}(y_{\mathbf{i}}|\mathbf{1}^T(\mathbf{u}_{i_1}^1 \circ \cdots \circ \mathbf{u}_{i_K}^K), \tau^{-1}),$$

where

$$q^{\backslash \mathbf{i}}(\tau) = \text{Gam}(\tau|a^{\backslash \mathbf{i}}, b^{\backslash \mathbf{i}}), \quad q^{\backslash \mathbf{i}}(\mathbf{u}_{i_k}^k) = \mathcal{N}(\mathbf{u}_{i_k}^k|\mathbf{m}_{i_k}^k, \mathbf{S}_{i_k}^k),$$

with

$$a^{\backslash \mathbf{i}} = a_0 + \sum_{\mathbf{j} \in \mathcal{S}, \mathbf{j} \neq \mathbf{i}} a_{\mathbf{j}} - N + 1,$$

$$b^{\backslash \mathbf{i}} = b_0 + \sum_{\mathbf{j} \in \mathcal{S}, \mathbf{j} \neq \mathbf{i}} b_{\mathbf{j}},$$

$$\mathbf{S}_{i_k}^k = \left( \sum_{\mathbf{j} \in \mathcal{S}, \mathbf{j} \neq \mathbf{i}, j_k = i_k} (\mathbf{S}_{\mathbf{j}}^k)^{-1} + v\mathbf{I} \right)^{-1},$$

$$\mathbf{m}_{i_k}^k = \mathbf{S}_{i_k}^k \left( \sum_{\mathbf{j} \in \mathcal{S}, \mathbf{j} \neq \mathbf{i}, j_k = i_k} (\mathbf{S}_{\mathbf{j}}^k)^{-1} \mathbf{m}_{\mathbf{j}}^k + v\boldsymbol{\beta}_{i_k}^k \right).$$

The next step is to compute conditional moments with respect to the conditional tilted distribution given $\tau$ and $\mathbf{u_i}^{\backslash k} = \{\mathbf{u}_{i_1}^1, \cdots, \mathbf{u}_{i_K}^K\}$ fixed, which can be expressed as

$$\hat{p}_{\mathbf{i}}(\mathbf{u}_{i_k}^k|\mathbf{u_i}^{\backslash k}, \tau) \propto \mathcal{N}(\mathbf{u}_{i_k}^k|\mathbf{m}_{i_k}^k, \mathbf{S}_{i_k}^k) \mathcal{N}(y_{\mathbf{i}}|\mathbf{1}^T(\mathbf{u}_{i_1}^1 \circ \cdots \circ \mathbf{u}_{i_K}^K), \tau^{-1}).$$

It can be observed that this is a Gaussian distribution with covariance and mean given by

$$\text{cov}(\mathbf{u}_{i_k}^k|\mathbf{u_i}^{\backslash k}, \tau) = \left[ (\mathbf{S}_{i_k}^k)^{-1} + \tau(\mathbf{z_i}^{\backslash k} \mathbf{z_i}^{\backslash k^T}) \right]^{-1},$$

$$\mathbb{E}(\mathbf{u}_{i_k}^k|\mathbf{u_i}^{\backslash k}, \tau) = \text{cov}(\mathbf{u}_{i_k}^k|\mathbf{u_i}^{\backslash k}, \tau) \left[ (\mathbf{S}_{i_k}^k)^{-1} \mathbf{m}_{i_k}^k + \tau y_{\mathbf{i}} \mathbf{z_i}^{\backslash k} \right].$$

According to Lemma 4, the optimal factor is given by $\tilde{f}_{\mathbf{i}}^k(\mathbf{u}_{i_k}^k) = \mathcal{N}(\mathbf{u}_{i_k}^k | \mathbf{m}_{\mathbf{i}}^{k*}, \mathbf{S}_{\mathbf{i}}^{k*})$ with

$$\mathbf{S}_{\mathbf{i}}^{k*} = \left( \langle \tau \rangle \langle \mathbf{z}_{\mathbf{i}}^{\backslash k} \mathbf{z}_{\mathbf{i}}^{\backslash k^T} \rangle \right)^{-1},$$

$$\mathbf{m}_{\mathbf{i}}^{k*} = \mathbf{S}_{\mathbf{i}}^{k*} (y_{\mathbf{i}} \langle \tau \rangle \langle \mathbf{z}_{\mathbf{i}}^{\backslash k} \rangle).$$

It is worth noting that the message factors can be calculated in parallel, which can significantly reduce time consumption. After obtaining all the message factors, we can merge them to obtain the approximation distribution. Based on (37), the optimal variational distribution for $q(\mathbf{u}_s^k)$ is given by $q^*(\mathbf{u}_s^k) = \mathcal{N}(\mathbf{u}_s^k | \boldsymbol{\mu}_s^{k*}, \boldsymbol{\Sigma}_s^{k*})$, where

$$\boldsymbol{\mu}_s^{k*} = \boldsymbol{\Sigma}_s^{k*} \left( \langle \tau \rangle \sum_{\mathbf{i} \in \mathcal{S}, i_k = s} y_{\mathbf{i}} \langle \mathbf{z}_{\mathbf{i}}^{\backslash k} \rangle + v \boldsymbol{\beta}_s^k \right), \tag{38}$$

$$\boldsymbol{\Sigma}_s^{k*} = \left( \langle \tau \rangle \sum_{\mathbf{i} \in \mathcal{S}, i_k = s} \langle \mathbf{z}_{\mathbf{i}}^{\backslash k} \mathbf{z}_{\mathbf{i}}^{\backslash k^T} \rangle + v \mathbf{I} \right)^{-1}.$$

Comparing (36) and (38), it can be seen that the optimal variational distributions obtained by VMP and CEP are the same.

For noise precision $\tau$, the conditional tilted distribution is given by

$$\hat{p}_{\mathbf{i}}(\tau | \mathbf{u}_{\mathbf{i}}) = \text{Gam}(\tau | \hat{a}, \hat{b}), \tag{39}$$

where

$$\hat{a} = a^{\backslash \mathbf{i}} + \frac{1}{2}, \tag{40}$$

$$\hat{b} = b^{\backslash \mathbf{i}} + \frac{1}{2} [y_{\mathbf{i}} - \mathbf{1}^T \mathbf{z}_{\mathbf{i}}]^2.$$

Then the optimal message factor can be calculated as $\tilde{f}_{\mathbf{i}}(\tau) = \text{Gam}(\tau | a_{\mathbf{i}}^*, b_{\mathbf{i}}^*)$ with

$$a_{\mathbf{i}}^* = \frac{1}{2}, \tag{41}$$

$$b_{\mathbf{i}}^* = \frac{1}{2} [y_{\mathbf{i}}^2 - 2y_{\mathbf{i}} \langle \mathbf{1}^T \mathbf{z}_{\mathbf{i}} + \langle (\mathbf{1}^T \mathbf{z}_{\mathbf{i}})^2 \rangle].$$

Merging these factors through (37) will leads to $q^*(\tau) = \text{Gam}(\tau | a^*, b^*)$, where

$$a^* = a_0 + \frac{N}{2},$$

$$b^* = b_0 + \frac{1}{2} \sum_{\mathbf{i} \in \mathcal{S}} [y_{\mathbf{i}}^2 - 2y_{\mathbf{i}} \langle \mathbf{1}^T \mathbf{z}_{\mathbf{i}} \rangle + \langle (\mathbf{1}^T \mathbf{z}_{\mathbf{i}})^2 \rangle],$$

which are the same as in VMP. Consequently, we can conclude that the update of variables in VMP and CEP are the same. These closed-form updates demonstrate promising accuracy and empirically show a fast convergence in many real-world applications (Wang & Zhe, 2020).

## 4.4 Streaming VMP

The connection of VMP and CEP enables the algorithm to be easily adapted to a streaming version. In the streaming version, we assume that every time we receive one data point and the current approximation is $q(\mathbf{u}_s^k) = \mathcal{N}(\mathbf{u}_s^k | \boldsymbol{\mu}_s^k, \boldsymbol{\Sigma}_s^k)$ and $q(\tau) = \text{Gam}(\tau | a, b)$. Then based on (23), the posterior update of the streaming version is given by $q^*(\mathbf{u}_s^k) = \hat{p}_{\mathbf{i}}(\mathbf{u}_{i_k}^k | \langle \mathbf{u}_{\mathbf{i}}^{\backslash k} \rangle, \langle \tau \rangle) = \mathcal{N}(\mathbf{u}_s^k | \boldsymbol{\mu}_s^{k*}, \boldsymbol{\Sigma}_s^{k*})$ with

$$\boldsymbol{\Sigma}_s^{k*} = [(\boldsymbol{\Sigma}_s^k)^{-1} + \langle \tau \rangle (\langle \mathbf{z}_{\mathbf{i}}^{\backslash k} \mathbf{z}_{\mathbf{i}}^{\backslash k^T} \rangle)]^{-1}, \tag{42}$$

$$\boldsymbol{\mu}_s^{k*} = \boldsymbol{\Sigma}_s^{k*} [(\boldsymbol{\Sigma}_s^k)^{-1} \boldsymbol{\mu}_s^k + \langle \tau \rangle y_{\mathbf{i}} \langle \mathbf{z}_{\mathbf{i}}^{\backslash k} \rangle],$$

Table 1: Reconstruction error (RMSE) for different images using VMP and CEP.

| Image | VMP | CEP |
|---|---|---|
| facade | 0.0184 | 0.0184 |
| car | 0.0365 | 0.0365 |
| airplane | 0.0360 | 0.0360 |
| sailboat | 0.0403 | 0.0403 |

and $q^*(\tau) = \hat{p}_\mathbf{i}(\tau|\langle \mathbf{u_i}\rangle) = \mathrm{Gam}(\tau|a^*, b^*)$ with

$$a^* = a + \frac{1}{2},$$
$$b^* = b + \frac{1}{2}[y_\mathbf{i}^2 - 2y_\mathbf{i}\langle \mathbf{1}^T\mathbf{z_i}\rangle + \langle(\mathbf{1}^T\mathbf{z_i})^2\rangle].$$

It is important to note that the natural parameters used here are different from those used in CEP (see (42) and (38)) since the calibrating distribution is replaced with the full approximation from the previous iteration. Additionally, the developed algorithm can be readily extended to the scenario where data are arrived in a batch version. The resulting algorithm is the same as probabilistic streaming tensor decomposition (POST) (Du et al., 2018b), which is flexible and demonstrates promising performance in this task.

## 4.5 Numerical Experiments

We evaluate the proposed method on the image inpainting task using four RGB benchmark images[5], each represented as a third-order tensor of size $256 \times 256 \times 3$. The CP tensor rank is set to 40, and all pixel values are normalized to the range $[0, 1]$. A random sampling strategy is used, with an observation rate of 20%. The metirc for performance evaluation is the root mean squared error (RMSE). Table 1 reports the average reconstruction error across five independent trials for both VMP and CEP on each image. As expected, the reconstruction results of the two methods are identical, reflecting their equivalent update rules.

Fig. 5 shows a representative example of the convergence behavior for both methods, plotting RMSE against the number of iterations. Both methods exhibit the same convergence trends and reach a low reconstruction error within a few iterations, demonstrating the effectiveness of Bayesian CP decomposition for image completion. Additionally, Fig. 6 presents the visual reconstruction results of VMP and CEP. The recovered images are visually indistinguishable, further confirming the theoretical equivalence of the two methods in practice.

Furthermore, to assess the robustness of both methods beyond the conditions assumed in Theorem 1, we conduct two additional experiments. First, we consider a setting where the group-wise update assumption of CEP is violated. Instead of aggregating messages for each variable group before updating, we adopt a entry-wise update strategy commonly used in standard EP or online implementations, where each update is based on a single data point. As shown in the left panel of Figure 7, CEP still converges reliably despite the lack of a formal convergence guarantee. Moreover, this entry-wise update strategy reduces the number of iterations required, though each iteration incurs higher computational cost.

Second, we test performance of both methods under heteroscedastic noise. Each observed entry is corrupted by Gaussian noise with a data-dependent variance, where $\sigma_i \sim \mathcal{N}(0, 1)$ and the noise is sampled as $\mathcal{N}(0, \sigma_i^2)$. As shown in the right panel of Figure 7, both VMP and CEP maintain identical RMSE levels, demonstrating stability under this non-i.i.d. setting. These results suggest that the convergence and performance consistency between VMP and CEP extend beyond the idealized conditions in the theoretical analysis.

---

[5]Avaialble: http://sipi.usc.edu/database/database.php

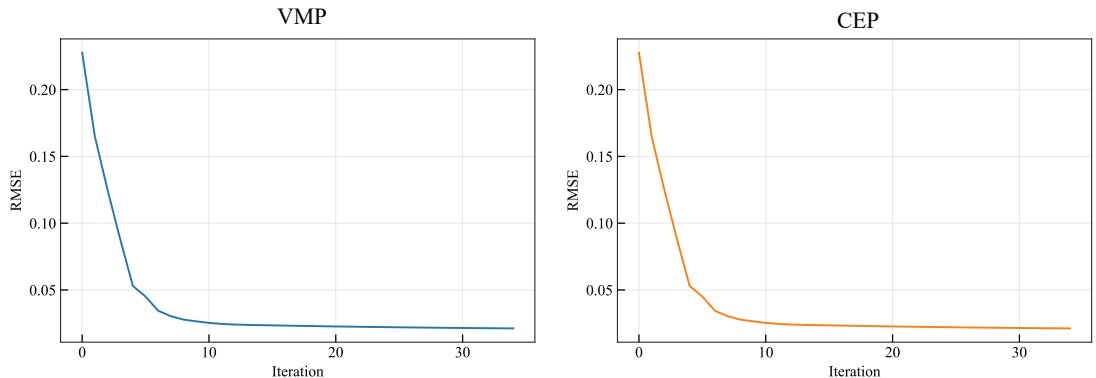

Figure 5: RMSE versus number of iterations for VMP and CEP on the *sailboat* image.

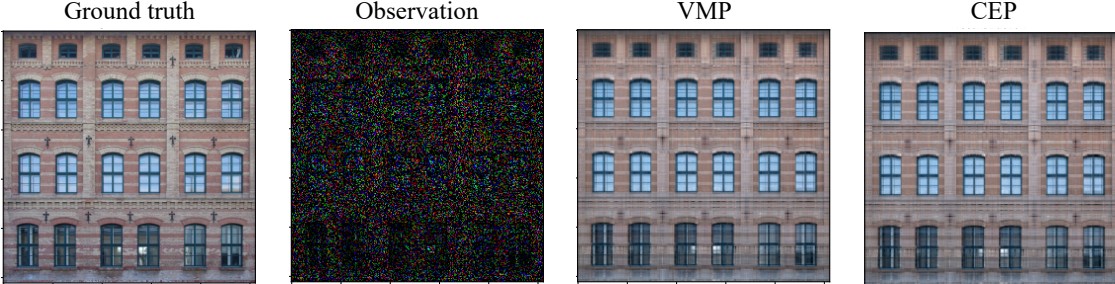

Figure 6: Visual comparison of reconstructed *facade* image using VMP and CEP.

## 5 Discussions and Future Directions

This paper investigates the theoretical connections among different ABI methods, starting by bridging the VMP and CEP. Specifically, we have demonstrated a strong link between these two methods under mild conditions. This newly identified connection not only guarantees the convergence of CEP but also allows for the seamless construction of a streaming version of the VMP algorithm. The key insight is that the variable updates in VMP and CEP are intrinsically merging the messages sent by all the data points and they share a common objective of approximating the conditional marginal distribution. Additionally, this finding provides insights into the underlying relationships and distinct characteristics of other ABI methods, including the same expressions between ADF and streaming VI updates.

Generally, VMP and CEP are considered distinct classes of ABI methods, each with different properties and performance characteristics. This work, for the first time, establishes a close connection between them under mild conditions, providing new insights into the structure of these algorithms and informing the development of more advanced inference methods. However, our theoretical analysis is restricted to the conjugate-exponential family of models. It would be interesting to explore the application of these connections in other model families or non-conjugate scenarios. We believe that these explorations will open new avenues for future research on efficient and accurate Bayesian learning algorithms, particularly in the context of streaming and large-scale data.

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

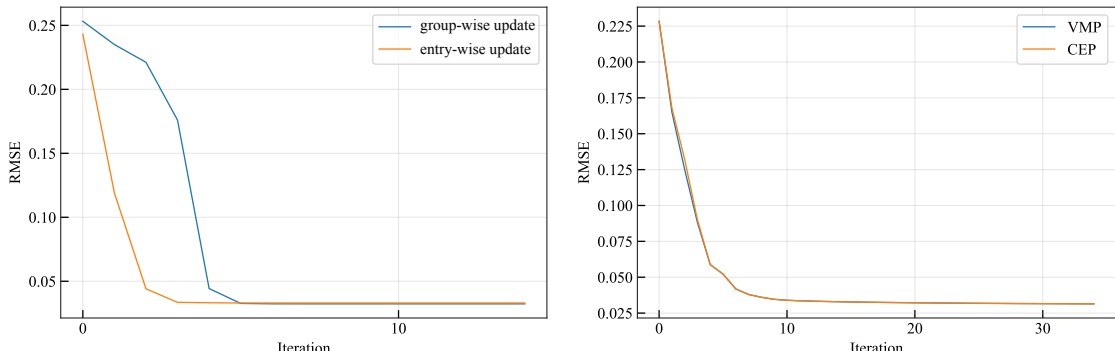

Figure 7: RMSE curves under violation of Theorem 1 assumptions. Left: Convergence behavior of CEP under different update strategies. Right: RMSE curves of VMP and CEP with non-i.i.d. Gaussian noise.

Tamara Broderick, Nicholas Boyd, Andre Wibisono, Ashia C Wilson, and Michael I Jordan. Streaming variational bayes. *Advances in neural information processing systems*, 26, 2013.

Lawrence D Brown. Fundamentals of statistical exponential families: with applications in statistical decision theory. Institute of Mathematical Statistics, 1986.

Antoni B. Chan and Nuno Vasconcelos. Layered dynamic textures. *IEEE Transactions on Pattern Analysis and Machine Intelligence*, 31(10):1862–1879, 2009.

Lei Cheng, Zhongtao Chen, Qingjiang Shi, Yik-Chung Wu, and Sergios Theodoridis. Towards flexible sparsity-aware modeling: Automatic tensor rank learning using the generalized hyperbolic prior. *IEEE Transactions on Signal Processing*, 70:1834–1849, 2022a.

Lei Cheng, Feng Yin, Sergios Theodoridis, Sotirios Chatzis, and Tsung-Hui Chang. Rethinking bayesian learning for data analysis: The art of prior and inference in sparsity-aware modeling. *IEEE Signal Processing Magazine*, 39(6):18–52, 2022b.

Shay B. Cohen and Noah A. Smith. Covariance in unsupervised learning of probabilistic grammars. *Journal of Machine Learning Research*, 11(101):3017–3051, 2010.

Thomas M Cover. *Elements of information theory.* John Wiley & Sons, 1999.

Jean Daunizeau, Vincent Adam, and Lionel Rigoux. Vba: a probabilistic treatment of nonlinear models for neurobiological and behavioural data. *PLoS computational biology*, 10(1):e1003441, 2014.

Jian Du, Shaodan Ma, Yik-Chung Wu, Soummya Kar, and José M. F. Moura. Convergence analysis of distributed inference with vector-valued gaussian belief propagation. *Journal of Machine Learning Research*, 18(172):1–38, 2018a.

Yishuai Du, Yimin Zheng, Kuang-chih Lee, and Shandian Zhe. Probabilistic streaming tensor decomposition. In *2018 IEEE International Conference on Data Mining (ICDM)*, pp. 99–108, 2018b.

Wentao Fan, Lin Yang, and Nizar Bouguila. Unsupervised grouped axial data modeling via hierarchical bayesian nonparametric models with watson distributions. *IEEE Transactions on Pattern Analysis and Machine Intelligence*, 44(12):9654–9668, 2022.

Shikai Fang, Robert M. Kirby, and Shandian Zhe. Bayesian streaming sparse tucker decomposition. In Cassio de Campos and Marloes H. Maathuis (eds.), *Proceedings of the Thirty-Seventh Conference on Uncertainty in Artificial Intelligence*, volume 161 of *Proceedings of Machine Learning Research*, pp. 558–567. PMLR, 27–30 Jul 2021a.

Shikai Fang, Zheng Wang, Zhimeng Pan, Ji Liu, and Shandian Zhe. Streaming bayesian deep tensor factorization. In *International Conference on Machine Learning*, pp. 3133–3142. PMLR, 2021b.

Brendan J Frey and David MacKay. A revolution: Belief propagation in graphs with cycles. *Advances in neural information processing systems*, 10, 1997.

Christopher Heje Grønbech, Maximillian Fornitz Vording, Pascal N Timshel, Casper Kaae Sønderby, Tune H Pers, and Ole Winther. scVAE: variational auto-encoders for single-cell gene expression data. *Bioinformatics*, 36(16):4415–4422, 05 2020.

Leonard Hasenclever, Stefan Webb, Thibaut Lienart, Sebastian Vollmer, Balaji Lakshminarayanan, Charles Blundell, and Yee Whye Teh. Distributed bayesian learning with stochastic natural gradient expectation propagation and the posterior server. *Journal of Machine Learning Research*, 18(106):1–37, 2017.

Michael I. Jordan, Zoubin Ghahramani, Tommi S. Jaakkola, and Lawrence K. Saul. *An introduction to variational methods for graphical models*, pp. 105161. MIT Press, Cambridge, MA, USA, 1999. ISBN 0262600323.

Mohammad Emtiyaz Khan and Håvard Rue. The bayesian learning rule. *Journal of Machine Learning Research*, 24(281):1–46, 2023.

Bin Li, Qinliang Su, and Yik-Chung Wu. Fixed points of gaussian belief propagation and relation to convergence. *IEEE Transactions on Signal Processing*, 67(23):6025–6038, 2019.

Bin Li, Nan Wu, and Yik-Chung Wu. Distributed inference with variational message passing in gaussian graphical models: Trade-offs in message schedules and convergence conditions. *IEEE Transactions on Signal Processing*, 2024.

Ximing Li, Changchun Li, Jinjin Chi, Jihong Ouyang, and Wenting Wang. Black-box expectation propagation for bayesian models. In *Proceedings of the 2018 siam international conference on data mining*, pp. 603–611. SIAM, 2018.

Yingzhen Li, José Miguel Hernández-Lobato, and Richard E Turner. Stochastic expectation propagation. *Advances in neural information processing systems*, 28, 2015.

Andres R Masegosa, Ana M Martinez, Helge Langseth, Thomas D Nielsen, Antonio Salmeron, Dario Ramos-Lopez, and Anders L Madsen. d-vmp: Distributed variational message passing. In *Conference on Probabilistic Graphical Models*, pp. 321–332. PMLR, 2016.

P.S. Maybeck. *Stochastic Models, Estimation, and Control*. Mathematics in Science and Engineering. Elsevier Science, 1982. ISBN 9780124807037.

Thomas Minka. Power ep. Technical report, Microsoft Research, Cambridge, 2004.

Thomas Minka. Divergence measures and message passing. Technical Report TR-2005-173, Microsoft Research, 2005.

Thomas P. Minka. Expectation propagation for approximate bayesian inference, 2013.

Thomas P. Minka and Rosalind Picard. *A family of algorithms for approximate Bayesian inference*. PhD thesis, USA, 2001.

Kevin P. Murphy. *Probabilistic Machine Learning: An introduction*. MIT Press, 2022.

Manfred Opper, Ole Winther, and Michael J Jordan. Expectation consistent approximate inference. *Journal of Machine Learning Research*, 6(12), 2005.

Zhimeng Pan, Zheng Wang, and Shandian Zhe. Streaming nonlinear bayesian tensor decomposition. In Jonas Peters and David Sontag (eds.), *Proceedings of the 36th Conference on Uncertainty in Artificial Intelligence (UAI)*, volume 124 of *Proceedings of Machine Learning Research*, pp. 490–499. PMLR, 03–06 Aug 2020.

Alex J Smola, SVN Vishwanathan, and Eleazar Eskin. Laplace propagation. In *Advances in Neural Inf. Proc. Systems (NIPS)*, pp. 441–448, 2004.

Jae Woong Soh and Nam Ik Cho. Variational deep image restoration. *IEEE Transactions on Image Processing*, 31:4363–4376, 2022.

S. Theodoridis. *Machine Learning: From the Classics to Deep Networks, Transformers, and Diffusion Models.* 3nd Ed., Academic Press, 2025.

Aki Vehtari, Andrew Gelman, Tuomas Sivula, Pasi Jylänki, Dustin Tran, Swupnil Sahai, Paul Blomstedt, John P Cunningham, David Schiminovich, and Christian P Robert. Expectation propagation as a way of life: A framework for bayesian inference on partitioned data. *Journal of Machine Learning Research*, 21 (17):1–53, 2020.

Martin J. Wainwright and Michael I. Jordan. Graphical models, exponential families, and variational inference. *Foundations and Trends in Machine Learning*, 1(1-2):1–305, 2008. ISSN 1935-8237.

Chong Wang and David M Blei. Variational inference in non-conjugate models. *Journal of Machine Learning Research*, 2013.

Zheng Wang and Shandian Zhe. Conditional expectation propagation. In *Uncertainty in Artificial Intelligence*, pp. 28–37. PMLR, 2020.

John Winn, Christopher M Bishop, and Tommi Jaakkola. Variational message passing. *Journal of Machine Learning Research*, 6(4), 2005.

Boyang Xue, Jianwei Yu, Junhao Xu, Shansong Liu, Shoukang Hu, Zi Ye, Mengzhe Geng, Xunying Liu, and Helen M. Meng. Bayesian transformer language models for speech recognition. *ICASSP 2021 - 2021 IEEE International Conference on Acoustics, Speech and Signal Processing (ICASSP)*, pp. 7378–7382, 2021.

Cheng Zhang, Judith Bütepage, Hedvig Kjellström, and Stephan Mandt. Advances in variational inference. *IEEE Transactions on Pattern Analysis and Machine Intelligence*, 41(8):2008–2026, 2019.

Jackson Zhou, John T Ormerod, and Clara Grazian. Fast expectation propagation for heteroscedastic, lasso-penalized, and quantile regression. *Journal of Machine Learning Research*, 24(314):1–39, 2023.

## A   Summary of Algorithms

The procedure of the VMP and CEP methods are summarized in **Algorithm 1** and **Algorithm 2**, respectively.

## B   Lemmas

Here we present some useful lemmas to offer a deeper understanding of the exponential family and the KL divergence.

**Lemma 1**(Minka, 2013): *If $p(\boldsymbol{\theta})$ is an arbitrary fixed distribution and $q(\boldsymbol{\theta})$ is in the exponential family, then minimizing the divergence $KL(p\|q)$ with respect to $q$ gives*

$$\mathbb{E}_{q(\boldsymbol{\theta})}[\phi(\boldsymbol{\theta})] = \mathbb{E}_{p(\boldsymbol{\theta})}[\phi(\boldsymbol{\theta})],$$

*where $\phi(\boldsymbol{\theta})$ is the sufficient statistics of $q(\boldsymbol{\theta})$.*

Lemma 1, commonly referred to as moment matching or moment projection, reveals that the KL divergence can be minimized by equating expectations of the sufficient statistics of $q(\boldsymbol{\theta})$ to their expectations with respect to $p(\boldsymbol{\theta})$. It is noteworthy that if $p(\boldsymbol{\theta})$ belongs to the exponential family and shares the same sufficient statistics as $q(\boldsymbol{\theta})$ (i.e., they possess the same distributional form), the moment matching procedure guarantees that their natural parameters become identical. As exponential family distributions are uniquely determined by their sufficient statistics and natural parameters, moment matching leads to the equality of $q(\boldsymbol{\theta})$ and $p(\boldsymbol{\theta})$. Consequently, the KL divergence between the two distributions is reduced to zero.

---

**Algorithm 1** Variational Message Passing (VMP)

---

**Input:** joint probability distribution $p(\mathcal{D}, \boldsymbol{\theta})$.
1: Initialise each factor distribution $q(\boldsymbol{\theta}_m)$.
2: **while** not converge **do**
3:    **for** each variable group **do**
4:       Calculate moment of the natural parameters $\mathbb{E}_{q(\boldsymbol{\theta}_{\backslash m})}[\boldsymbol{\eta}_m(\boldsymbol{\theta}_{\backslash m}, \mathcal{D})]$ using the messages sent from other nodes.
5:       Update the factor distribution $q^*(\boldsymbol{\theta}_m)$ via (6).
6:    **end for**
7: **end while**
**Output:** variational distribution $q(\boldsymbol{\theta}) = \prod_m q^*(\boldsymbol{\theta}_m)$.

---

**Lemma 2**(Bishop, 2006): *Assume $p(\boldsymbol{\theta})$ is a fixed distribution and $q(\boldsymbol{\theta})$ factorizes with respect to variable groups, i.e.,*

$$q(\boldsymbol{\theta}) = \prod_m q(\boldsymbol{\theta}_m),$$

*then minimizing the divergence $KL(p\|q)$ with respect to q gives*

$$q^*(\boldsymbol{\theta}_m) = p(\boldsymbol{\theta}_m), \forall m. \tag{43}$$

Lemma 2 shows that the optimal solution of each factor distribution $q(\boldsymbol{\theta}_m)$ is given by the corresponding marginal distribution of $p(\boldsymbol{\theta})$.

## C   The Delta Approximation Method

The delta approximation method approximates the expectation of a function of a random variable by evaluating the function at the mean of that variable. Specifically, for a function $f(\boldsymbol{\theta})$ and a distribution $q(\boldsymbol{\theta})$, the expectation can be approximated as:

$$\mathbb{E}_{q(\boldsymbol{\theta})}[f(\boldsymbol{\theta})] = \int f(\boldsymbol{\theta})q(\boldsymbol{\theta})\mathrm{d}\boldsymbol{\theta} \approx \int f(\boldsymbol{\theta})\delta(\boldsymbol{\theta} - \mathbf{m})\mathrm{d}\boldsymbol{\theta} = f(\mathbf{m}),$$

where $\delta(\cdot)$ is the Dirac delta function and $\mathbf{m}$ is the mean of $q(\boldsymbol{\theta})$. Since the Dirac delta function can be viewed as the limiting case of a Gaussian distribution with vanishing variance, the delta approximation method can be regarded as a special case of the Laplace approximation (Bishop, 2006).

This approximation can also be interpreted from the perspective of a first-order Taylor expansion. Given a differentiable function $f(\boldsymbol{\theta})$ and a distribution $q(\boldsymbol{\theta})$ with mean $\mathbf{m}$, the first-order Taylor expansion of $f$ around $\mathbf{m}$ yields:

$$\mathbb{E}_{q(\boldsymbol{\theta})}(f(\boldsymbol{\theta})) \approx \mathbb{E}_q \left[ f(\mathbf{m}) + (\boldsymbol{\theta} - \mathbf{m})^T \nabla_{\boldsymbol{\theta}} f(\mathbf{m}) \right] \approx f(\mathbf{m}),$$

where $\nabla$ is the differential operator. In the context of CEP, a similar approximation can be applied to nested expectations. For example, the outer expectation of a function $h(\boldsymbol{\phi}(\boldsymbol{\theta}_{\backslash m}))$ can be approximated as:

$$\mathbb{E}_{q(\boldsymbol{\theta}_{\backslash m})}[h(\boldsymbol{\phi}(\boldsymbol{\theta}_{\backslash m}))] \approx \mathbb{E}_q \left[ h(\mathbb{E}_q(\boldsymbol{\phi}(\boldsymbol{\theta}_{\backslash m}))) + (\boldsymbol{\phi}(\boldsymbol{\theta}_{\backslash m}) - \mathbb{E}_q(\boldsymbol{\phi}(\boldsymbol{\theta}_{\backslash m})))^T \nabla h(\mathbb{E}_q(\boldsymbol{\phi}(\boldsymbol{\theta}_{\backslash m}))) \right] \approx h(\mathbb{E}_q(\boldsymbol{\phi}(\boldsymbol{\theta}_{\backslash m}))).$$

Note that in CEP the approximation is applied to the sufficient statistics, rather than the variables themselves.

---

**Algorithm 2** Conditional Expectation Propagation (CEP)

---

**Input:** joint probability distribution $p(\mathcal{D}, \boldsymbol{\theta})$.
 1: Initialise each message factor $\tilde{f}_i(\boldsymbol{\theta}_m)$.
 2: **while** not converge **do**
 3:  **for** each variable group **do**
 4:   **for** each factor $\tilde{f}_i(\boldsymbol{\theta}_m)$ **do**
 5:    Calculate the calibrating distribution, $q^{\backslash i}(\boldsymbol{\theta}_m) = q(\boldsymbol{\theta}_m)/\tilde{f}_i(\boldsymbol{\theta}_m)$.
 6:    Derive a new posterior $q^{\natural}(\boldsymbol{\theta}_m)$ via conditional moment matching (11).
 7:    Update the message factor $\tilde{f}_i(\boldsymbol{\theta}_m) \propto q^{\natural}(\boldsymbol{\theta}_m)/q^{\backslash i}(\boldsymbol{\theta}_m)$.
 8:   **end for**
 9:   Merge the message: $q^*(\boldsymbol{\theta}_m) \propto \prod_i \tilde{f}_i(\boldsymbol{\theta}_m)$.
10:  **end for**
11: **end while**
**Output:** variational distribution $q(\boldsymbol{\theta}) = \prod_m q^*(\boldsymbol{\theta}_m)$.

---

# D Proofs

## D.1 Proof of Lemma 3

To prove Lemma 3, we rewrite the KL divergence as

$$\mathrm{KL}(p\|q) = \int p(\boldsymbol{\theta}) \ln \frac{p(\boldsymbol{\theta})}{q(\boldsymbol{\theta})} d\boldsymbol{\theta}$$

$$= H[p(\boldsymbol{\theta})] - \int p(\boldsymbol{\theta}) \ln q(\boldsymbol{\theta}) d\boldsymbol{\theta},$$

where $H[\cdot]$ is the entropy. Since the entropy is a constant, minimizing $\mathrm{KL}(p\|q)$ is equivalent to maximizing $\mathcal{L}(q) = \int p(\boldsymbol{\theta}) \ln q(\boldsymbol{\theta}) d\boldsymbol{\theta}$. Exploiting the factorized property, it can be further decomposed as

$$\mathcal{L}(q) = \int p(\boldsymbol{\theta}) \sum_m \ln q(\boldsymbol{\theta}_m) d\boldsymbol{\theta}$$

$$= \sum_m \int p(\boldsymbol{\theta}) \ln q(\boldsymbol{\theta}_m) d\boldsymbol{\theta}$$

$$= \sum_m \int \left( \int p(\boldsymbol{\theta}) d\boldsymbol{\theta}_{\backslash m} \right) \ln q(\boldsymbol{\theta}_m) d\boldsymbol{\theta}_m$$

$$= \sum_m \int p(\boldsymbol{\theta}_m) \ln q(\boldsymbol{\theta}_m) d\boldsymbol{\theta}_m$$

$$= \sum_m L_m(q(\boldsymbol{\theta}_m)),$$

where we denote $\int p(\boldsymbol{\theta}_m) \ln q(\boldsymbol{\theta}_m) d\boldsymbol{\theta}_m$ as $L_m(q(\boldsymbol{\theta}_m))$. Since the variable groups are mutually independent, maximizing $\mathcal{L}(q)$ with respect to $q(\boldsymbol{\theta})$ is equivalent to maximizing each $L_m$ with respect to $q(\boldsymbol{\theta}_m)$. For each variable group, the optimum is given by

$$\max_{q(\boldsymbol{\theta}_m)} L_m(q(\boldsymbol{\theta}_m)) = \max_{q(\boldsymbol{\theta}_m)} \int p(\boldsymbol{\theta}_m) \ln q(\boldsymbol{\theta}_m) d\boldsymbol{\theta}_m$$

$$= \min_{q(\boldsymbol{\theta}_m)} - \int p(\boldsymbol{\theta}_m) \ln q(\boldsymbol{\theta}_m) d\boldsymbol{\theta}_m$$

$$= \min_{q(\boldsymbol{\theta}_m)} H[p(\boldsymbol{\theta}_m)] - \int p(\boldsymbol{\theta}_m) \ln q(\boldsymbol{\theta}_m) d\boldsymbol{\theta}_m$$

$$= \min_{q(\boldsymbol{\theta}_m)} \mathrm{KL}(p(\boldsymbol{\theta}_m)\|q(\boldsymbol{\theta}_m)),$$

where the third equation holds because the entropy of the marginal distribution $H[p(\boldsymbol{\theta}_m)]$ is irrelevant to $q(\boldsymbol{\theta}_m)$. Using Lemma 1, the optimal solution is achieved by the moment matching

$$\mathbb{E}_{q(\boldsymbol{\theta}_m)}[\phi(\boldsymbol{\theta}_m)] = \mathbb{E}_{p(\boldsymbol{\theta}_m)}[\phi(\boldsymbol{\theta}_m)].$$

Additionally, the left-hand side of (12) can be expressed as

$$\begin{aligned} \mathbb{E}_{q(\boldsymbol{\theta})}[\phi(\boldsymbol{\theta}_m)] &= \int q(\boldsymbol{\theta})\phi(\boldsymbol{\theta}_m)d\boldsymbol{\theta} \\ &= \int \left[ \int q(\boldsymbol{\theta}_m, \boldsymbol{\theta}_{\backslash m})d\boldsymbol{\theta}_{\backslash m} \right] \phi(\boldsymbol{\theta}_m)d\boldsymbol{\theta}_m \\ &= \int q(\boldsymbol{\theta}_m)\phi(\boldsymbol{\theta}_m)d\boldsymbol{\theta}_m \\ &= \mathbb{E}_{q(\boldsymbol{\theta}_m)}[\phi(\boldsymbol{\theta}_m)]. \end{aligned}$$

Similarly, the right-hand side of (12) can be expressed as $\mathbb{E}_{p(\boldsymbol{\theta})}[\phi(\boldsymbol{\theta}_m)] = \mathbb{E}_{p(\boldsymbol{\theta}_m)}[\phi(\boldsymbol{\theta}_m)]$. Thus we have

$$\mathbb{E}_{q(\boldsymbol{\theta})}[\phi(\boldsymbol{\theta}_m)] = \mathbb{E}_{q(\boldsymbol{\theta}_m)}[\phi(\boldsymbol{\theta}_m)] = \mathbb{E}_{p(\boldsymbol{\theta}_m)}[\phi(\boldsymbol{\theta}_m)] = \mathbb{E}_{p(\boldsymbol{\theta})}[\phi(\boldsymbol{\theta}_m)],$$

which completes the proof. ∎

### D.2 Proof of Lemma 4

From (13) and Lemma 3, it can be seen that the calculation of $q^{\natural}(\boldsymbol{\theta}_m)$ in CEP is essentially solving the following problem

$$\begin{aligned} &\min_{q(\boldsymbol{\theta}_m)} \mathrm{KL}(\hat{p}_i(\boldsymbol{\theta})\|q(\boldsymbol{\theta})) \\ &\text{s.t. } q(\boldsymbol{\theta}) = \prod_m q(\boldsymbol{\theta}_m), \end{aligned}$$

where $q(\boldsymbol{\theta}_m)$ belongs to the exponential family. If the $\hat{p}_i(\boldsymbol{\theta}_m)$ is also in the exponential family and has the same form as $q^{\natural}(\boldsymbol{\theta}_m)$, then the moment matching leads to

$$q^{\natural}(\boldsymbol{\theta}_m) = \hat{p}_i(\boldsymbol{\theta}_m),$$

where the marginal posterior can be further written as

$$\begin{aligned} \hat{p}_i(\boldsymbol{\theta}_m) &= \int \hat{p}_i(\boldsymbol{\theta}_m, \boldsymbol{\theta}_{\backslash m})d\boldsymbol{\theta}_{\backslash m} \\ &= \int \hat{p}_i(\boldsymbol{\theta}_{\backslash m})\hat{p}_i(\boldsymbol{\theta}_m|\boldsymbol{\theta}_{\backslash m})d\boldsymbol{\theta}_{\backslash m} \\ &= \mathbb{E}_{\hat{p}_i(\boldsymbol{\theta}_{\backslash m})}[\hat{p}_i(\boldsymbol{\theta}_m|\boldsymbol{\theta}_{\backslash m})]. \end{aligned}$$

In CEP, two approximations are made to derive an analytical form of the update. The first is to use $q(\boldsymbol{\theta}_{\backslash m})$ as a surrogate for $\hat{p}_i(\boldsymbol{\theta}_{\backslash m})$. The second is to use the delta approximation method to approximate the expectation of the conditional distribution. Based on these approximations, the optimal approximate posterior $q^{\natural}(\boldsymbol{\theta}_m)$ can be expressed as

$$\begin{aligned} q^{\natural}(\boldsymbol{\theta}_m) &= \mathbb{E}_{\hat{p}_i(\boldsymbol{\theta}_{\backslash m})}[\hat{p}_i(\boldsymbol{\theta}_m|\boldsymbol{\theta}_{\backslash m})] \\ &\approx \mathbb{E}_{q(\boldsymbol{\theta}_{\backslash m})}[\hat{p}_i(\boldsymbol{\theta}_m|\boldsymbol{\theta}_{\backslash m})] \\ &\approx \hat{p}_i(\boldsymbol{\theta}_m|\mathbb{E}_q[\phi(\boldsymbol{\theta}_{\backslash m})]). \end{aligned}$$

Note that the delta approximation is made on the sufcient statistics, rather than the random variables themselves.

Generally, $\hat{p}_i(\boldsymbol{\theta}_m)$ is not in the exponential family, so the moment matching step is used to minimize the KL divergence. However, in a conjugate-exponential model, each complete conditional, including $\hat{p}_i(\boldsymbol{\theta}_m|\boldsymbol{\theta}_{\backslash m})$, is in the exponential family. Additionally, $\hat{p}_i(\boldsymbol{\theta}_m|\boldsymbol{\theta}_{\backslash m})$ shares the same sufficient statistics as $q^{\natural}(\boldsymbol{\theta}_m)$ due to the conjugacy property. As a result, $\hat{p}_i(\boldsymbol{\theta}_m|\mathbb{E}_q[\boldsymbol{\phi}(\boldsymbol{\theta}_{\backslash m})])$ is used as a surrogate for $q^{\natural}(\boldsymbol{\theta}_m)$ in CEP. Thus the update of $\tilde{f}_i(\boldsymbol{\theta}_m)$ can be expressed as

$$\tilde{f}_i(\boldsymbol{\theta}_m) \propto \frac{\hat{p}_i(\boldsymbol{\theta}_m|\mathbb{E}_q[\boldsymbol{\phi}(\boldsymbol{\theta}_{\backslash m})])}{q^{\backslash i}(\boldsymbol{\theta}_m)},$$

which completes the proof. ∎

### D.3 Proof of Corollary 1

It has been established in Winn et al. (2005); Minka (2005) that VMP updates are guaranteed to converge to a local minimum of the KL divergence under the conditions stated in Theorem 1. Since CEP follows the same update equations as VMP under these conditions, its convergence property directly follows.

### D.4 Proof of Corollary 2

In ADF, the optimal variational distribution in each iteration can be expressed as

$$q^*(\boldsymbol{\theta}_m) = \hat{p}_i(\boldsymbol{\theta}_m) = \mathbb{E}_{\hat{p}_i(\boldsymbol{\theta}_{\backslash m})}[\hat{p}_i(\boldsymbol{\theta}_m|\boldsymbol{\theta}_{\backslash m})].$$

With the two conditions in Corollary 2, the optimal distribution can be reformulated as

$$\begin{aligned} q^*(\boldsymbol{\theta}_m) &= \mathbb{E}_{\hat{p}_i(\boldsymbol{\theta}_{\backslash m})}[\hat{p}_i(\boldsymbol{\theta}_m|\boldsymbol{\theta}_{\backslash m})] \\ &\approx \mathbb{E}_{q(\boldsymbol{\theta}_{\backslash m})}[\hat{p}_i(\boldsymbol{\theta}_m|\boldsymbol{\theta}_{\backslash m})] \\ &\approx \hat{p}_i(\boldsymbol{\theta}_m|\mathbb{E}_q[\boldsymbol{\phi}(\boldsymbol{\theta}_{\backslash m})]). \end{aligned}$$

Similarly, the optimal distribution in streaming VMP is given by

$$\begin{aligned} \ln q^*(\boldsymbol{\theta}_m) &= \mathbb{E}_{q(\boldsymbol{\theta}_{\backslash m})}[\ln \hat{p}_i(\boldsymbol{\theta}_m|\boldsymbol{\theta}_{\backslash m})] \\ &= \ln \hat{p}_i(\boldsymbol{\theta}_m|\mathbb{E}_q[\boldsymbol{\phi}(\boldsymbol{\theta}_{\backslash m})]), \end{aligned}$$

which is the same as in ADF. This completes the proof.

## E Derivation of the VMP in Tensor Decomposition

In the Bayesian tensor decomposition problem, the unknown parameter set $\boldsymbol{\theta}$ consists of the latent factor matrices $\mathcal{U}$ and hyperparameter $\tau$. The optimal variational distribution for each $\boldsymbol{\theta}_m$ is given by

$$\ln q^*(\boldsymbol{\theta}_m) = \mathbb{E}_{q(\boldsymbol{\theta}_{\backslash m})}[\ln p(\boldsymbol{\theta}, \mathcal{D})] + \text{const.} \tag{44}$$

From (35), the logarithm of the joint density function $\ln p(\boldsymbol{\theta}, \mathcal{D})$ can be expressed as

$$\begin{aligned} \ln p(\boldsymbol{\theta}, \mathcal{D}) &= \ln p(\{y_{\mathbf{i}}\}_{\mathbf{i} \in \mathcal{S}}, \mathcal{U}, \tau) \tag{45} \\ &= \frac{N}{2} \ln \tau - \sum_{\mathbf{i} \in \mathcal{S}} \frac{\tau}{2}[y_{\mathbf{i}} - \mathbf{1}^T(\mathbf{u}_{i_1}^1 \circ \cdots \circ \mathbf{u}_{i_K}^K)]^2 \\ &\quad - \sum_{k=1}^{K} \sum_{s=1}^{d_k} \frac{v}{2}(\mathbf{u}_s^k - \boldsymbol{\beta}_s^k)^T(\mathbf{u}_s^k - \boldsymbol{\beta}_s^k) \\ &\quad (a_0 - 1) \ln \tau - b_0 \tau + \text{const.} \end{aligned}$$

By substituting (45) into (44), we obtain $q^*(\mathbf{u}_s^k)$:

$$
\begin{aligned}
\ln q^*(\mathbf{u}_s^k) &= \mathbb{E}_q\{-\frac{\tau}{2}\sum_{\mathbf{i}\in\mathcal{S},i_k=s}[y_\mathbf{i}-\mathbf{1}^T(\mathbf{u}_{i_1}^1\circ\cdots\circ\mathbf{u}_{i_K}^K)]^2 \\
&\quad -\frac{v}{2}(\mathbf{u}_s^k-\boldsymbol{\beta}_s^k)^T(\mathbf{u}_s^k-\boldsymbol{\beta}_s^k)\} \\
&= \mathbb{E}_q\{-\frac{\tau}{2}\sum_{\mathbf{i}\in\mathcal{S},i_k=s}\left[y_\mathbf{i}^2-2y_\mathbf{i}(\mathbf{u}_s^k)^T\mathbf{z}_\mathbf{i}^{\backslash k}+(\mathbf{u}_s^k)^T\mathbf{z}_\mathbf{i}^{\backslash k}\mathbf{z}_\mathbf{i}^{\backslash k^T}\mathbf{u}_s^k\right] \\
&\quad -\frac{v}{2}\left[(\mathbf{u}_s^k)^T\mathbf{u}_s^k-2(\mathbf{u}_s^k)^T\boldsymbol{\beta}_s^k+(\boldsymbol{\beta}_s^k)^T\boldsymbol{\beta}_s^k\right]\} \\
&= -\frac{1}{2}(\mathbf{u}_s^k)^T\left[\langle\tau\rangle\sum_{\mathbf{i}\in\mathcal{S},i_k=s}\langle\mathbf{z}_\mathbf{i}^{\backslash k}\mathbf{z}_\mathbf{i}^{\backslash k^T}\rangle+v\mathbf{I}\right]\mathbf{u}_s^k \\
&\quad +(\mathbf{u}_s^k)^T\left[\langle\tau\rangle\sum_{\mathbf{i}\in\mathcal{S},i_k=s}y_\mathbf{i}\langle\mathbf{z}_\mathbf{i}^{\backslash k}\rangle+v\boldsymbol{\beta}_s^k\right].
\end{aligned}
$$

We can see from (46) that $\mathbf{u}_s^k$ follows a Gaussian distribution $q^*(\mathbf{u}_s^k) = \mathcal{N}(\mathbf{u}_s^k|\boldsymbol{\mu}_s^{k*},\boldsymbol{\Sigma}_s^{k*})$, of which the mean and covariance are given by

$$
\boldsymbol{\mu}_s^{k*} = \boldsymbol{\Sigma}_s^{k*}\left(\langle\tau\rangle\sum_{\mathbf{i}\in\mathcal{S},i_k=s}y_\mathbf{i}\langle\mathbf{z}_\mathbf{i}^{\backslash k}\rangle+v\boldsymbol{\beta}_s^k\right),
$$

$$
\boldsymbol{\Sigma}_s^{k*} = \left(\langle\tau\rangle\sum_{\mathbf{i}\in\mathcal{S},i_k=s}\langle\mathbf{z}_\mathbf{i}^{\backslash k}\mathbf{z}_\mathbf{i}^{\backslash k^T}\rangle+v\mathbf{I}\right)^{-1}.
$$

The expression of $q^*(\tau)$ can be found as

$$
\begin{aligned}
\ln q^*(\tau) &= \mathbb{E}_q\{\frac{N}{2}\ln\tau-\sum_{\mathbf{i}\in\mathcal{S}}\frac{\tau}{2}[y_\mathbf{i}-\mathbf{1}^T(\mathbf{u}_{i_1}^1\circ\cdots\circ\mathbf{u}_{i_K}^K)]^2 \\
&\quad (a_0-1)\ln\tau-b_0\tau\},
\end{aligned}
$$

which is a Gamma distribution $q^*(\tau) = \text{Gam}(\tau|a^*,b^*)$ with $a^*$ and $b^*$ given by

$$
a^* = a_0 + \frac{N}{2},
$$

$$
b^* = b_0 + \frac{1}{2}\sum_{\mathbf{i}\in\mathcal{S}}[y_\mathbf{i}^2-2y_\mathbf{i}\langle\mathbf{1}^T\mathbf{z}_\mathbf{i}\rangle+\langle(\mathbf{1}^T\mathbf{z}_\mathbf{i})^2\rangle].
$$

