# OpenReview forum: "Bridging VMP and CEP: Theoretical Insights for Connecting Different Approximate Bayesian Inference Methods"
_TMLR — Rejected by TMLR_

### Review · Reviewer_X5pz · 2025-05-22

**Summary Of Contributions:**

The authors highlight a connection between variational message passing (VMP) and conditional expectation propagation (CEP). They claim that this connection proves convergence of CEP, and motivates a new streaming version of VMP.

**Audience:**

Yes

**Claims And Evidence:**

No

**Requested Changes:**

As discussed above, to the best of my understanding the central result of the paper is flawed, and so it would be difficult for me to recommend acceptance unless this can be resolved or the authors can explain why my concerns are unfounded. I would also hope the authors could show how their contribution on streaming VMP differs from earlier works.  I have other, smaller, concerns that would need addressing in order for me to recommend acceptance, but unless these more fundamental issues can be addressed it does not feel helpful to list them all at this stage.

**Strengths And Weaknesses:**

The main strength of the this paper is that it attempts to highlight interesting connections between the existing approximate inference methods VMP and CEP.  Unfortunately, the main result of this paper -- from which all other contributions follow -- appears to me to follow flawed reasoning.

In the proof of Lemma 4, Appendix C.2
\\begin{align}
q^\natural(\theta_m) &= \mathbb{E}\_{\\hat{p}\_i(\theta_{/m})}\left[ \hat{p}\_i(\\theta\_m\\mid\\theta\_{/m})\right] \\\\
&\approx \mathbb{E}\_{q(\theta_{/m})}\left[ \\hat{p}\_i(\theta\_m\\mid\\theta\_{/m})\right] \\\\
&\approx \mathbb{E}\_{q(\theta_{/m})}\left[ \\hat{p}\_i(\theta\_m\\mid \\mathbb{E}[\\theta\_{/m}])\right]
\\end{align}

The first approximate equality (second line) above is the approximation made by CEP. The second approximate equality (third line), is an additional approximation made by the authors, based on what they call the _multivariate delta method_$^*$ (MDV). The original CEP paper uses a _similar_ trick, but to obtain a Taylor approximation of the the conditional moments (see equation 10 of [1]) - here the authors are obtaining a Taylor approximation of the _density_, which is fundamentally different.

However, if we take for granted that this additional approximation does result in a form of CEP, the bigger issue is in Lemma 5, from which all other results follow. In the proof of Lemma 5, the optimal VMP terms are shown to be
\\[\mathbb{E}\_{q(\theta\_m)}[f\_i(\theta)] = \mathbb{E}\_{q(\theta\_{/m})}\left[\ln \hat{p}_i(\theta\_m\mid\theta\_{/m})\right] - \ln q^{/i}(\theta\_m) \\]
and then, the authors say:

"_by utilizing the multivariate delta approximation, the expectation $\mathbb{E}\_{q(\theta\_m)}[f\_i(\theta)]$  can be expressed as_
\\[\ln \hat{p}_i(\theta\_m\mid\mathbb{E}\_{q(\theta\_{/m})}[\theta\_{/m}]) - \ln q^{/i}(\theta\_m) \\]
_which shows that $\mathbb{E}\_{q(\theta\_m)}[f\_i(\theta)]$ and $\ln\tilde f\_i(\theta\_m)$ are equivalent under the multivariate delta approximation. According to Lemma 5, we can conclude that VMP and CEP yield the same update equations under the
specified conditions._"

This is not showing that the updates of VMP and CEP are equivalent under these conditions, it is showing that a Taylor approximation of the VMP update is equivalent to their own approximation of the CEP update. This is, as far as I can see, a fundamental flaw in the reasoning here, and so any results that follow from this later in the paper (e.g. proving convergence of CEP) are not justified.

The authors also claim that this perspective allows them to construct _'a streaming version of VMP'_. It is not clear to me that the extension of VMP to streaming data is at all related to the earlier theoretical results, but in any case, streaming or online VMP has been explored extensively, see e.g. [2,3,4,5.6].

It is not clear to me what Section 3.3 "Interpretation via Graphical Models" is contributing -- it seems to just introduce a lot of new notation without adding anything meaningful to the paper.

Other, more minor issues:
* Abstract: it is claimed that CEP is widely employed. Is this true? I would want some citations to back up this claim if so. I at least was not familiar with it before reviewing this paper.
* At a couple of points in the paper, it is even claimed that a connection is made between VI and EP. This is even more of a stretch, given that -- for reasons discussed earlier -- I do not believe a meaningful connection has been shown between VMP and CEP.
* Page 6: _"EP also approximates the posterior by minimizing the KL divergence, but in the opposite direction."_ - this is not accurate, EP only performs local minimisations of that KL, it does not minimize it globally.
* The authors use the "reparameterization trick" to refer to something entirely different than most readers would expect.
* Page 14: the authors change the notation used for expectation halfway through the paper (halfway through a series of equations, in fact) from $\mathbb{E}[.]$ to $\langle.\rangle$. Why not use a consistent notation throughout the paper?

[1] Zheng Wang, Shandian Zhe (2020) - _Conditional Expectation Propagation_

[2] Ghahramani and Attias (2000) - _Online variational Bayesian learning_

[3] Sato (2001) - _Online model selection based on the variational Bayes_

[4] Broderick et al. (2013) - _Streaming variational Bayes._

[5] Bui et al. (2017) - _Streaming sparse Gaussian process approximations_

[6] Nguyen et al. (2018) - _Variational continual learning_

*I'm not sure multivariate delta method is the correct terminology here, the authors are just using a first-order Taylor approximation of an expectation of a function around the mean of the random variable.

---

> ### Author Response · Authors · 2025-07-15
>
> We sincerely thank the reviewer for the careful reading and for providing insightful and constructive feedback. We have made revisions to the manuscript based on your comments and suggestions, with the changes highlighted in blue. Below, we summarize and address each of your main concerns:
>
> - Validity and interpretation of the delta approximation
> - Justification of the main theorem
> - Differences with existing streaming Bayes literature
> - Contribution of section "Interpretation via Graphical Models"
>
> **C1**. *In the proof of Lemma 4, Appendix C.2*
>
> $$
> \begin{aligned}
>     q^\natural(\boldsymbol{\theta}_m) &= \mathbb{E}\_{\hat{p}\_i(\boldsymbol{\theta}\_{\setminus m})} [\hat{p}\_i(\boldsymbol{\theta}\_{m}|\boldsymbol{\theta}\_{\setminus m})] \\\\
>     &\approx \mathbb{E}\_{q(\boldsymbol{\theta}\_{\setminus m})} [\hat{p}\_i(\boldsymbol{\theta}\_{m}|\boldsymbol{\theta}\_{\setminus m})]  \\\\
>     &\approx \mathbb{E}\_{q(\boldsymbol{\theta}\_{\setminus m})} [\hat{p}\_i(\boldsymbol{\theta}\_{m}|\mathbb{E}[\boldsymbol{\theta}\_{\setminus m}])].
> \end{aligned}
> $$
>
> *The first approximate equality (second line) above is the approximation made by CEP. The second approximate equality (third line), is an additional approximation made by the authors, based on what they call the multivariate delta method (MDV). The original CEP paper uses a similar trick, but to obtain a Taylor approximation of the conditional moments (see equation 10 of [1]) — here the authors are obtaining a Taylor approximation of the density, which is fundamentally different.*
>
> **R1**. Thank you for pointing this out. As the reviewer correctly noted, we apply a Taylor expansion to the probability density function $\hat{p}\_i(\boldsymbol{\theta}\_{m}|\boldsymbol{\theta}\_{\setminus m})$. However, we would like to clarify that the expansion is performed with respect to the conditional variable $\boldsymbol{\theta}\_{\setminus m}$, not the variable $\boldsymbol{\theta}\_m$. Importantly, $\hat{p}\_i(\boldsymbol{\theta}\_{m}|\boldsymbol{\theta}\_{\setminus m})$ is treated as a likelihood function of the conditional variable $\boldsymbol{\theta}\_{\setminus m}$. Since the approximation is applied to a parameter in the PDF, the result remains a valid PDF, preserving its essential properties.
>
> From a probabilistic perspective, this approximation can be viewed as follows. The expectation of a function $f(\boldsymbol{\theta})$ under a distribution $q(\boldsymbol{\theta})$ can be approximated as:
>
> \begin{equation\*}
>     \mathbb{E}\_{q(\boldsymbol{\theta})}[f(\boldsymbol{\theta})]=\int f(\boldsymbol{\theta}) q(\boldsymbol{\theta}) \text{d}\boldsymbol{\theta} \\\\
>     \approx \int f(\boldsymbol{\theta}) \delta(\boldsymbol{\theta}-\mathbf{m}) \text{d}\boldsymbol{\theta}=f(\mathbf{m}),
> \end{equation\*}
>
> where $\delta(\cdot)$ is the Dirac delta function [1] and $\mathbf{m}$ is the mean of $q(\boldsymbol{\theta})$.
> In our case, the expectation of the likelihood function $\hat{p}\_i(\boldsymbol{\theta}\_{m}|\boldsymbol{\theta}\_{\setminus m})$ with respect to distribution $q(\boldsymbol{\theta}\_{\setminus m})$ is approximated as
> $$
> \begin{aligned}
> \mathbb{E}\_{q(\boldsymbol{\theta}\_{\setminus m})} [\hat{p}\_i(\boldsymbol{\theta}\_{m}|\boldsymbol{\theta}\_{\setminus m})] &= \int \hat{p}\_i(\boldsymbol{\theta}\_{m}|\boldsymbol{\theta}\_{\setminus m}) q(\boldsymbol{\theta}\_{\setminus m}) \, \mathrm{d} \boldsymbol{\theta}\_{\setminus m} \\\\
> &\approx \hat{p}\_i(\boldsymbol{\theta}\_{m}|\mathbb{E}\_q[\boldsymbol{\theta}\_{\setminus m}]).
> \end{aligned}
> $$
>
> Here, the distribution $q(\boldsymbol{\theta}\_{\setminus m})$ is approximated by a multivariate Dirac delta function centered at its mean. This is why we refer to this method as the multivariate delta method. It can also be interpreted as a special case of the Laplace approximation, where the variance of the Gaussian tends to zero, leaving only the mean [2, 3].
>
> To address potential confusion, we have added this explanation to the appendix and updated the terminology from "multivariate delta method" to "delta approximation" for clarity and consistency.
>
> [1] Arfken, G.B., Weber, H.J. and Harris, F.E., 2011. *Mathematical methods for physicists: a comprehensive guide*. Academic press.
>
> [2] Bishop, C.M., 2006. *Pattern recognition and machine learning* . New York: springer.
>
> [3] MacKay, D.J., 2003. *Information theory, inference and learning algorithms*. Cambridge university press.

---

> ### Author Response · Authors · 2025-07-15
>
> **C2**. *This is not showing that the updates of VMP and CEP are equivalent under these conditions, it is showing that a Taylor approximation of the VMP update is equivalent to their own approximation of the CEP update. This is, as far as I can see, a fundamental flaw in the reasoning here, and so any results that follow from this later in the paper (e.g. proving convergence of CEP) are not justified.*
>
> **R2:**  Thank you for raising this concern. We sincerely apologize for any confusion caused by the way our proof was presented. To clarify, VMP does not require a Taylor approximation. The updates derived in VMP are exact under the conditions specified in Theorem 1. Below, we reformulate the proof of Theorem 1 to clearly demonstrate this.
>
> **Theorem 1:** *Consider a conjugate-exponential probabilistic model represented as a Bayesian network. Suppose the variational distribution follows the mean-field assumption and the observations are i.i.d. Then the CEP and VMP yield the same update equations under the following conditions:*
>
> - The update in CEP is performed on the variable groups.
> - The expectations in CEP are approximated using the delta approximation method.
>
> *Proof:* From Lemma 5, a sufficient condition for the equivalence between the update equations of CEP and VMP is
>
> $$
> \begin{align}
> \ln \tilde{f}\_{i}(\boldsymbol{\theta}\_m) &= \mathbb{E}\_{q(\boldsymbol{\theta}\_{\setminus m})}[\ln f_i(\boldsymbol{\theta})]. \tag{1}
> \end{align}
> $$
>
> Thus, we only need to show that equation (1) holds under the conditions specified in Theorem 1. From Lemma 4, the logarithm of message factor $\tilde{f}\_{i}(\boldsymbol{\theta}\_m)$ in CEP can be represented as
>
> $$
> \begin{align}
> \ln \tilde{f}\_{i}(\boldsymbol{\theta}\_m) &= \ln \hat{p}\_i(\boldsymbol{\theta}\_{m}|\mathbb{E}\_q[\boldsymbol{\theta}\_{\setminus m}])- \ln q^{\setminus i}(\boldsymbol{\theta}\_m). \tag{2}
> \end{align}
> $$
> In VMP, the logarithm of the optimal distribution in VMP is given by
>
> $$
> \begin{aligned}
> \ln q^*(\boldsymbol{\theta}\_m) &= \mathbb{E}\_{q(\boldsymbol{\theta}\_{\setminus m})}[\ln p(\boldsymbol{\theta}, \mathcal{D})] + \text{const} \\\\
> &= \mathbb{E}\_{q(\boldsymbol{\theta}\_{\setminus m})}[\sum\_i \ln f\_{i}(\boldsymbol{\theta})] + \text{const} \\\\
> &= \sum\_i \mathbb{E}\_{q(\boldsymbol{\theta}\_{\setminus m})}[\ln f\_{i}(\boldsymbol{\theta})] + \text{const}.
> \end{aligned}
> $$
>
> Each term in the summation can be expressed as:
> $$
> \begin{align}
> \mathbb{E}\_{q(\boldsymbol{\theta}\_{\setminus m})}[\ln f\_{i}(\boldsymbol{\theta})] &= \mathbb{E}\_{q(\boldsymbol{\theta}\_{\setminus m})}[\ln f\_{i}(\boldsymbol{\theta})] + \ln q^{\setminus i}(\boldsymbol{\theta}\_m) - \ln q^{\setminus i}(\boldsymbol{\theta}\_m) \\\\
> &= \mathbb{E}\_{q(\boldsymbol{\theta}\_{\setminus m})}[\ln f\_{i}(\boldsymbol{\theta})q^{\setminus i}(\boldsymbol{\theta}\_m)] - \ln q^{\setminus i}(\boldsymbol{\theta}\_m) \\\\
> &= \mathbb{E}\_{q(\boldsymbol{\theta}\_{\setminus m})}[\ln \hat{p}\_i(\boldsymbol{\theta}\_{m}|\boldsymbol{\theta}\_{\setminus m})] - \ln q^{\setminus i}(\boldsymbol{\theta}\_m). \tag{3}
> \end{align}
> $$
>
> Comparing (2) and (3), we see that the equation (1) holds if and only if
>
> $$
> \mathbb{E}\_{q(\boldsymbol{\theta}\_{\setminus m})}[\ln \hat{p}\_i(\boldsymbol{\theta}\_{m}|\boldsymbol{\theta}\_{\setminus m})] = \ln \hat{p}\_i(\boldsymbol{\theta}\_{m}|\mathbb{E}\_q[\boldsymbol{\theta}\_{\setminus m}]).
> $$
> Now, we show that this equality holds under the conditions in Theorem 1. Since the model is conjugate-exponential, the conditional distribution $\hat{p}\_i(\boldsymbol{\theta}\_{m}|\boldsymbol{\theta}\_{\setminus m})$ is in the exponential family and can be expressed as:
>
> $$
> \begin{aligned}
> \hat{p}\_i(\boldsymbol{\theta}\_{m}|\boldsymbol{\theta}\_{\setminus m}) = h(\boldsymbol{\theta}\_m) \exp\left\\{ \boldsymbol{\eta}\_m(\boldsymbol{\theta}\_{\setminus m})^{T} \boldsymbol{\phi}(\boldsymbol{\theta}\_m) - Z\_m(\boldsymbol{\eta}\_m(\boldsymbol{\theta}\_{\setminus m})) \right\\}.
> \end{aligned}
> $$
>
> where $\boldsymbol{\phi}(\boldsymbol{\theta}\_m)$ is the vector of sufficient statistics; $\boldsymbol{\eta}\_m$ are the natural parameters; and $Z\_m(\cdot)$ is the log partition function. Its logarithm can be expressed by:
>
> $$
> \begin{align}
> \ln \hat{p}\_i(\boldsymbol{\theta}\_{m}|\boldsymbol{\theta}\_{\setminus m}) = \ln h(\boldsymbol{\theta}\_m) + \boldsymbol{\eta}\_m(\boldsymbol{\theta}\_{\setminus m})^{T} \boldsymbol{\phi}(\boldsymbol{\theta}\_m) - Z\_m(\boldsymbol{\eta}\_m(\boldsymbol{\theta}\_{\setminus m})).
> \end{align}
> $$
>
> Taking expectation with respect to $q(\boldsymbol{\theta}\_{\setminus m})$ yields:
> $$
> \begin{aligned}
> \mathbb{E}\_{q(\boldsymbol{\theta}\_{\setminus m})}[\ln \hat{p}\_i(\boldsymbol{\theta}\_{m}|\boldsymbol{\theta}\_{\setminus m})] = \ln h(\boldsymbol{\theta}\_m) + \mathbb{E}\_{q(\boldsymbol{\theta}\_{\setminus m})}[\boldsymbol{\eta}\_m(\boldsymbol{\theta}\_{\setminus m})]^{T} \boldsymbol{\phi}(\boldsymbol{\theta}\_m)  + \text{const}.
> \end{aligned}
> $$

---

> > ### Author Response · Authors · 2025-07-15
> >
> > For a conjugate-exponential model (see the original paper of VMP), $\ln \hat{p}\_i(\boldsymbol{\theta}\_{m}|\boldsymbol{\theta}\_{\setminus m})$ is a *multi-linear* function of the natural statistic functions of $\boldsymbol{\theta}\_{m}$ and the variables in $\boldsymbol{\theta}\_{m}$ (i.e., $\phi(\boldsymbol{\theta}\_{j}), \forall j=1,\cdots,M$).
> >
> > As a result, $\boldsymbol{\eta}\_m(\boldsymbol{\theta}\_{\setminus m})$ is a linear function with the statistic parameters of $\boldsymbol{\theta}\_{\setminus m}$ and we have
> >
> > $$
> > \begin{align}
> > \mathbb{E}\_{q(\boldsymbol{\theta}\_{\setminus m})}[\ln \hat{p}\_i(\boldsymbol{\theta}\_{m}|\boldsymbol{\theta}\_{\setminus m})] &= \ln h(\boldsymbol{\theta}\_m) + \mathbb{E}\_{q(\boldsymbol{\theta}\_{\setminus m})}[\boldsymbol{\eta}\_m(\boldsymbol{\theta}\_{\setminus m})]^{T} \boldsymbol{\phi}(\boldsymbol{\theta}\_m) + \text{const} \\\\
> > &= \ln h(\boldsymbol{\theta}\_m) + \boldsymbol{\eta}\_m(\mathbb{E}\_{q}[\boldsymbol{\theta}\_{\setminus m}])^{T}\boldsymbol{\phi}(\boldsymbol{\theta}\_m) + \text{const} \\\\
> > &= \ln \hat{p}\_i(\boldsymbol{\theta}\_{m}|\mathbb{E}\_q[\boldsymbol{\theta}\_{\setminus m}]). \tag{4}
> > \end{align}
> > $$
> >
> > Thus, equation (1) holds, and by Lemma 5, we conclude that the updates of CEP and VMP are equivalent under the specified conditions.
> >
> > Note that equation (4) can be seen as a reparameterization process, where the statistic parameters of $\boldsymbol{\theta}\_{\setminus m}$ are replaced with their expectations regarding the corresponding distribution. This procedure is thoroughly discussed in the original VMP paper (see Eqs. (17)-(18)), along with a simple illustrative example.
> >
> > To summarize, VMP does not require any approximation, and all results that follow later in the paper (e.g., proving convergence of CEP) are fully justified. We have revised the proof in the manuscript for greater clarity and included a discussion of the reparameterization process to avoid further confusion.

---

> > > ### Comment · Reviewer_X5pz · 2025-07-29
> > >
> > > For conjugate exponential family models, $\\ln \\hat{p}\_i(\\boldsymbol{\\theta}\_{m}|\\boldsymbol{\\theta}\_{\\setminus m})$ is a multi-linear function of the statistic functions $\phi(\\boldsymbol{\\theta}\_{i})\ \forall\ i=1, \ldots, m$, but *not* of $\theta_i$, directly.  Winn and Bishop (2004) are quite explicit that the assumption is about multi-linearity in the statistic functions, and not in the variables themselves. Pick your favourite conjugate exponential family model and you will see that this is not typically true.
> > >
> > > The second equality in (4) above is therefore not correct.  That equality would only be true in the the specific case that $\eta\_m(\theta\_{\setminus m})$ is linear in $\theta\_{\setminus m}$, which is not generally the case for conjugate exponential family models.  To assume the second equality in the general case then would require making a further approximation, in which case you would no longer be proving a claim about VMP.
> > >
> > > I'm happy to be convinced otherwise but I don't believe the misunderstanding here lies on my side.

---

> > > > ### Author Response · Authors · 2025-08-02
> > > >
> > > > Thank you for pointing this out. We agree with you that the second equality in equation (4) holds only under the specific condition that $\boldsymbol{\eta}\_m(\boldsymbol{\theta}\_{\setminus m})$ is a *linear* function of the variables $\boldsymbol{\theta}\_{\setminus m}$. In the general case, where $\boldsymbol{\eta}_m(\boldsymbol{\theta}\_{\setminus m})$ is a *non-linear* function, there exists a Jensen gap [1] between $\mathbb{E}\_{q(\boldsymbol{\theta}\_{\setminus m})}[\boldsymbol{\eta}\_m(\boldsymbol{\theta}\_{\setminus m})]$ and $\boldsymbol{\eta}\_m(\mathbb{E}\_{q}[\boldsymbol{\theta}\_{\setminus m}])$. Therefore, the second equality in equation (4) does not generally hold for conjugate-exponential family models.
> > > >
> > > > We acknowledge the imprecise formulation in the original version and have corrected this issue in the revised manuscript. We also emphasize that *although some expressions have been revised for clarity, all theoretical results in the paper remain valid, as they were derived based on the correct underlying interpretation.* The following provides a more detailed explanation.
> > > >
> > > > In a conjugate-exponential family model, the logarithm of the conditional distribution $\hat{p}\_i(\boldsymbol{\theta}\_{m}|\boldsymbol{\theta}\_{\setminus m})$ can be written as
> > > >
> > > > $$
> > > > \begin{align}
> > > > 	\ln \hat{p}\_i(\boldsymbol{\theta}\_{m}|\boldsymbol{\theta}\_{\setminus m})
> > > > 	= \ln h(\boldsymbol{\theta}\_m) + \boldsymbol{\eta}\_m(\boldsymbol{\theta}\_{\setminus m})^{T} \boldsymbol{\phi}(\boldsymbol{\theta}\_m)
> > > > 	- Z\_m(\boldsymbol{\eta}\_m(\boldsymbol{\theta}\_{\setminus m})),
> > > > \end{align}
> > > > $$
> > > >
> > > > which is a multi-linear function of the sufficient statistics $\boldsymbol{\phi}(\boldsymbol{\theta}\_j)$ for $j = 1,\dots,M$. This implies that the natural parameter function $\boldsymbol{\eta}\_m(\boldsymbol{\theta}\_{\setminus m})$ must itself be a multi-linear function of the sufficient statistics $\boldsymbol{\phi}(\boldsymbol{\theta}\_j)$ for all $j \neq m$. In the previous version, $\boldsymbol{\eta}\_m(\boldsymbol{\theta}\_{\setminus m})$ was used as a shorthand for $\boldsymbol{\eta}\_m(\boldsymbol{\phi}(\boldsymbol{\theta}\_{\setminus m}))$, where $\boldsymbol{\phi}(\boldsymbol{\theta}\_{\setminus m}) = \\{\boldsymbol{\phi}(\boldsymbol{\theta}\_j)\\}\_{j\neq m}$ represents the collection of sufficient statistics for all variables except $\boldsymbol{\theta}\_m$. However, this simplification can be misleading, as it obscures the actual dependency of the natural parameter on the sufficient statistics. To avoid confusion, we have adopted the more precise notation $\boldsymbol{\eta}\_m(\boldsymbol{\phi}(\boldsymbol{\theta}\_{\setminus m}))$ in the revised manuscript.
> > > >
> > > > Since $\boldsymbol{\eta}\_m(\boldsymbol{\phi}(\boldsymbol{\theta}\_{\setminus m}))$ is a multi-linear function of $\boldsymbol{\phi}(\boldsymbol{\theta}\_{\setminus m})$ and variational distribution $q(\boldsymbol{\theta}\_{\setminus m})$ factorizes with different variable groups (mean-field assumption), its expectation with respect to $q(\boldsymbol{\theta}\_{\setminus m})$ can be written as
> > > >
> > > > $$
> > > > \begin{align}
> > > > 	\mathbb{E}\_{q(\boldsymbol{\theta}\_{\setminus m})}[\boldsymbol{\eta}\_m(\boldsymbol{\phi}(\boldsymbol{\theta}\_{\setminus m}))] = \boldsymbol{\eta}\_m(\mathbb{E}\_{q(\boldsymbol{\theta}\_{\setminus m})}[\boldsymbol{\phi}(\boldsymbol{\theta}\_{\setminus m})]).
> > > > \end{align}
> > > > $$
> > > >
> > > > With this clarification, equation (4) can be reformulated as
> > > >
> > > > $$
> > > > \begin{aligned}
> > > > 	\mathbb{E}\_{q(\boldsymbol{\theta}\_{\setminus m})}[\ln \hat{p}\_i(\boldsymbol{\theta}\_{m}|\boldsymbol{\theta}\_{\setminus m})]
> > > > 	&= \ln h(\boldsymbol{\theta}\_m) + \mathbb{E}\_{q(\boldsymbol{\theta}\_{\setminus m})}[\boldsymbol{\eta}\_m(\boldsymbol{\phi}(\boldsymbol{\theta}\_{\setminus m}))]^{T} \boldsymbol{\phi}(\boldsymbol{\theta}\_m) + \text{const} \\\\
> > > > 	&= \ln h(\boldsymbol{\theta}\_m) + \boldsymbol{\eta}\_m(\mathbb{E}\_{q}[\boldsymbol{\phi}(\boldsymbol{\theta}\_{\setminus m})])^{T} \boldsymbol{\phi}(\boldsymbol{\theta}\_m) + \text{const} \\\\
> > > > 	&= \ln \hat{p}\_i(\boldsymbol{\theta}\_{m}|\mathbb{E}\_q[\boldsymbol{\phi}(\boldsymbol{\theta}\_{\setminus m})]).
> > > > \end{aligned}
> > > > $$
> > > >
> > > > This reformulated expression exactly corresponds to the "reparameterisation process" described in the original VMP paper (see equation (17) in [Winn and Bishop]), where the sufficient statistics $\boldsymbol{\phi}(\boldsymbol{\theta}\_{\setminus m})$ in $\boldsymbol{\eta}\_m(\boldsymbol{\phi}(\boldsymbol{\theta}\_{\setminus m}))$ are replaced by their expectations under the corresponding variational distributions.

---

> > > > ### Author Response · Authors · 2025-08-02
> > > >
> > > > Besides, in the original CEP paper, the delta approximation or Taylor expansion is also applied to a function of the sufficient statistics, denoted as $h(\boldsymbol{\phi}(\boldsymbol{\theta}\_{\setminus m}))$ (see equation (10) in the original CEP paper or the last equation in Appendix C).
> > > > Using the revised notation, the $i$th factor $\tilde{f}\_{i}(\boldsymbol{\theta}\_m)$ can be expressed as
> > > > $$
> > > > \begin{aligned}
> > > > 	\tilde{f}\_{i}(\boldsymbol{\theta}\_m) \propto \frac{\hat{p}\_i(\boldsymbol{\theta}\_{m}|\mathbb{E}\_q[\boldsymbol{\phi}(\boldsymbol{\theta}\_{\setminus m})])}{q^{\setminus i}(\boldsymbol{\theta}\_m)}.
> > > > \end{aligned}
> > > > $$
> > > > This shows that both VMP and CEP involve taking expectations with respect to the sufficient statistics while not variables, and therefore their conceptual connection remains valid under the revised and more precise notation.
> > > >
> > > > Additionally, the derivations of the various algorithms in the context of Bayesian tensor decomposition remain valid, as they only involve a notational revision.
> > > > For example, in the VMP derivation of Bayesian tensor decomposition, the embedding vector $\boldsymbol{u}\_s^k$ follows a Gaussian distribution $q(\boldsymbol{u}\_{s}^k) = \mathcal{N}(\boldsymbol{u}\_{s}^k|\boldsymbol{\mu}\_s^k, \boldsymbol{\Sigma}\_{s}^{k})$, and its logarithm is given by (see equation (46) in Appendix E)
> > > > $$
> > > > \begin{aligned}
> > > >  	\ln q(\boldsymbol{u}\_s^k|\mathcal{U}\_{\setminus \boldsymbol{u}\_s^k}, \tau)
> > > >  	&= (\boldsymbol{\Sigma}\_{s}^{k})^{-1} \boldsymbol{\mu}\_s^k \cdot \boldsymbol{u}\_s^k -\frac{1}{2}(\boldsymbol{u}\_s^k)^T (\boldsymbol{\Sigma}\_{s}^{k})^{-1} \boldsymbol{u}\_s^k + \text{const} \\\\
> > > >  	&= \boldsymbol{\eta}(\mathcal{U}\_{\setminus \boldsymbol{u}\_s^k}, \tau)^T \boldsymbol{\phi}(\boldsymbol{u}\_s^k) + \text{const},
> > > > \end{aligned}
> > > > $$
> > > > where $\boldsymbol{\eta}(\mathcal{U}\_{\setminus \boldsymbol{u}\_s^k}, \tau) = [(\boldsymbol{\Sigma}\_{s}^{k})^{-1} \boldsymbol{\mu}\_s^k,\ -\frac{1}{2}\text{vec}((\boldsymbol{\Sigma}\_{s}^{k})^{-1})]^T$ and $\boldsymbol{\phi}(\boldsymbol{u}\_s^k) = [\boldsymbol{u}\_s^k,\ \text{vec}(\boldsymbol{u}\_s^k(\boldsymbol{u}\_s^k)^T)]^T$ are the natural parameter and sufficient statistic of the multivariate Gaussian distribution [2].
> > > > In our derivation, we take the expectation of the natural parameter function with respect to the sufficient statistics $\boldsymbol{\phi}(\mathcal{U}\_{\setminus \boldsymbol{u}\_s^k}, \tau)$, rather than the random variables themselves, and this leads to the results shown in Section 4.2. This is fully consistent with the corrected formulation discussed above. Therefore, the experimental results in the main paper also remain valid.
> > > >
> > > > To summarize, we have carefully proofread the manuscript, revised the relevant notation, and added further explanation to improve clarity and avoid confusion. Thank you again for your insightful comment. Please let us know if you have any additional concerns.
> > > >
> > > > [1] Gao, X., Sitharam, M. and Roitberg, A.E., 2017. Bounds on the jensen gap, and implications for mean-concentrated distributions. arXiv preprint arXiv:1712.05267.
> > > >
> > > > [2] Murphy, K.P., 2012. Machine learning: a probabilistic perspective. MIT press.

---

> > > > > ### Comment · Reviewer_X5pz · 2025-08-02
> > > > >
> > > > > I'm happy that with that change the VMP updates are now correctly stated in the paper.  However, as you mentioned, the update for CEP has now also changed between manuscript versions:
> > > > >
> > > > > Old version:
> > > > > $$\begin{aligned} \tilde{f}\_{i}(\boldsymbol{\theta}\_m) \propto \frac{\hat{p}\_i(\boldsymbol{\theta}\_{m}|\mathbb{E}\_q[\boldsymbol{\theta}\_{\setminus m}])}{q^{\setminus i}(\boldsymbol{\theta}\_m)}. \end{aligned}$$
> > > > >
> > > > > New version:
> > > > > $$\begin{aligned} \tilde{f}\_{i}(\boldsymbol{\theta}\_m) \propto \frac{\hat{p}\_i(\boldsymbol{\theta}\_{m}|\mathbb{E}\_q[\boldsymbol{\phi}(\boldsymbol{\theta}\_{\setminus m})])}{q^{\setminus i}(\boldsymbol{\theta}\_m)}. \end{aligned}$$
> > > > >
> > > > > These are fundamentally different quantities, and so presumably at most one of these can be correctly interpreted as the CEP update - it seems a stretch to put the difference down to notational imprecision.  Without investing significantly more time it would be difficult for me to be say with a lot of confidence that the latter of these corresponds to CEP as proposed in Wang (2019), but I'm willing to give the benefit of the doubt from here that this is indeed the case.
> > > > >
> > > > > I believe that showing equivalence of CEP and VMP under the stated conditions is novel, and would be of interest to people. My reservations remain that many of the claims made in the paper are unjustified, in my view. To summarise some of the most significant ones - from the abstract:
> > > > >
> > > > > _"we prove the convergence of CEP"_ - the theoretical result only guarantees convergence for a particular class of model, for a particular approximation, and when updates are applied serially (VMP updates are not guaranteed to converge when applied in parallel).
> > > > >
> > > > > Also from the abstract: _" and enable an online variant of VMP through this connection"_. As I discussed in an earlier comment, online VMP has been studied extensively before, and it seems almost completely unrelated to the theroetical result.  The response that the paper's perspective is different because it considers a less general setting than previous work is not a convincing argument for stating this as one of the main contributions
> > > > >
> > > > > From the conclusion:
> > > > >
> > > > > _"Generally, VI and EP have different properties ... This work, for the first time, demonstrates that their variants are closely related under certain conditions, which sheds new light on the understanding and development of further advanced ABI methods."_ - this is a very grandiose claim which is completely unjustified. The main result of the paper shows that CEP under the stated conditions is not EP at all, but is in fact just VMP - it says nothing of the relationship between VI and **EP**.
> > > > >
> > > > > Similarly, in several places the paper claims that a connection has been made between ADF and streaming VI. e.g. _" Our results, particularly Corollary 2, provide a theoretical connection between ADF and streaming VI,"_ - but this is not true. A connection has been made between streaming VMP and **CEP** (when applied in an ADF-style fashion, under certain conditions, and for a particular class of models).
> > > > >
> > > > > There are several other claims in the paper that I believe are unjustified, but I have focused here on those that are stated as main contributions.  I believe the theoretical result alone would be of interest to some (assuming it is correct), but it could have been shown in a far shorter paper, without (for example) the long section on graphical models, the material on streaming VMP, or even any experiments (if the main result is that the updates of two methods are identical, then are experiments really necessary?).

---

> > > > > > ### Author Response · Authors · 2025-08-07
> > > > > >
> > > > > > Thank you for your insightful comments and for recognizing the novelty of establishing the equivalence between CEP and VMP under the stated conditions. Following your suggestion, we have revised the presentation and clarified the claims in the manuscript accordingly. Below is our point-by-point response.
> > > > > >
> > > > > > **C1.** *Without investing significantly more time it would be difficult for me to be say with a lot of confidence that the latter of these corresponds to CEP as proposed in Wang (2019), but I'm willing to give the benefit of the doubt from here that this is indeed the case.*
> > > > > >
> > > > > > **R1:**  We appreciate your willingness to consider the new formulation as correct. We agree that the two versions represent different quantities, and we have carefully proofread and revised the manuscript to ensure that the notations now accurately reflect the intended CEP update.
> > > > > >
> > > > > > To clarify which version is correct, we emphasize that the new formulation more accurately reflects the original definition of CEP. Specifically, in both the original CEP paper and our preliminary Section 2.2.2, expectations are consistently taken with respect to the sufficient statistics, not the variables themselves. See, for instance, equation (11) and the associated discussion in the main text.
> > > > > >
> > > > > > Additionally, the delta approximation is applied to function of the sufficient statistics, not directly to the variables. This is explicitly stated in equation (10) of the original CEP paper and reiterated in the last equation of our Appendix C. We believe the revised structure and textual explanations now make this point much clearer.
> > > > > >
> > > > > > This correction is further supported by the example in Section 4 of our manuscript. In that example, the $i$th factor $\tilde{f}\_{\mathbf{i}}^k(\mathbf{u}\_{i\_k}^k)$ for variable $\mathbf{u}\_{i\_k}^k$ follows a Gaussian distribution:
> > > > > >
> > > > > > $$
> > > > > > \tilde{f}\_{\mathbf{i}}^k(\mathbf{u}\_{i\_k}^k) \sim \mathcal{N}(\mathbf{u}\_{i\_k}^k \mid \mathbf{m}\_{\mathbf{i}}^k, \mathbf{S}\_{\mathbf{i}}^k),
> > > > > > $$
> > > > > > with mean and variance given by
> > > > > > $$
> > > > > > \begin{aligned}
> > > > > > 	\mathbf{S}\_{\mathbf{i}}^k &= \left( \langle \tau \rangle \langle \mathbf{z}\_{\mathbf{i}}^{\setminus k} \mathbf{z}\_{\mathbf{i}}^{\setminus k^T} \rangle \right)^{-1}, \\\\
> > > > > > 	\mathbf{m}\_{\mathbf{i}}^k &= \mathbf{S}\_{\mathbf{i}}^k \left( y\_{\mathbf{i}} \langle \tau \rangle \langle \mathbf{z}\_{\mathbf{i}}^{\setminus k} \rangle \right),
> > > > > > \end{aligned}
> > > > > > $$
> > > > > >
> > > > > > where $\mathbf{z}\_{\mathbf{i}}^{\setminus k} = \mathbf{u}\_{i\_1}^1 \circ \cdots \circ \mathbf{u}\_{i\_{k-1}}^{k-1} \circ \mathbf{u}\_{i\_{k+1}}^{k+1} \circ \cdots \circ \mathbf{u}\_{i\_K}^K$. Note that in the variance matrix we calculated the expectations with respect to the sufficient statistics, including the first moment $\langle \mathbf{z}\_{\mathbf{i}}^{\setminus k} \rangle$ and the second moment $\langle \mathbf{z}\_{\mathbf{i}}^{\setminus k} {\mathbf{z}\_{\mathbf{i}}^{\setminus k}}^T \rangle$. It can be easily seen that the new version gives a more accurate presentation. This example confirms that the new formulation provides a more accurate and internally consistent representation.
> > > > > >
> > > > > > We hope this addresses your concern and clarifies the rationale behind the revision.
> > > > > >
> > > > > > **C2.** *"we prove the convergence of CEP" - the theoretical result only guarantees convergence for a particular class of model, for a particular approximation, and when updates are applied serially (VMP updates are not guaranteed to converge when applied in parallel).*
> > > > > >
> > > > > > **R2:** Thank you for your instructive comment. Following your suggestion, we have revised the relevant statement to clarify that the convergence proof holds under certain conditions. We also would like to emphasize that in many practical applications, such as the tensor decomposition example discussed in the paper, these conditions are typically satisfied.
> > > > > >
> > > > > > First, the specific class of models considered in our analysis—the conjugate-exponential family—is widely used and covers a broad range of applications. Without this assumption, closed-form updates often do not exist, and significant computational effort is required to implement ABI methods.
> > > > > >
> > > > > > Second, the delta approximation is commonly adopted in CEP-related works (see [1-3]). In particular, the outer expectation in equation (10), which involves integration over variables except $\boldsymbol{\theta}\_m$, is generally intractable for models of practical interest. Therefore, we believe this approximation is necessary for many practical cases of real world applications.
> > > > > >
> > > > > > Third, the group-wise update strategy used in CEP is more practical than the entry-wise update. In large-scale settings, the number of data entries is typically much larger than the number of variable groups. As a result, group-wise updates are computationally more efficient and are widely adopted in practice.
> > > > > >
> > > > > > In the main text, we provide a discussion on the conditions required in Theorem 1. To improve clarity, we have revised the relevant sentences and added references accordingly.

---

> > > > > > ### Author Response · Authors · 2025-08-07
> > > > > >
> > > > > > **C3.** *Also from the abstract: "and enable an online variant of VMP through this connection". As I discussed in an earlier comment, online VMP has been studied extensively before, and it seems almost completely unrelated to the theoretical result. The response that the paper's perspective is different because it considers a less general setting than previous work is not a convincing argument for stating this as one of the main contributions.*
> > > > > >
> > > > > > **R3:**  Thank you for your comment. We agree that online or streaming variational Bayesian learning has been extensively studied, and it is indeed possible to develop such methods without relying on the connection established in our work.
> > > > > >
> > > > > > However, we would like to emphasize that our results differ from existing approaches. Specifically, we consider the conjugate-exponential model with mean-field assumption, which is a general class of model and finds wide applications in different domains. Additionally, previous works typically develop VMP and its online or streaming variants independently. Given a model, one can derive a streaming VMP algorithm using a general framework that treats the posterior from earlier mini-batches as the prior for new data. While this is a valid and widely adopted approach, it does not establish a direct algorithmic or theoretical connection between the offline and online versions of VMP.
> > > > > >
> > > > > > In contrast, our work focuses on precisely this connection. It is known that ADF is an online Bayesian inference method, and EP can be viewed as its offline counterpart. Due to the local nature of EP updates, an EP algorithm can often be directly adapted to a streaming setting. Our goal is to investigate whether a similar transformation exists between VMP and its streaming variant.
> > > > > >
> > > > > > By establishing a connection between VMP and CEP, we show that such a relationship does exist. Moreover, we demonstrate that the update rules of the streaming versions of CEP and VMP are equivalent under certain conditions. This equivalence follows from, and is directly linked to, the theoretical results developed in the paper.
> > > > > >
> > > > > > **C4.** *From the conclusion:  "Generally, VI and EP have different properties ... This work, for the first time, demonstrates that their variants are closely related under certain conditions, which sheds new light on the understanding and development of further advanced ABI methods." - this is a very grandiose claim which is completely unjustified. The main result of the paper shows that CEP under the stated conditions is not EP at all, but is in fact just VMP - it says nothing of the relationship between VI and EP.*
> > > > > >
> > > > > > **R4:**  Thank you for your insightful comment. You are correct that our main result establishes a connection between CEP and VMP under certain conditions, and does not directly address the relationship between EP and VI. The original wording in the conclusion may be misleading in this regard.
> > > > > >
> > > > > > To avoid this confusion, we have revised the conclusion to clarify that our contribution lies in revealing the connection between VMP and CEP, while not VI and EP.

---

> > > > > > ### Author Response · Authors · 2025-08-07
> > > > > >
> > > > > > **C5.** *Similarly, in several places the paper claims that a connection has been made between ADF and streaming VI. e.g. "Our results, particularly Corollary 2, provide a theoretical connection between ADF and streaming VI," - but this is not true. A connection has been made between streaming VMP and CEP (when applied in an ADF-style fashion, under certain conditions, and for a particular class of models).*
> > > > > >
> > > > > > **R5:**  Thank you for your comment. You are right that the precise connection established in our work is between streaming VMP and CEP, not directly between ADF and streaming VI.
> > > > > >
> > > > > > We would like to clarify that VMP is a specific implementation of variational inference for conjugate-exponential models. Under this assumption, VMP and VI yield equivalent updates. Therefore, in the context of conjugate-exponential models, streaming VMP is equivalent to streaming VI.
> > > > > >
> > > > > > In Corollary 2, we focus on ADF rather than streaming CEP because ADF serves as a more general framework, while CEP can be viewed as a specific approximation strategy. In this sense, instead of describing our result as "implementing CEP in an ADF-style fashion," it is more accurate to say that our analysis applies the CEP approximation within the ADF framework. This interpretation is also supported by prior work [1, 4, 5], which shows that CEP approximation can be used within ADF to handle broader classes of models.
> > > > > >
> > > > > > However, we agree that the original presentation may have caused confusion, and we have revised the text to clarify this relationship more precisely and accurately.
> > > > > >
> > > > > > **C6.** *There are several other claims in the paper that I believe are unjustified, but I have focused here on those that are stated as main contributions. I believe the theoretical result alone would be of interest to some (assuming it is correct), but it could have been shown in a far shorter paper, without (for example) the long section on graphical models, the material on streaming VMP, or even any experiments (if the main result is that the updates of two methods are identical, then are experiments really necessary?).*
> > > > > >
> > > > > > **R6:** Thank you for your feedback. We understand your concern regarding the length and scope of the manuscript, and we appreciate your acknowledgment that the theoretical result could be of interest on its own.
> > > > > >
> > > > > > We agree that the main theoretical result is self-contained and could, in principle, be presented in a more concise form. However, our goal was not only to prove the equivalence between the two methods under specific conditions but also to provide the necessary context and interpretations to make the result accessible and applicable to a broader audience. For example, the section on graphical models offers a helpful interpretation of VMP and CEP in terms of message passing, which is valuable for readers who may not be familiar with this perspective. Similarly, the section on streaming VMP illustrates a direct application of our main theorem and shows how it can be used in practice.
> > > > > >
> > > > > > While experts in the area may find some of this material familiar, we believe it adds pedagogical value for readers who are less experienced with different ABI methods. Moreover, we followed the suggestions of other reviewers (e.g., **Reviewer qKMR** and **Reviewer TSkv**) to restructure the manuscript, improve clarity, and include empirical results.
> > > > > >
> > > > > > Although our theoretical results guarantee the equivalence of the update rules under certain assumptions, we believe that experiments are still meaningful. They serve not only to confirm the equivalence in practical implementations but also to explore the behavior of the algorithms when the assumptions are relaxed. These empirical findings help illustrate the robustness and practical relevance of the theoretical result.
> > > > > >
> > > > > > We have revised the manuscript accordingly and hope the changes improve the clarity and usefulness of the presentation.

---

> ### Author Response · Authors · 2025-07-15
>
> **C3**. *The authors also claim that this perspective allows them to construct 'a streaming version of VMP'. It is not clear to me that the extension of VMP to streaming data is at all related to the earlier theoretical results, but in any case, streaming or online VMP has been explored extensively, see e.g. [2,3,4,5.6].*
>
> **R3:**  Thank you for pointing this out. We acknowledge that extensive prior work has explored streaming or online variational Bayesian learning within a general framework. Our proposed approach is indeed closely related to these works, as they share the same foundational principle: treating the posterior obtained from previous mini-batches as the new prior for incoming data points.
>
> However, the specific settings and methodologies differ across approaches, and we outline these distinctions below:
>
> 1. Existing works typically focus on one direction of KL divergence for posterior approximation. For example, minimizing $\text{KL}(q \|\| p)$ leads to streaming versions of VI-class methods. Minimizing $\text{KL}(p \|\| q)$ results in streaming versions of EP-class methods, such as the ADF algorithm. In contrast, our work highlights the connection between these two cases. This perspective not only provides insight into their relationship but also offers practical guidance for algorithm development.
>
> 2. Unlike many existing methods, which assume a general prior distribution, our approach considers a specific prior structure. In our model, the prior is a factorizable distribution, with each subfactor belonging to the exponential family. This setting has not been explicitly addressed in the existing literature and is key to extending VMP in our framework.
>
> 3. The streaming version of VMP in our work is not presented as a novel algorithm in itself, but rather as a natural byproduct of the main theoretical results. Specifically, our claim is that the VMP algorithm developed for existing models can be seamlessly extended to a streaming or online setting under certain conditions. Furthermore, we show that the streaming VMP approach is closely related to the ADF algorithm under specific scenarios.
>
> **C4**. *It is not clear to me what Section 3.3 "Interpretation via Graphical Models" is contributing -- it seems to just introduce a lot of new notation without adding anything meaningful to the paper.*
>
> **R4:**  We thank the reviewer for this comment. The goal of Section 3.3 is to offer an alternative interpretation of our main theoretical results using the framework of graphical models. We believe this perspective helps clarify the underlying structure of the algorithms and can assist readers in understanding the similarities and differences between VMP and CEP.
>
> Our work analyzes the connection between Variational Message Passing (VMP) and Conditional Expectation Propagation (CEP), both of which are inherently linked to graphical model representations:
>
> - VMP was originally developed and analyzed within the context of Bayesian networks, where the update rules are naturally interpreted as message passing between nodes.
> - CEP, as a variant of Expectation Propagation (EP), is closely related to factor graphs and also relies on local updates governed by the graph's conditional dependencies.
>
> As the reviewer correctly pointed out, introducing graphical model terminology can add additional notation such as "nodes," "messages," and "factors." To address this and avoid confusion, we have added a concise introduction to graphical model concepts in Section II of the revised manuscript. We hope this provides sufficient background and helps readers follow the interpretation provided in Section 3.3.

---

### Review · Reviewer_qKMR · 2025-06-02

**Summary Of Contributions:**

This paper investigates the relationships between variational message passing (VMP) and conditional expectation propagation (CEP), two widely-used approximate Bayesian inference (ABI) methods. The key contributions are: (1) establish conditions under which VMP and CEP are defined by equivalent update equations, (2) provide convergence guarantees for CEP by leveraging its connection to VMP, (3) describe their applications to sequential data settings and Bayesian tensor decomposition

**Audience:**

Yes

**Broader Impact Concerns:**

I have no Broader Impact concerns.

**Claims And Evidence:**

No

**Requested Changes:**

1) I think the organisation and presentation of the paper should be significantly improved for the paper to be accepted for publication. I provide specific suggestions below.

2) I don’t think related works are discussed adequately in this paper. It is important to explain what new results this paper establishes in the context of existing results. This can be done nicely while describing Fig. 1.

3) Section 4: Would it be possible to include numerical results for the tensor decomposition application? Without empirical validation, the mathematical derivations provide limited practical insights.
---
Suggestions regarding 1):
- Fig. 1: some acronyms in the figure aren’t defined before Fig. 1. It might be better to clarify all acronyms in the caption. This figure effectively summarizes the paper's contribution and would serve well as an opening illustration. However, to maximise its utility in the Introduction, either the figure should be simplified to show only the elements explained early on, or the Introduction should provide brief explanations for all illustrated connections. The figure does not complement the text well right now.

- p.2 second last paragraph: “…which refine the factor that represents the **contribution** of the posterior from each data point.”—> “the factor node that represents how each data point contributes to the posterior distribution”

- what is “clutter problem”?

- There are some uncommon and slightly awkward phrasings in this paper, consider replacing them with something more direct, and consider replacing nouns with verbs which are often more powerful. e.g. p.2 “…due to its local optimization nature”—>”as message passing updates are local” p.3 “cross-pollination of their respective strengths”—>”allows one to combine their respective strengths.”. p.3 “ensures the attainment of convergence.”—> ensures convergence. “adaptability”—>”adaptivity”

- p.4: when you introduce the M disjoint groups, can you add a sentence or two to explain why one would do this? I guess you want to say the probability distribution q(\theta) is assumed to factorize into M groups.

- the “+const” terms in many equations on p.4 are confusing because they represent different constants in different equations. Consider using c1, c2, … for different constants.

- p.5: “equation 4”—>(4)

- Given the large number of acronyms in this paper, I would recommend keeping the full name of each method alongside its acronyms in section titles e.g. variational message passing (VMP) for Section 2.1.2.

- h(\theta_m) is undefined in (5), I guess this denotes the prior.

- The explanation in multiple places of the paper is vague and unclear. For instance, Section 2.1.2, without defining or describing the underlying graphical model, the “nodes” terminology is used directly. It is unclear what messages are passed between nodes. Also I think the variable nodes and factor nodes should be distinguished instead of being both referred to as nodes. The terminology introduced here overlaps significantly with what's needed in Section 2.2.1. Consider introducing these terms clearly in this section and minimising additional new terminology later to improve consistency and readability.

- Section 2.2.1: the range of values that i takes is missing from several equations and should be added, e.g, the last euation on p.5. Each datapoint xi is boldfaced here but isn’t in the beginning of Section 2. Make notation consistent.

- Avoid vague description like “the ith likelihood term p(xi|\theta)”. I think you want to say the likelihood of the ith datapoint, or paragraph above eqn (8): “It operates by cycling through the factors and refining them one at a time.” or “local refinement” what do these mean? These descriptions are a bit too abstract to be helpful. Consider adding specific examples, equations, or algorithmic steps to make the methodology clearer to readers. It is hard for me to appreciate the distinctions and connections between the methods in Section 2.1 and those in Section 2.2.

- Formatting issue in last paragraph of Section 2.2.

- Section 2.4 feels out of place. Since the KL divergence and moment matching in variational inference with exponential families are concepts that the first part of Section 2 heavily relies on, I think the conceptual version of these lemmas should be presented when KL and moment matching are first introduced and discussed, maybe even alongside these lemmas themselves.

- Section 3.3: I think a large fraction of this section that explains the graphical model formulation VMP and CEP should be refactored and presented earlier on when these methods are first introduced.

- Section 3.4 and Fig. 3: this section is essential for synthesising the theoretical contributions but is poorly presented. The figure is confusing rather than illuminating, and the text lacks clear organisation. A cleaner presentation would significantly improve the paper's accessibility.

**Strengths And Weaknesses:**

The paper is quite comprehensive: it describes various ABI methods, including their extensions for handling sequential data. As a practical example, the paper details the mathematical forms that VMP and CEP take respectively when applied to tensor decomposition. The theoretical results seem largely valid, with their proofs based on existing results. Corollary 1 seems to be a useful result showing the convergence conditions of CEP.

This paper reads more like a technical report cataloging relationships between existing methods rather than a paper with interesting new insights. I had a hard time parsing the writing because it feels like a compilation of many scattered technical observations. For instance, I understand that Fig. 1 and Fig. 3 try to show connections but they become cluttered rather than illuminating. The benefits of establishing the connections between VMP and CEP should be presented more clearly. Also, would it be possible to include some numerical results for the tensor decomposition application?

---

> ### Author Response · Authors · 2025-07-15
>
> We sincerely thank the reviewer for the careful reading and constructive feedback. We have made revisions to the manuscript based on your suggestions, highlighting the changes in blue, which have significantly improved its readability and clarity. Below is our detailed response.
>
> **C1**. *I think the organisation and presentation of the paper should be significantly improved for the paper to be accepted for publication. I provide specific suggestions below.*
>
> **R1:** We sincerely thank the reviewer for the constructive suggestions regarding the organisation and presentation of our manuscript. In response, we have thoroughly revised the paper to enhance its clarity, consistency, and readability. The key changes are summarised as follows:
>
> (1) **Revision of Figures and Associated Descriptions**: We have revised Figure 1 by expanding all acronyms and adding explanations for several arrows in the caption. Additional clarifications have also been included in the Introduction to better explain the relationships illustrated in the figure. These changes ensure that all visual connections are explicitly discussed in the text or caption, thereby improving consistency and enhancing the figure's role as an overview.
>
> In response to the reviewer's concern about clarity, we simplified Figure 3 to focus on the connections among the key theoretical results and provided more detailed explanations in the main text. These revisions improve both readability and interpretability.
>
> Furthermore, we have added two illustrative figures of graphical models---specifically, a Bayesian network and a factor graph (see the revised Section II)---to support readers' understanding of the underlying concepts.
>
> (2) **Structural Reorganisation and Clarification of Key Concepts**: We have restructured Section 3.3 (graphical model interpretation) and moved the core background material to earlier in Section 2, incorporating additional explanations and figures to enhance clarity. This change improves consistency and accessibility for readers less familiar with graphical models.
>
> Additionally, we linked the moment matching and KL divergence concepts introduced in the early part of Section 2 to the two key lemmas discussed later in the section, improving both consistency and clarity. Relevant citations have also been included to strengthen the presentation.
>
> To further clarify the relationships between the methods, we added a new subsection in Section 2 that discusses the connections and distinctions between variational inference (VI) and expectation propagation (EP)-based methods, linking back to Figure 1. This subsection serves as a bridge to the subsequent content and improves the overall flow of the paper.
>
> We also revised Section 3.4 to enhance the paper's accessibility by providing additional explanations of Figure 3 (now Figure 4 in the revised manuscript) and rephrasing the practical suggestions. The updated Section 3.4 and revised Figure 3 now more clearly synthesize our theoretical contributions.
>
> (3) **Language Improvements and Terminology Clarification**: We revised multiple phrases and word choices to make the text more concise and natural, as suggested. Examples include: adaptability -> adaptivity, contribution of the posterior -> contributes to the posterior, local optimization nature -> updates are local, etc. We also replaced ambiguous instances of ''+const'' in equations with distinct constants such as $constant1$ and $constant2$ to reduce confusion. Section titles now include both acronyms and their full forms (e.g., ''Variational message passing (VMP)'') to aid comprehension. We also defined previously missing terms such as $h(\theta)$ in equation (5), and clarified expressions like ''clutter problem,'' ''local refinement,'' and ''likelihood term,'' with additional context and references.
>
> We have also addressed several minor notational inconsistencies across the manuscript, which can be found in the updated PDF. We believe these revisions significantly improve the paper's clarity and accessibility, while also making the theoretical connections between different inference methods more transparent. We thank the reviewer again for the valuable suggestions.

---

> ### Author Response · Authors · 2025-07-15
>
> **C2**. *I do not think related works are discussed adequately in this paper. It is important to explain what new results this paper establishes in the context of existing results. This can be done nicely while describing Fig. 1.*
>
> **R2:** We thank the reviewer for pointing out the need to better situate our contributions within the context of existing work. As mentioned in our response to Comment C1, we have substantially revised both the structure and content of the manuscript to address this. Specifically: (1) We revised and expanded the explanations around Figures 1 and 3 to more clearly highlight the theoretical contributions of the paper and how they connect to prior methods. (2) To strengthen the contextualisation of our results, we added a new subsection in Section 2 that explicitly discusses the relationships and differences between existing approximate inference approaches and related methods. This subsection makes it easier for readers to see how our work distinguishes itself from prior approaches. (3) Furthermore, to support the theoretical findings, we introduced empirical results that demonstrate the practical value of our analysis (see response to Comment C3 below).
>
> We believe these additions make the relationship between our contributions and prior work much clearer and improve the overall coherence of the paper.
>
> **C3**. *Section 4: Would it be possible to include numerical results for the tensor decomposition application? Without empirical validation, the mathematical derivations provide limited practical insights.*
>
> **R3:** We appreciate the reviewer's suggestion regarding empirical validation. In response, we have added a new subsection to Section 4, presenting numerical results for the tensor decomposition application. For more details, please refer to Section 4.5 of the revised manuscript.

---

> > ### Comment · Reviewer_qKMR · 2025-07-24
> >
> > I thank the authors for taking my comments into account, improving the structure and clarity of the paper, and adding numerical experiments. I would recommend acceptance of the paper if the technical flaw that Reviewer X5pz pointed out is addressed too.

---

### Review · Reviewer_TSkv · 2025-07-20

**Summary Of Contributions:**

1. The paper establishes a strong theoretical link between VMP and CEP under mild assumptions, highlighting their shared goal of approximating conditional marginal distributions through message merging.

2. Leveraging the connection with VMP, the authors provide a convergence guarantee for CEP, which previously lacked formal justification under general conditions.

3.  The identified connection facilitates the construction of a streaming version of the VMP algorithm, extending its applicability to online or large-scale settings.

4. The work sheds light on the underlying relationships among various amortized Bayesian inference methods, particularly showing the equivalence in update rules between Assumed Density Filtering (ADF) and streaming Variational Inference (VI).

5. In the updated version, the equivalence of the update rules between VMP and CEP is validated through numerical experiments.

**Audience:**

Yes

**Broader Impact Concerns:**

N/A.

**Claims And Evidence:**

No

**Requested Changes:**

1. In light of Weakness (2), it would strengthen the paper to include examples where the theoretical conditions are not strictly satisfied, yet VMP and CEP still exhibit comparable performance. This would provide practical insight into the robustness of the equivalence beyond the idealized settings covered by Theorem 1.

2. Regarding Weakness (3), I recommend removing or relocating less informative figures (e.g., Figures 2–4) to the appendix. Similarly, the extensive discussion of preliminaries and existing methods could be significantly condensed. Theoretical results established in prior work, such as Lemmas 1 and 2, should also be moved to the appendix to improve the clarity and flow of the main text.

3. I would caution against offering strong practical recommendations for ABI methods based solely on theoretical conditions. For practitioners focused on applications, verifying such assumptions may be difficult, which could limit the usability and adoption of the proposed guidance.

**Strengths And Weaknesses:**

# Strengths:

1. The paper provides thorough background and context for related methods in the Preliminaries section (e.g., VI, VMP, EP, CEP, and ADF), along with a concise discussion of their connections and differences. These details make the paper accessible and reader-friendly, especially for those less familiar with the underlying techniques.

2. The theoretical analysis is clear and well-structured, particularly as it builds logically on the foundational Lemmas 1 and 2. Notably, the paper establishes the convergence of CEP under mild conditions by leveraging its equivalence with VMP—a novel contribution that, to the best of my knowledge, has not been previously explored.

3. While the main focus of the paper is to provide a theoretical understanding of the connection between VMP and CEP, the authors also support their analysis with additional numerical experiments for validation.

# Weakness:

1. The primary contribution of the paper—Theorem 1—focuses on establishing a theoretical connection between two existing methods, VMP and CEP. While this is valuable for conceptual understanding, the novelty is somewhat limited. Moreover, the theoretical analysis is constrained by specific assumptions, including the use of models within the conjugate-exponential family, which restricts the generality of the results.

2. The numerical experiments mainly demonstrate that VMP and CEP exhibit nearly identical performance under the conditions assumed in Theorem 1. However, it would be more insightful to investigate whether the performance remains comparable—and whether CEP continues to converge—when some of these conditions are violated. Such robustness analysis would strengthen the empirical support for the theory.

3. Certain parts of the paper, such as the detailed discussion of Lemmas 1 and 2, could be streamlined or moved to an appendix, as they may not warrant such prominence in the main text. Additionally, some figures (e.g., Figures 2–4) are not particularly informative and may not meet the standards expected in a published paper. Improving figure clarity and focusing on essential visualizations would enhance the overall presentation.

4. [Minor] Based on the convergence result for CEP, the authors recommend assessing whether the model satisfies the conjugate-exponential assumption with independent variable partitions as a first step. However, this type of assumption-checking is rarely performed in practice, as most methods rely on assumptions that are often taken for granted rather than explicitly validated. Requiring such validation may limit the practical usability of the proposed guidance. Moreover, since the stated conditions are sufficient (but not necessary), it is likely that the streaming ABI algorithm could still perform well even when these conditions are not strictly met.

---

> ### Author Response · Authors · 2025-07-25
>
> We thank the reviewer for carefully reading our paper and for acknowledging its contributions. We also appreciate the constructive comments, which have helped us improve the clarity and overall quality of the revised manuscript. Below, we provide point-by-point responses to each of the reviewer's comments.
>
> **C1.** *The primary contribution of the paper---Theorem 1---focuses on establishing a theoretical connection between two existing methods, VMP and CEP. While this is valuable for conceptual understanding, the novelty is somewhat limited. Moreover, the theoretical analysis is constrained by specific assumptions, including the use of models within the conjugate-exponential family, which restricts the generality of the results.*
>
> **R1:** Thank you for your comments. As clarified in the revised manuscript, VMP and CEP originate from two distinct lines of ABI methods: VMP is rooted in variational inference (VI), while CEP is based on expectation propagation (EP). These methods were developed from different theoretical perspectives and have distinct algorithmic properties. As a result, they have traditionally been treated as separate frameworks, with little attention given to their potential connections.
>
> In this work, we show for the first time that two advanced variants of VI and EP are closely related under mild conditions. As the reviewer noted, this theoretical connection has not been previously explored. We believe this contributes a novel insight that not only deepens the conceptual understanding of these methods but also provides a foundation for unifying or combining their strengths. This, in turn, could inspire the development of improved ABI algorithms that leverage a shared understanding of existing techniques.
>
> Regarding the assumptions in Theorem 1, we would like to clarify that some of the conditions---such as the use of conjugate-exponential models or the i.i.d. data assumption---are not imposed by our theorem itself, but rather inherited from the standard formulations of VMP and CEP. These can be viewed as preconditions dictated by the methods rather than restrictions introduced by our analysis. The specific assumptions required for our analysis are explicitly stated in the main theorem.
>
> It is also noteworthy that the conjugate-exponential family is widely used in practice due to its analytical tractability and the existence of closed-form updates. While the assumption may limit theoretical generality, it still covers a broad class of practical models. In non-conjugate settings, closed-form updates often do not exist, making it difficult to establish a theoretical connection between different ABI methods.
>
> Although the theoretical equivalence may not extend to all model classes, we agree that it is important to study the empirical behavior of different methods when the assumptions are relaxed or violated. Following your suggestion, we have included additional experiments in the revised manuscript that explore such cases. Details are provided in the response to Comment C2.

---

> ### Author Response · Authors · 2025-07-25
>
> **C2.** *The numerical experiments mainly demonstrate that VMP and CEP exhibit nearly identical performance under the conditions assumed in Theorem 1. However, it would be more insightful to investigate whether the performance remains comparable---and whether CEP continues to converge---when some of these conditions are violated. Such robustness analysis would strengthen the empirical support for the theory.*
>
> **R2:** Thank you for your valuable suggestion. In response, we have added two new experiments in the main text to evaluate the empirical behavior of VMP and CEP when the conditions of Theorem 1 are not satisfied. These experiments aim to assess the robustness of both methods beyond the theoretical assumptions.
>
> The first experiment investigates the case where the group-wise update assumption in Theorem 1 is violated. While the theorem assumes that CEP updates are performed by first aggregating messages for each group of variables, standard EP and streaming CEP implementations typically update variable groups sequentially, one data point at a time. In the experiment, we modify CEP to use this entry-wise update strategy.
> As discussed in the main text (see Figure 7), the results show that CEP still converges reliably under this setup, despite the lack of a formal convergence guarantee. Moreover, we observe that entry-wise updates lead to faster convergence in terms of iteration count, though with increased computational cost per iteration.
>
> The second experiment examines the robustness of VMP and CEP under non-i.i.d. noise. We introduce heteroscedastic Gaussian noise by sampling the standard deviation for each data point $i$ from a standard normal distribution, $\sigma_i \sim \mathcal{N}(0, 1)$, and then generating the noise as $\mathcal{N}(0, \sigma_i^2)$. The results, presented in Figure 7 of the main text, show that both VMP and CEP maintain nearly identical performance, suggesting that the empirical stability of the methods extends beyond the theoretical conditions of Theorem 1.
>
> **C3.** *Certain parts of the paper, such as the detailed discussion of Lemmas 1 and 2, could be streamlined or moved to an appendix, as they may not warrant such prominence in the main text. Additionally, some figures (e.g., Figures 2-4) are not particularly informative and may not meet the standards expected in a published paper. Improving figure clarity and focusing on essential visualizations would enhance the overall presentation.*
>
> **R3:** Thank you for the constructive feedback. In line with your suggestion, we have moved Subsection 2.4, which contains Lemmas 1 and 2, to the appendix. This change helps improve the clarity and flow of the main text by reducing technical detail in the early sections.
>
> Figures 2 and 3 illustrate typical graphical models used in the VMP and CEP algorithms. These visualizations are intended to support readers who may be less familiar with graphical model structures by providing intuitive explanations of the architectures and definitions. We believe that such figures are useful for accessibility. That said, we agree that their presentation can be improved. In the revised manuscript, we have merged Figures 2 and 3 into a single figure to reduce redundancy and improve clarity.
>
> Regarding Figure 4, it serves as a roadmap for the theoretical results presented in the paper. We believe it plays an important role in helping readers follow the structure and logic of the analysis. To address your concern, we have revised the figure for better readability and have added more explanations to clarify its components.

---

> ### Author Response · Authors · 2025-07-25
>
> **C4.** *[Minor] Based on the convergence result for CEP, the authors recommend assessing whether the model satisfies the conjugate-exponential assumption with independent variable partitions as a first step. However, this type of assumption-checking is rarely performed in practice, as most methods rely on assumptions that are often taken for granted rather than explicitly validated. Requiring such validation may limit the practical usability of the proposed guidance. Moreover, since the stated conditions are sufficient (but not necessary), it is likely that the streaming ABI algorithm could still perform well even when these conditions are not strictly met.*
>
> **R4:** Thank you for the insightful comment. We agree that many inference methods are based on modeling assumptions that are often adopted without explicit validation. However, in the context of our theoretical analysis, we would like to clarify that verifying whether a model belongs to the conjugate-exponential family with independent variable partitions is generally straightforward in practice.
>
> The structure of conjugate-exponential models has been extensively studied, and the conjugate relationships for common likelihood-prior pairs are well-documented in the literature. For example, the original VMP paper provides a summary table of such conjugate pairs (see Table 1 in [Winn, 2006]), which serves as a practical reference. Moreover, it is common in practice to deliberately construct models within the conjugate-exponential family to ensure analytical tractability and enable closed-form updates. As a result, checking whether these conditions are satisfied typically requires minimal effort.
>
> That said, we agree with your observation that the assumptions in Theorem 1 are sufficient but not necessary. As noted in our response to Comment C2, we have added experiments that examine settings where these conditions are not strictly met. The results show that VMP and CEP continue to perform well, and their theoretical connection remains empirically valid. This robustness further supports the practical utility of our analysis and the guidance derived from it.

---

> > ### Comment · Reviewer_TSkv · 2025-08-08
> >
> > I would like to thank the authors for their detailed answers to my questions and for revising the manuscript accordingly!
> >
> > The additional experiments are particularly interesting; the observation that the methods maintain comparable performance even when some conditions of Theorem 1 are violated could provide valuable insights for future work.
> >
> > I will recommend this paper for acceptance if the authors properly address the questions raised by Reviewer X5pz.

---

### Decision · Action_Editor_pXDY · 2025-09-03

**Recommendation:** Reject

**Audience:**

Yes

**Audience Explanation:**

This paper should be interesting to probabilistic modelling community.

**Claims And Evidence:**

No

**Claims Explanation:**

This paper's contribution is mainly on the theoretical side: the authors have attempted to provide a connection between conditional expectation propagation (CEP) and variational message passing (VMP).

While reviewers welcomed the theoretical contribution, a major flaw for the main theoretical result was pointed out by an expert reviewer. The authors have provided major revisions based on the reviewer's suggestions, however, this revision of the theoretical result may lead to different indications (compared with the discussions in the original presentation). Based on this, I recommend the authors to revise and re-submit the manuscript to TMLR.

PS -- there has been papers on connecting VI and EP via divergence minimisation perspective, see e.g., https://arxiv.org/abs/1602.02311. It would be beneficial to discuss the paper's contribution as compared with such line of work.

**Resubmission Of Major Revision:**

The authors may consider submitting a major revision at a later time.